# Towards Efficient Optimizer Design for LLM via Structured Fisher Approximation with a Low-Rank Extension

**Wenbo Gong**[*]
Microsoft Research
wenbogong@microsoft.com

**Meyer Scetbon**[*†]
Citadel Securities
meyerscetbon@gmail.com

**Chao Ma**[*]
Microsoft Research
chao.ma@microsoft.com

**Ted Meeds**
Microsoft Research
Ted.Meeds@microsoft.com

## Abstract

Designing efficient optimizers for large language models (LLMs) with low-memory requirements and fast convergence is an important and challenging problem. This paper makes a step towards the systematic design of such optimizers through the lens of structured Fisher information matrix (FIM) approximation. We show that many state-of-the-art efficient optimizers can be viewed as solutions to FIM approximation (under the Frobenius norm) with specific structural assumptions. Building on these insights, we propose two design recommendations of practical efficient optimizers for LLMs, involving the careful selection of structural assumptions to balance generality and efficiency, and enhancing memory efficiency of optimizers with general structures through a novel low-rank extension framework. We demonstrate how to use each design approach by deriving new memory-efficient optimizers: Row and Column Scaled SGD (RACS) and Adaptive low-dimensional subspace estimation (Alice). Experiments on LLaMA pre-training (up to 1B parameters) validate the effectiveness, showing faster and better convergence than existing memory-efficient baselines and Adam with little memory overhead. Notably, Alice achieves better than $2\times$ faster convergence over Adam, while RACS delivers strong performance on the 1B model with SGD-like memory.

## 1 Introduction

Adaptive optimizers are critical in training large language models (LLMs). Yet, as models and datasets continue to grow, it brings several implications for training, including increased GPU requirements or reduced per-device batch size, which lowers overall training throughput. Adam, for instance, triples memory requirements due to the storage of two internal exponential moving average (EMA) states, while other optimizers (Gupta et al., 2018; Vyas et al., 2024) with faster convergence (in terms of training steps) can further inflate the total memory. Meanwhile, some memory efficient optimizers, like stochastic gradient descent (SGD), fail to train the LLMs effectively. Thus, designing efficient optimizers has become increasingly important.

There are several promising lines of research that have advanced one or more of these aspects of efficiency: (1) removing the optimizer state (Ma et al., 2024; Jordan et al., 2024; Zhang et al., 2024; Xu et al., 2024; Zhu et al., 2024), or (2) using ad-hoc low-rank design to reduce memory (Hu et al., 2021; Lialin et al., 2023; Zhao et al., 2024a; Chen et al., 2024a; Si et al., 2024). However, developing new efficient optimizers remains challenging. This paper explores structured FIM approximation as a potential framework for this task.

To demonstrate the effectiveness of this framework, we begin by showing that many existing optimizers and gradient operators, including Adam, Shampoo, gradient normalization and whitening

---

[*]Equal contribution
[†]This work is done when Meyer was affiliated with Microsoft Research

(Zhang et al., 2024; Gupta et al., 2018; Vyas et al., 2024; You et al., 2019; Ma et al., 2024; Jordan et al., 2024), can be recast under structured FIM approximation. We then go on to show how to derive two generalizations of Adam[1] with more relaxed structural assumptions, named One-sided SOAP (OSOAP) and SOAP/AdaDiag++ (Vyas et al., 2024; Nguyen et al., 2025). Previously, OSOAP is considered as a simple one-side projection version of SOAP, but we reveal that they correspond to different underlying approximation structures. Although this framework provides a clear link between structures and optimizers, working with more general structures can come at the cost of efficiency. For example, SOAP can require 7 times more memory than SGD.

Building on these insights, our first design recommendation proposes to choose structural assumptions that balance generality with practical efficiency. We demonstrate this by choosing a structure that generalizes gradient normalization, leading to a new efficient optimizer, Row and Column Scaled SGD (RACS), with SGD-like memory requirements.

For optimizers with more general structures it may not be always possible to achieve such a balance. Instead of rejecting their potentials, our second design recommendation proposes to apply a novel low-rank extension framework to improve their efficiency. This framework consists of three steps that can convert full-rank optimizers into their low-rank approximations with reduced memory and computational costs. We demonstrate this by deriving a low-rank extension of OSOAP, called Adaptive low-dimensional subspace estimation (Alice). We also prove its convergence under the continuous-time setup.

We emphasize that the main contribution is the unifying viewpoint that allows a systematic design for efficient optimizers. RACS and Alice are two examples. More novel optimizers can be derived by (1) designing new structural approximations; or (2) adapting existing optimizers with our low-rank extension. This allows one to go beyond the ad hoc design and focus on a well grounded strategy.

With pre-training LLaMA (Touvron et al., 2023) with C4 dataset (Raffel et al., 2020), we demonstrate that RACS and Alice outperform Adam and several memory-efficient baselines. Alice achieves better than $2\times$ speed-ups compared to Adam, and RACS performs strongly on pre-training the 1B LLaMA.

To summarize, our contributions are:

- We propose structured FIM approximation as a practical framework for optimizer design and show that several existing optimizers can be viewed as special cases within this framework.

- We propose to design optimizers by choosing structures that balance generality and efficiency and demonstrate it with a new optimizer, RACS.

- We present a low-rank extension framework to convert full-rank optimizers and demonstrate it with a new optimizer Alice. We also prove its convergence property.

- We demonstrate the effectiveness of RACS and Alice on LLaMa pre-training tasks when compared to Adam and other baselines.

Fig. 1 and Fig. 2 summarize the main relationships among optimizers and the procedure of our proposed low-rank extension framework.

**Scope of this paper**    We want to emphasize and clearly define the scope of this paper. Specifically, we **DO NOT** aim to provide the definitive and complete answers on how to design an optimizer for LLM and their theoretical properties. Instead, our primary contribution is to establish that structured FIM approximation is a **potential candidate** as a unifying design principle for LLM optimizers, followed by two design guidelines with exampled efficient optimizers. In summary, we aim to provide a potential direction for future LLM optimizer research.

---

[1]Generality of structural assumption defines how relaxed this assumption is. We say structure $A$ is more general than $B$ iff. $B$ can be recovered by applying further constraints on $A$. The general structures tend to give better approximations to FIM than the less general one.

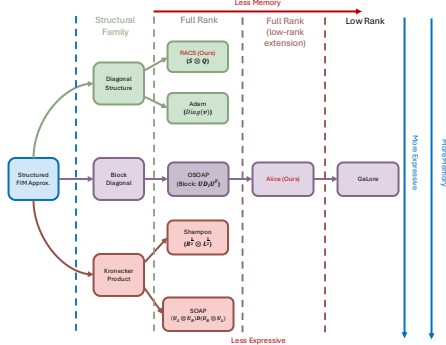

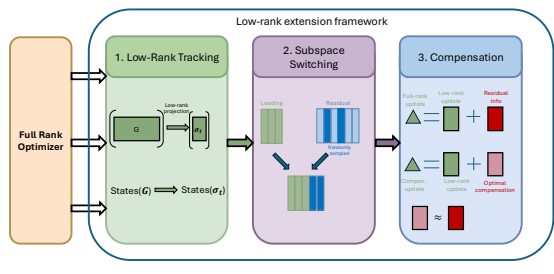

Figure 1: Summary of relationship between different optimizers and their corresponding structural assumptions.

Figure 2: Summary of the low-rank extension framework to convert full-rank optimizer into its low-rank counterpart.

## 2 PRELIMINARIES

### 2.1 BASIC NOTATIONS AND SETUP

Throughout the paper we consider 2D matrix parameters $\boldsymbol{W}$ (i.e. layer weights) of size $m \times n$ and $m \leq n$; the operation $\text{Vec}(\cdot)$ vectorizes the input matrix by stacking its columns; $\text{Mat}(\cdot)$ is the inverse of $\text{Vec}(\cdot)$ and reshapes the vector back into a matrix. We use $\mathcal{L}_\theta$ as the loss function where $\theta = \text{Vec}(\boldsymbol{W})$. $\boldsymbol{G} = \nabla_{\boldsymbol{W}}\mathcal{L}$ is the matrix gradient and $\vec{g}$ is the vectorized gradient, i.e. $\vec{g} = \text{Vec}(\boldsymbol{G})$. $\boldsymbol{g}_i$ denotes the $i^{\text{th}}$ column of $\boldsymbol{G}$. In the paper, we assume $\boldsymbol{W}$ are parameters of one layer. $\otimes$ indicates the Kronecker product. By default, $\boldsymbol{M}^2$ and $\sqrt{\boldsymbol{M}}$ indicates the elementwise multiplication and square-root. $\text{EVD}(\boldsymbol{M}, r)$ performs the eigen-value decomposition (EVD) and keeps the top $r$ eigenvectors ordered by the descending eigenvalues. If $r$ is omitted, we keep all eigenvectors. $\text{QR}(\boldsymbol{M})$ with orthonormal $\boldsymbol{M} \in \mathbb{R}^{m \times r}$ obtains the remaining $m - r$ basis via QR decomposition.

We also introduce the following diagonal operations: $\text{Diag}(\boldsymbol{M})$ will extract the diagonals of $\boldsymbol{M}$ to a vector. $\text{Diag}_{\text{B}}(\boldsymbol{M}_1, \ldots, \boldsymbol{M}_n)$ will stack the input sequence of matrices to form a larger block diagonal matrix. $\text{Diag}_{\text{v}}(\boldsymbol{v})$ will expand the input vector $\boldsymbol{v}$ to a pure diagonal matrix. $\text{Diag}_{\text{M}}(\boldsymbol{M})$ will expand the input $\boldsymbol{M}$ to a larger pure diagonal matrix by stacking the element of $\boldsymbol{M}$ in a column-wise order. We demonstrate the above operations with examples in Sec. C. Note that results in this paper may involve the expectation $\mathbb{E}[\cdot]$, we assume the use of an EMA as its practical estimation.

### 2.2 FISHER INFORMATION AND NATURAL GRADIENT DESCENT

Fisher information is a fundamental concept in statistics that measures the amount of information a random variable carries about a parameter of interest. In this paper, we ignore dependence between layers and treat each layer independently. Under the context of LLMs, with the vectorized mini-batched gradient of one layer $\vec{g}$, we define the empirical FIM for that layer as $\boldsymbol{F} = \mathbb{E}[\vec{g}\vec{g}^T]$. Natural gradient descent (NGD) leverages the inverse of FIM to smooth the local geometry to improve convergence and stabilize training (Martens, 2020). In practice, the square-root inverse of FIM may be preferred due to its properties and improved performance on certain optimization problems (Yang & Laaksonen, 2008; Lin et al., 2024; Loshchilov & Hutter, 2016; Bergstra & Bengio, 2012; Choi, 2019). The corresponding update of $\boldsymbol{W}$ is:

$$\boldsymbol{W} \leftarrow \boldsymbol{W} - \lambda \, \text{Mat}(\boldsymbol{F}^{-\frac{1}{2}} \nabla_\theta \mathcal{L}). \tag{1}$$

Due to the large size of $\boldsymbol{F} \in \mathbb{R}^{mn \times mn}$, computing its inverse is a significant impediment to applying this update rule in practice. One way to address this is to enforce certain structures to approximate FIM. In this paper, we consider two main classes of structures: block diagonal and Kronecker product matrices. They possess favorable properties that can significantly reduce computational complexity. Sec. D.2 provides a brief introduction and summary of their key properties. We also include a comprehensive background in Sec. D to be more self-contained.

Table 1: The summary of underlying structure assumptions of existing and newly proposed optimizers, along with practical efficiency. "Generalizes" refers to the optimizer whose structure is generalized, e.g. OSOAP's structure generalizes Adam (see Sec. I.1 for further details). Optimizers in blue color are new. The "Computation" is the cost of per-step of the optimizer update. We assume $n \geq m \gg r$.

| | Adam | Shampoo | OSOAP/AdaDiag | SOAP/AdaDiag++ | GaLore | RACS | Alice |
|---|---|---|---|---|---|---|---|
| Stucture | $\mathrm{Diag}_v(\boldsymbol{v})$ | $\boldsymbol{R}_n^{\frac{1}{2}} \otimes \boldsymbol{L}_m^{\frac{1}{2}}$ | $\mathrm{Diag}_B(\{\boldsymbol{U}_f\boldsymbol{D}_i\boldsymbol{U}_f^T\}_i)$ | $(\boldsymbol{U}_R \otimes \boldsymbol{U}_L)\tilde{\boldsymbol{D}}(\boldsymbol{U}_R \otimes \boldsymbol{U}_L)^T$ | Approx. Alice | $\boldsymbol{S} \otimes \boldsymbol{Q}$ | Low-rank to Gadam |
| Generalizes | N/A | N/A | Adam | Gadam + Shampoo | N/A | Normalization | GaLore |
| Computation | $O(mn)$ | $O(m^3 + n^3)$ | $O(m^3)$ | $O(m^3 + n^3)$ | $O(mnr + m^2n/K)$ | $O(mn)$ | $O(mnr + m^2r/K)$ |
| Memory | $3mn$ | $mn + m^2 + n^2$ | $3mn + 2m^2$ | $3mn + 2m^2 + 2n^2$ | $mn + 2nr + mr$ | $mn + m + n + 1$ | $mn + 2nr + mr + n + r^2$ |
| Full-rank update | ✓ | ✓ | ✓ | ✓ | ✗ | ✓ | ✓ |
| Section | Sec. 3.1 | Sec. 3.2 | Sec. 3.4 | Sec. E | Sec. D.11 | Sec. 4 | Sec. 5.4 |

# 3 STRUCTURAL APPROXIMATION TO FIM

The structured FIM approximation framework consists of three steps: first, choose a structure to approximate FIM; second, find the solution $\tilde{\boldsymbol{F}}$ by minimizing the following objective:

$$\min_{\tilde{\boldsymbol{F}} \in \mathcal{H}} \|\tilde{\boldsymbol{F}} - \boldsymbol{F}\|_F^2, \tag{2}$$

where $\boldsymbol{F}$ is the empirical FIM, $\mathcal{H}$ is the matrix family corresponding to the structural assumption; third, derive the square-root NGD update with approximated $\tilde{\boldsymbol{F}}$. In this section, we show that many existing optimizers and gradient operators can be reformulated within this framework by varying structural assumptions, thereby establishing connections between the structure and optimizers.

## 3.1 ADAM: PURELY DIAGONAL STRUCTURE

When Adam was originally proposed, it was argued that the purpose of the second moment was to approximate the diagonal elements of FIM (Kingma, 2014; Hwang, 2024).

**Proposition 3.1** (diagonal approximation). *Assuming $\mathcal{H} = \{\mathrm{Diag}_v(\boldsymbol{v}); v_i > 0\}$, then Eq. (2) has analytic solution $\tilde{\boldsymbol{F}}^* = \mathrm{Diag}_v(\mathbb{E}[\vec{g}^2])$ where $\vec{g}^2$ indicates the element-wise square of $\vec{g} = \mathrm{Vec}(\boldsymbol{G})$.*

It is trivial to show the resulting square-root NGD recovers Adam's second moment when using the EMA to estimate $\mathbb{E}$. Together with the first moment, one can recover Adam.

## 3.2 SHAMPOO: KRONECKER PRODUCT STRUCTURE

Although previous work (Morwani et al., 2024) has demonstrated the connection of Shampoo (Gupta et al., 2018) to the Kronecker product approximation through power iteration algorithm, we provide an alternative view: Shampoo's pre-conditioner can be derived by minimizing an upper bound of Eq. (2) with

$$\mathcal{H} = \{\boldsymbol{R}_n^{\frac{1}{2}} \otimes \boldsymbol{L}_m^{\frac{1}{2}}; \boldsymbol{R}_n \in \mathbb{R}^{n \times n}, \boldsymbol{L}_m \in \mathbb{R}^{m \times m}\}$$

where $\boldsymbol{R}_n$ and $\boldsymbol{L}_m$ are symmetric positive definite (SPD) matrices.

**Theorem 3.2** (Shampoo pre-conditioner). *Assume the above structural assumption, then we have an upper bound of Eq. (2)*

$$\|\tilde{\boldsymbol{F}} - \boldsymbol{F}\|_F^2 \leq 3(mn\|\boldsymbol{A}^2 - \boldsymbol{C}^2\|_F\|\boldsymbol{B}^2 - \boldsymbol{C}^2\|_F + \sqrt{mn}\|\boldsymbol{C}\|_F^2(\|\boldsymbol{A}^2 - \boldsymbol{C}^2\|_F + \|\boldsymbol{B}^2 - \boldsymbol{C}^2\|_F))$$

*where $\boldsymbol{A} = \boldsymbol{R}_n^{\frac{1}{2}} \otimes \boldsymbol{I}_m$, $\boldsymbol{B} = \boldsymbol{I}_n \otimes \boldsymbol{L}_m^{\frac{1}{2}}$, $\boldsymbol{C} = \mathbb{E}[\vec{g}\vec{g}^T]^{\frac{1}{2}}$, and $^2$ indicates the matrix square. Minimizing the upper-bound admits $\boldsymbol{R}_n^* = \frac{1}{m}\mathbb{E}[\boldsymbol{G}^T\boldsymbol{G}]$ and $\boldsymbol{L}_m^* = \frac{1}{n}\mathbb{E}[\boldsymbol{G}\boldsymbol{G}^T]$.*

Sec. F.1 shows that the corresponding square-root NGD leads to the Shampoo's update formula. Therefore, the structure behind Shampoo is the Kronecker product of two square-root SPD matrices.

## 3.3 NORMALIZATION AND WHITENING: SIMPLE BLOCK DIAGONAL STRUCTURES

For an input $\boldsymbol{G}$, the whitening and normalization operator are defined as

$$\mathrm{Whitening}(\boldsymbol{G}) = (\boldsymbol{G}\boldsymbol{G}^T)^{-\frac{1}{2}}\boldsymbol{G}; \quad \mathrm{Norm}(\boldsymbol{G}) = \boldsymbol{G}\boldsymbol{S}^{-\frac{1}{2}}$$

where $\mathrm{Diag}(\boldsymbol{S})$ contains the squared column $l_2$ norm of $\boldsymbol{G}$. Next, we show that these two operators also corresponds to minimizing Eq. (2) with different structural assumption.

**Proposition 3.3** (Normalization and whitening). *With $\mathcal{H} = \{\boldsymbol{I}_n \otimes \boldsymbol{M}\}$ and $\mathcal{H} = \{\boldsymbol{S} \otimes \boldsymbol{I}_m; S_{ii} > 0, \forall i\}$, where $\boldsymbol{M} \in \mathbb{R}^{m \times m}$ is SPD and $\boldsymbol{S} \in \mathbb{R}^{n \times n}$ is a positive diagonal matrix. Then, minimizing Eq. (2) admits*

$$\boldsymbol{M}^* = \frac{1}{n}\mathbb{E}[\boldsymbol{G}\boldsymbol{G}^T] \tag{3}$$

$$\boldsymbol{S}^* = \frac{1}{m}\mathbb{E}[\text{Diag}_\text{v}(\boldsymbol{g}_1^T\boldsymbol{g}_1, \ldots, \boldsymbol{g}_n^T\boldsymbol{g}_n)] \tag{4}$$

The proof can be found in Sec. G.3, where we prove a much more general solution (Theorem G.4), and Theorem 3.3 is a special case. The square-root NGD update with Eq. (3) and Eq. (4) are $\text{Mat}(\tilde{\boldsymbol{F}}^{-\frac{1}{2}}\vec{g}) = \sqrt{n}\mathbb{E}[\boldsymbol{G}\boldsymbol{G}^T]^{-\frac{1}{2}}\boldsymbol{G}$ and $\sqrt{m}\boldsymbol{G}\boldsymbol{S}^{*-\frac{1}{2}}$, respectively (refer to Sec. F.2 for derivation). We can view $\text{Whitening}(\cdot)$ and $\text{Norm}(\cdot)$ as special cases with one-sample estimate for $\mathbb{E}$.

**Muon and many others are special cases** Many recently proposed optimizers, such as Muon, SWAN, LARS, and LAMB (Jordan et al., 2024; Ma et al., 2024; You et al., 2017; 2019), rely on normalization and/or whitening. These gradient operators improve convergence (You et al., 2017; 2019; Jordan et al., 2024) and can remove Adam's internal states (Ma et al., 2024). In fact, Muon simply whitens the gradient, and thus is a special case of this framework. Interestingly, the original Muon lacks a $\sqrt{n}$ coefficient predicted by Theorem 3.3. This was later added back by (Liu et al., 2025), which is proved to be critical for LLM training. We include a thorough discussion in Sec. I.5

### 3.4 OSOAP AND SOAP: TWO GENERALIZATIONS OF ADAM

All structures considered above are do not strictly generalize Adam's purely diagonal structure. Next, we consider two structures that strictly generalize Adam, normalization, and whitening: a block diagonal matrix with a shared eigen-space; and its combination with the Kronecker product.

First, consider a block-diagonal structure that generalizes Adam:

$$\mathcal{H} = \{\text{Diag}_\text{B}(\boldsymbol{M}_1, \ldots, \boldsymbol{M}_n); \boldsymbol{M}_i = \boldsymbol{U}_f\boldsymbol{D}_i\boldsymbol{U}_f^T\} \tag{5}$$

where $\boldsymbol{U}_f$ defines a shared **full-rank** eigen-space, and $\boldsymbol{D}_i$ is an eigenvalue matrix. Adam is a special case by constraining $\boldsymbol{U}_f = \boldsymbol{I}$. The structures in Sec. 3.3 are also special cases by setting $\boldsymbol{D}_i = \boldsymbol{D}$ (whitening); and $\boldsymbol{U}_f = \boldsymbol{I}, \boldsymbol{D}_i = s_i\boldsymbol{I}$ (normalization). However, this structure does not lead to an analytic solution for Eq. (2). Instead, we propose an approximation by solving a 1-step refinement.

**Theorem 3.4** (1-step refinement). *For the structure in Eq. (5), we consider the following 1-step refinement: (i) assuming all $\boldsymbol{D}_i$ are equal, and find $\boldsymbol{U}_f$; (ii) fix the $\boldsymbol{U}_f$ and find $\boldsymbol{D}_i$. Then, step (i) admits the analytic solution from Theorem 3.3:*

$$\boldsymbol{U}_f = \text{EVD}(\mathbb{E}[\boldsymbol{G}\boldsymbol{G}^T]). \tag{6}$$

*where* EVD *is the eigenvalue decomposition; step (ii) admits the analytic solution:*

$$\tilde{\boldsymbol{D}} = \text{Diag}_\text{M}(\mathbb{E}[(\boldsymbol{U}_f^T\boldsymbol{G})^2]) \tag{7}$$

*where* $\tilde{\boldsymbol{D}} = \text{Diag}_\text{B}(\boldsymbol{D}_1, \ldots, \boldsymbol{D}_n)$.

Based on this result, we can derive the corresponding square-root NGD(refer to Sec. F.3):

$$\text{Mat}(\tilde{\boldsymbol{F}}^{-\frac{1}{2}}\vec{g}) = \boldsymbol{U}_f\frac{\boldsymbol{U}_f^T\boldsymbol{G}}{\sqrt{\mathbb{E}[(\boldsymbol{U}_f^T\boldsymbol{G})^2]}}. \tag{8}$$

This can be viewed as applying Adam's update on a space "rotated" by $\boldsymbol{U}_f$. Consequently, we can design an optimizer based on Eq. (8), called OSOAP. The following procedures describes its update:

$$\boldsymbol{m}_t = \beta_1\boldsymbol{m}_{t-1} + (1 - \beta_1)\boldsymbol{G}_t; \quad \boldsymbol{Q}_t = \beta_3\boldsymbol{Q}_{t-1} + (1 - \beta_3)\boldsymbol{G}_t\boldsymbol{G}_t^T; \quad \boldsymbol{U}_{f,t} = \text{EVD}(\boldsymbol{Q}_t);$$

$$\boldsymbol{v}_t = \beta_2\boldsymbol{v}_{t-1} + (1 - \beta_2)(\boldsymbol{U}_{f,t}^T\boldsymbol{G}_t)^2; \quad \Delta_t = \boldsymbol{U}_{f,t}\frac{\boldsymbol{U}_{f,t}^T\boldsymbol{m}_t}{\sqrt{\boldsymbol{v}_t}} \tag{9}$$

Algorithm 7 in Sec. D.6 summarizes its algorithm. In Vyas et al. (2024), they regard OSOAP and SOAP as being from the same optimizer family and the former as the one-sided projection verion of the latter. In the following, we will show SOAP and OSOAP correspond to different underlying structure assumptions, indicating that OSOAP should not be simply treated as the cheaper approximation to SOAP.

**SOAP/AdaDiag++** Despite the structural generality of OSOAP, it only focuses on the block diagonals of FIM. To go beyond this, one can combine the structure of OSOAP with Shampoo's Kronecker product to obtain a structure that generalizes all the others in this paper. Interestingly, this turns out to be SOAP/AdaDiag++. Refer to Sec. E for details.

## 4 RACS: MEMORY-EFFICIENT OPTIMIZER FROM A CAREFULLY SELECTED STRUCTURE

The structured FIM approximation reveals two important insights: there exists a correspondence between structural assumption and optimizers, and structural generality often comes at the cost of practical efficiency. For example, while the structures of OSOAP and SOAP offer more accurate FIM approximations, they require expensive computation and memory consumption (Table 1). Building on this, our first design recommendation is to **select structures that balance generality and efficiency.**

To demonstrate this, we select a structure that generalizes gradient normalization, which scales both the rows and columns simultaneously:

$$\mathcal{H} = \{\boldsymbol{S} \otimes \boldsymbol{Q}\} \tag{10}$$

where $\boldsymbol{S} \in \mathbb{R}^{n \times n}$, $\boldsymbol{Q} \in \mathbb{R}^{m \times m}$ are positive diagonal matrices. The optimal solution of Eq. (2) can be solved by a fixed-point procedure:

**Proposition 4.1** (Two-sided scaling). *Assuming the structure of Eq. (10), and $\mathbb{E}[\boldsymbol{G}^2]$ contains only positive values, solving Eq. (2) admits an iterative fixed point procedure:*

$$\boldsymbol{s} = \frac{\mathrm{Diag}\left(\mathbb{E}[\boldsymbol{G}^T \boldsymbol{Q} \boldsymbol{G}]\right)}{\|\boldsymbol{Q}\|_F^2}, \quad \boldsymbol{q} = \frac{\mathrm{Diag}\left(\mathbb{E}[\boldsymbol{G} \boldsymbol{S} \boldsymbol{G}^T]\right)}{\|\boldsymbol{S}\|_F^2}. \tag{11}$$

*where $\boldsymbol{s} = \mathrm{Diag}(\boldsymbol{S})$, $\boldsymbol{q} = \mathrm{Diag}(\boldsymbol{Q})$. Additionally, the fixed point solution $\boldsymbol{s}$, $\boldsymbol{q}$ converges to the right and left principal singular vector of $\mathbb{E}[\boldsymbol{G}^2]$ up to a scaling factor with unique $\boldsymbol{S} \otimes \boldsymbol{Q}$.*

The corresponding square-root NGD update scales both rows and columns through $\boldsymbol{S}$ and $\boldsymbol{Q}$ (i.e. $\mathrm{Mat}(\tilde{\boldsymbol{F}}^{-\frac{1}{2}}\vec{\boldsymbol{g}}) = \boldsymbol{Q}^{-\frac{1}{2}}\boldsymbol{G}\boldsymbol{S}^{-\frac{1}{2}}$). We name this optimizer, Row and Column Scaled SGD (RACS) (Algorithm 3 in Sec. B). Although analytic solutions of $\boldsymbol{s}$, $\boldsymbol{q}$ exist, we perform 5 steps of Eq. (11) as an efficient approximation. To further stabilize training, we also incorporate the norm-growth limiter used in (Chen et al., 2024a). RACS is highly memory efficient since it only needs the storage of two diagonal matrices $\boldsymbol{S}$ and $\boldsymbol{Q}$ and a scalar for the limiter, consuming $m + n + 1$ memory. In Sec. I.5 we discuss connections to Adapprox, Apollo and Adafactor (Zhao et al., 2024b; Zhu et al., 2024; Shazeer & Stern, 2018). The above structure may not always be easy to find. Therefore, our next design guideline based on the novel low-rank extension is more widely applicable.

## 5 ALICE: MEMORY-EFFICIENT OPTIMIZER FROM LOW-RANK EXTENSION FRAMEWORK

We propose a novel low-rank framework consisting of three steps, low-rank **tracking**, subspace **switching**; and **compensation**. Tracking aims to reduce the memory cost of EMA states, whereas switching and compensation are designed to correct the potential issues caused by low-rank approximations. We demonstrate these by converting OSOAP to its low-rank version, Alice. While the procedure could be applied to SOAP in a similar manner, we leave this for future work.

### 5.1 TRACKING: LOW-RANK PROJECTIONS TO REDUCE MEMORY

By carefully examining the OSOAP's procedure (Eq. (9)), we notice the key internal state of OSOAP is the projection $\boldsymbol{U}_{f,t}$. To reduce the memory cost, we propose low-rank $\boldsymbol{U}_t$ by keeping only the top $r$ eigenvectors, and denote the remaining $m - r$ basis as $\boldsymbol{U}_{c,t}$ (i.e. $\boldsymbol{U}_{f,t} = [\boldsymbol{U}_t, \boldsymbol{U}_{c,t}]$). For $\boldsymbol{Q}_t$ we propose the following **low-rank** tracking:

$$\boldsymbol{\sigma}_t = \boldsymbol{U}_t^T \boldsymbol{G}_t; \quad \widetilde{\boldsymbol{Q}}_{t+1} = \beta_3 \widetilde{\boldsymbol{Q}}_t + (1 - \beta_3)\boldsymbol{\sigma}_t \boldsymbol{\sigma}_t^T \tag{12}$$

where $\boldsymbol{\sigma}_t$ is the projected gradient, and $\widetilde{\boldsymbol{Q}}_t$ is the low-rank tracking state. One can easily reconstruct back $\boldsymbol{Q}_t \approx \boldsymbol{U}_t \widetilde{\boldsymbol{Q}}_t \boldsymbol{U}_t^T$ when needed. This reduces the memory from $m^2$ of $\boldsymbol{Q}_t$ to $r^2$ of $\tilde{\boldsymbol{Q}}_t$.

However, this low-rank projection comes with two consequences: (1) the reconstructed state $\boldsymbol{Q}_t$ is no longer accurate; (2) the resulting parameter update $\Delta$ in Eq. (9) ignores the information within $\boldsymbol{U}_{c,t}$. Next, we propose two additional steps, switching and compensation, rooted in theoretical insights to address the two problems, respectively.

## 5.2 SWITCHING: MIXING LEADING BASIS WITH THE COMPLEMENTS

Since the projected gradient $\boldsymbol{\sigma}_t$ only maintains the information in the space spanned by $\boldsymbol{U}_t$, the low-rank tracking state $\widetilde{\boldsymbol{Q}}_t$ necessarily discards information in $\boldsymbol{U}_{c,t}$. Therefore, even if those directions should become the leading basis, $\widetilde{\boldsymbol{Q}}_t$ will ignore their contributions, causing the stability of the previous leading eigenvectors and preventing the exploration of other spaces. This effect is largely ignored by the previous work (Yen et al., 2023; Zhao et al., 2024a) and believe one should always keep the top eigenvectors. Here, we challenge this commonly adopted concept.

Theorem G.8 confirms this and shows that the true tracking state $\boldsymbol{Q}_t$ can be decomposed into a low-rank reconstructed state (i.e. $\boldsymbol{U}_t \widetilde{\boldsymbol{Q}}_t \boldsymbol{U}_t^T$) and a residual term quantifying the importance of $\boldsymbol{U}_{c,t}$. When the residual term becomes dominant, the remaining basis will become the new leading basis. Inspired by this theoretical insight, we propose a practical scheme to replace the original EVD in Eq. (8), called subspace **switching** (Algorithm 1). It deliberately replace some leading eigenvectors with randomly sampled basis, forcing the optimizer to explore other spaces.

---

**Algorithm 1** Subspace switching

1: **Input:** Reconstructed state $\boldsymbol{Q}$, rank $r$, leading basis number $l$, previous low-rank projection $\boldsymbol{U}_{t-1}$
2: $\boldsymbol{U}_t' = \text{Subspace iteration}(\boldsymbol{Q}, \boldsymbol{U}_{t-1})$
3: $\boldsymbol{U}_{t,1}' \leftarrow$ Keep top $l$ eigenvectors of $\boldsymbol{U}_t'$
4: Uniformly sample $r - l$ basis from the complement $\boldsymbol{U}_{c,t}' = \text{QR}(\boldsymbol{U}_t')$ as $\boldsymbol{U}_{t,2}'$
5: Combine basis $\boldsymbol{U} = [\boldsymbol{U}_{t,1}', \boldsymbol{U}_{t,2}']$
6: **Return** $\boldsymbol{U}$

---

**Algorithm 2** Compensation

1: **Input:** $\boldsymbol{G}_t$, projection $\boldsymbol{U}$, previous norm $\boldsymbol{p}$, limiter norm $\phi$, threshold $\gamma$, decay rate $\beta$
2: $\boldsymbol{p} \leftarrow \beta \boldsymbol{p} + (1 - \beta)(\mathbf{1}_m^T \boldsymbol{G}_t^2 - \mathbf{1}_r^T (\boldsymbol{U}^T \boldsymbol{G}_t)^2)$
3: Compute $\boldsymbol{C}_t$ as Eq. (14)
4: $\eta = \gamma / \max\{\frac{\|\boldsymbol{C}_t\|}{\phi}, \gamma\}$ if $\phi > 0$ else 1
5: $\phi = \|\eta \boldsymbol{C}_t\|$
6: $\boldsymbol{C}_t \leftarrow \eta \boldsymbol{C}_t$
7: **Return** $\boldsymbol{C}_t, \boldsymbol{p}, \phi$

---

## 5.3 COMPENSATION: CONVERT LOW-RANK UPDATE TO BE FULL-RANK

Another problem with low-rank projection $\boldsymbol{U}_t$ is the information loss in the resulting parameter update $\Delta$ at each step. The goal of compensation step is to compensate for this information loss with minimal memory overhead. We first show that the full-rank update for $\Delta$ with $\boldsymbol{U}_{f,t}$ in Eq. (8) can be decomposed into the low-rank and complement update controlled by $\boldsymbol{U}_t$ and $\boldsymbol{U}_{c,t}$ (proof in Sec. I.6). We omit the subscript $t$ for simplicity:

$$\Delta = \boldsymbol{U} \frac{\boldsymbol{U}^T \boldsymbol{G}}{\sqrt{\mathbb{E}[(\boldsymbol{U}^T \boldsymbol{G})^2]}} + \underbrace{\boldsymbol{U}_c \frac{\boldsymbol{U}_c^T \boldsymbol{G}}{\sqrt{\mathbb{E}[(\boldsymbol{U}_c^T \boldsymbol{G})^2]}}}_{\text{Mat}(\tilde{\boldsymbol{F}}_c^{-\frac{1}{2}} \vec{\boldsymbol{g}})} \tag{13}$$

where $\tilde{\boldsymbol{F}}_c = \text{Diag}_\text{B}(\boldsymbol{U}_c \boldsymbol{D}_{c,1} \boldsymbol{U}_c^T, \ldots, \boldsymbol{U}_c \boldsymbol{D}_{c,n} \boldsymbol{U}_c^T)$ is the approximated FIM corresponding to the complement basis, $\boldsymbol{D}_{c,i} = \text{Diag}_\text{v}(\mathbb{E}[(\boldsymbol{U}_c^T \boldsymbol{g}_i)^2])$ and $-\frac{1}{2}$ is the square-root pseudo-inverse.

We notice that $\text{Mat}(\tilde{\boldsymbol{F}}_c^{\frac{1}{2}} \vec{\boldsymbol{g}})$ is exactly the square-root NGD with FIM $\tilde{\boldsymbol{F}}_c$. We can directly leverage the FIM view point to approximate $\tilde{\boldsymbol{F}}_c$ with a carefully selected structure that leads to memory-efficient compensation procedures. One choice is the diagonal structure of normalization operator. Based on this, we propose the following compensation at each step:

$$\boldsymbol{C} = \text{Mat}((\boldsymbol{S} \otimes \boldsymbol{U}_c \boldsymbol{U}_c^T) \vec{\boldsymbol{g}}) = \boldsymbol{U}_c \boldsymbol{U}_c^T \boldsymbol{G} \boldsymbol{S} \tag{14}$$

where $S$ is a positive diagonal matrix, $U_c U_c^T G = (G - UU^T G)$ is the gradient information preserved in the remaining basis. This compensation admits an analytic solution of $S$ to Eq. (2) with $\tilde{F}_c$ as the target (Theorem G.9 in Sec. G.6):

$$\text{Diag}(S) = \frac{\sqrt{m-r}}{\sqrt{\mathbb{E}[\mathbf{1}_m^T G^2 - \mathbf{1}_r^T (U^T G)^2]}} \tag{15}$$

Algorithm 2 summarizes the compensation step with the norm growth limiter from (Chen et al., 2024a). $C_t$ will then be used to replace the $\text{Mat}(\tilde{F}_c^{\frac{1}{2}} \vec{g})$ in Eq. (13) for each update step.

## 5.4 ALICE OPTIMIZER AND ITS CONVERGENCE

Based on the above analyses, we propose Alice and a variant Alice without tracking (Alice-0), as low-rank extensions to OSOAP. Interestingly, GaLore, in fact, is an approximation to Alice when disabling tracking, switching and compensation. Our view point offers a connection of GaLore to OSOAP, revealing that it is a low-rank version of a more powerful optimizer than Adam, and providing an explanation of its effectiveness. Algorithm 4 in Sec. B summarizes Alice procedures.

**Convergence of Alice**   To further support the theoretical validity of Alice, we prove its convergence under the continuous-time setup (Maddison et al., 2018; Nguyen et al., 2025; Chen et al., 2023). We leave the general treatment of the convergence to future work. Please refer to Theorem H.2 in Sec. H.

**Connections to structured FIM approximation**   The proposed low-rank framework seems to be separated and independent to the structured FIM viewpoint. On the contrary, they are deeply connected. First, FIM approximation provides a base optimizer as a starting point for the low-rank extension to work on. More importantly, the critical compensation step is entirely inspired the structured FIM viewpoint. Since the information loss from low-rank projection has a NGD format (Eq. (13)) with a FIM-like preconditioner ($\tilde{F}_c$), we ask a principled question: How can we compensate this in an optimal way? Through structured FIM viewpoint, we aim to approximate this $\tilde{F}_c$ with simple structural assumptions, leading to a compensation term with minimal resources requirement.

## 6 EXPERIMENTS

### 6.1 PRETRAINING LLaMA WITH C4 DATASET

**Setup**   We evaluate the proposed RACS, Alice and its variant Alice-0 on pre-training LLaMA (Touvron et al., 2023) on the C4 dataset (Raffel et al., 2020). We train the following model sizes: 60M, 130M, 350M and 1.3B using a similar setup to (Zhao et al., 2024a; Zhu et al., 2024). An important consideration in our experiments is that **all previous low-rank methods rely on full-rank Adam to train the last layer, which is arguably one of the most important layers** (Zhao et al., 2024c). To thoroughly assess their effectiveness, we report performance for both cases for low-rank methods—training the last layer with and without Adam—but prioritize the latter. For baselines, we consider GaLore, Fira, Apollo-mini, Apollo-svd, Muon and Adam. The priority of this paper focuses on the memory-efficiency of optimizers. Therefore, we mainly adopt the training setup from Zhao et al. (2024a), where all training are under BF16 (i.e. model, parameter state, etc.). We leave the exploration of mixed-precision training in future work.

**Main results**   Table 2 reports the pretraining performance in terms of evaluation perplexity. RACS and Alice outperforms the other memory-efficient baselines and full-rank Adam consistently across model sizes. Alice and Alice-0 perform on par with each other, suggesting Alice-0 may be preferred for better memory efficiency. One major advantage of Alice is its fast convergence compared to Adam, and achieves more than $2\times$ speed-ups across model sizes, while maintaining similar memory consumption as GaLore. Fig. 3 shows the eval ppl. for the 1B model. Despite the simplicity of RACS, it performs well for 350M and 1.3B model, and surpasses the other scaling-based baseline, Apollo-mini and Apollo-svd, consistently. From the throughput (TP), we can observe Alice and RACS is not significantly slower than Adam with $15\%$ and $11\%$ drop for 1B model. Considering the speedup effect, we report the effective TP to represent how quickly the optimizers reach a target loss

Table 2: LLaMA pretraining performance. Ppl. is the evaluation perplexity, Mem. is the estimated memory consumption of optimizers. Regarding the last layer, we report ppl of GaLore and Fira for both cases. For baselines, the ppl. in bracket are directly cited from (Zhu et al., 2024), where the last layer is trained by Adam and consumes more memory. We also cite the numbers of Apollo-mini, Apollo-svd from (Zhu et al., 2024); and Muon from (Ma et al., 2024). The speed-up is measured against Adam in terms of steps. The TP is the number of training tokens processed per second, and effective TP is total training token used by Adam divided by time used by the candidate optimizers to reach the final eval ppl of Adam. **Effective TP is equivalent to wall-clock time performance.**

| Methods | 60M | | 130M | | 350M | | 1.3B | |
|---|---|---|---|---|---|---|---|---|
| | Ppl. | Mem. | Ppl. | Mem. | Ppl | Mem. | Ppl. | Mem. |
| AdamW | 33.94 | 0.32G | 25.03 | 0.75G | 19.24 | 2.05G | 16.44 | 7.48G |
| GaLore | 38.91 (34.88) | 0.21G(0.26G) | 27.18 (25.36) | 0.51G(0.57G) | 21.11 (18.95) | 1.2G(1.29G) | 16.60 (15.64) | 4.25G(4.43G) |
| Fira | 33.77 (31.06) | 0.21G(0.26G) | 25.21 (22.73) | 0.51G(0.57G) | 18.93 (17.03) | 1.2G(1.29G) | 15.14 (14.31) | 4.25G(4.43G) |
| Apollo-mini | 31.93 | 0.23G | 23.84 | 0.43G | 17.18 | 0.93G | 14.17 | 2.98G |
| Apollo-svd | 31.26 | 0.26G | 22.84 | 0.57G | 16.67 | 1.29G | 14.10 | 4.43G |
| Muon | 31.6 | 0.27G | 24.59 | 0.59G | 19.30 | 1.49G | 16.08 | 5.23G |
| RACS | 30.25 | 0.23G | 22.67 | 0.43G | **16.51** | 0.93G | **13.52** | 2.98G |
| Alice-0 | **28.83** (29.74) | 0.21G(0.26G) | **21.99** (22.43) | 0.51G | **16.66** (16.43) | 1.2G (1.29G) | **13.97** (13.47) | 4.25G (4.44G) |
| Alice | **28.69** (29.33) | 0.22G(0.26G) | **21.95** (21.79) | 0.53G | **16.61** (16.37) | 1.24G (1.33G) | **13.85** (13.52) | 4.42G (4.6G) |
| Speed-up in steps (Alice) | 2.22x | | 2.00x | | 2.45x | | 2.82x | |
| Throughput(TP)/Effect TP(AdamW) | 97748/97748 | | 82247/82447 | | 63139/63139 | | 53588/53588 | |
| Throughput(TP)/Effect TP (Alice) | 92589/**202058** | | 71583/**141148** | | 58847/**143088** | | 45523/123048 | |
| Throughput(TP)/Effect TP (RACS) | 98238/162423 | | 73233/123116 | | 55970/131372 | | 47488/**129817** | |

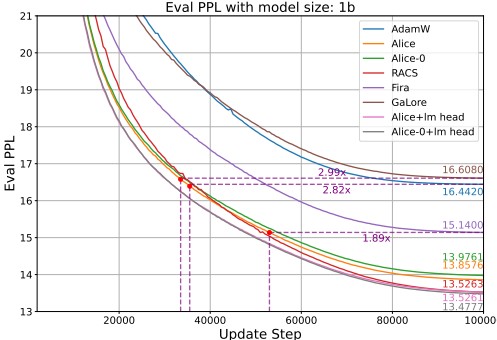

Figure 3: 1B LLaMA eval ppl. curve. "+lm head" represents the last layer is trained by Adam.

| Method | Mem. | 40K | 80K | 120K | 150K |
|---|---|---|---|---|---|
| 8-bit Adam (7B) | 26G | 18.09 | 15.47 | 14.83 | 14.61 |
| 8-bit Galore (7B) | 18G | 17.94 | 15.39 | 14.95 | 14.65 |
| Apollo (7B) | 15.03G | 17.55 | 14.39 | 13.23 | 13.02 |
| Apollo-mini (7B) | 13.53G | 18.03 | 14.60 | 13.32 | 13.09 |
| RACS (1B) | 2.98G | 16.43 | 14.26 | **13.20** | **13.01** |
| Alice (1B) | 4.6G | **15.93** | **14.08** | 13.15 | **12.97** |

Table 3: Eval ppl. at different training steps. For baselines, we cite the number from (Zhu et al., 2024).

| Components | Eval ppl. |
|---|---|
| No tracking, switch, compen. | 26.96 |
| Tracking | 27.35 |
| Tracking+Switch | 25.11 |
| Tracking+Switch+Compen. | 21.95 |

Table 4: Effectiveness of each components.

in wall-clock time. Alice and RACS achieve 123048 and 129817, respectively, compared to 53588 for Adam, resulting in more than 2× faster convergence in wall-clock time to reach Adam's final evaluation perplexity. For the actual memory footprints and additional training curves, see Sec. K.5.

**1B v.s. 7B LLaMA** To further demonstrate the effectiveness, we follow the setup of (Zhu et al., 2024), but train a 1B with Alice and RACS to compare with 7B LLaMA trained by baselines. Table 3 shows that 1B model with our optimizer consistently outperforms 7B model trained with baselines at different training checkpoints. For 150K steps with 8xA100, Apollo requires **15** days while Alice requires **3.8** days. We DO NOT claim 1B can outperform 7B model trained with standard AdamW, but demonstrate the effectiveness of our optimizers compared to other memory-efficient baselines.

## 6.2 ABLATION AND GLUE FINETUNING

**Effect of low-rank tracking and switching strategy** First, we verify whether low-rank tracking (Eq. (12)) is beneficial. As shown in Table 2, Alice-0 performs on par with Alice, suggesting that tracking does not provide a significant boost. Fig. 8(a) indicates that tracking can be helpful when compensation is disabled and used alongside switching. Without switching, tracking leads to inferior performance due to the stability of the eigenspace (Fig. 9). We also evaluate the effectiveness of switching strategy compared to other heuristics: (1) **Gaussian**: $U$ is sampled from a Gaussian distribution; (2) **Gaussian mix**: the leading basis is mixed with basis sampled from a Gaussian matrix; (3) **full basis**: instead of sampling the $r - l$ basis in Algorithm 1 solely from $m - r$ complement

basis, it is sampled jointly from the top $r - l$ and $m - r$ basis. Fig. 8(b) shows that our strategy outperforms the alternatives with clear margins.

**Compensation strategy, other ablations and GLUE finetune**    The closest work to our compensation step is Fira (Chen et al., 2024a), which introduces a heuristic-based compensation term. To evaluate the effectiveness of our optimal compensation, we compare it against Fira compensation and a no-compensation baseline. As shown in Fig. 8(c), our optimal compensation achieves better performance and convergence than all candidates. Table 4 summarizes the contributions of each component. We also performed additional ablations: (1) the effect of the last layer, (2) the effect of EMA in RACS, and (3) hyperparameter sensitivity analysis, and GLUE (Wang, 2018) finetuning experiment. For more details, see Sec. K.6 and Sec. K.7 for details.

## 7    CONCLUSION AND LIMITATIONS

In this paper, we take a step toward the systematic design of efficient optimizers for LLMs through structured FIM approximation. We establish the connection between structural assumptions and optimizers. Building on this, we propose two design approaches: (1) simple structure that balances generality and efficiency; (2) low-rank extension framework for more general structures. These are demonstrated with novel optimizers, RACS and Alice, respectively.

Despite the theoretical and empirical results, this paper does not aim to provide a complete recipe but serves as a starting point for future research. For example, a complete convergence treatment for all optimizers under this FIM framework is an important research question that is currently lacking in this paper. Also, how to incorporate the model structure and task into the optimizer design is another promising direction.

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

## A  RELATED WORK

**Optimizer based on structural approximation**  Due to the desirable properties and convergence of second-order optimization, various work has been proposed to efficiently approximate Hessian-like matrix, e.g. FIM. KFAC (Martens & Grosse, 2015) was one of the first work that goes beyond the simple diagonal approximations, and approximate the layer-wise FIM. Subsequent works extends KFAC beyond MLP layers (Grosse & Martens, 2016; Martens et al., 2018). Further refinement to KFAC are also proposed, including the refinement of eigenvalues (George et al., 2018), fixing the trace (Gao et al., 2021), and refinement by Kronecker product singular value decomposition (Koroko et al., 2022). Our proposed view point is different from KFAC, which express the FIM using the back-proped gradients and layer input. In addition, KFAC needs to re-derived for different type of layers. On the other hand, our proposed view point is closer to another line of work, aiming to approximate the full AdaGrad (Duchi et al., 2011). In particular, Shampoo (Gupta et al., 2018; Anil et al., 2020) is proposed as a Kronecker product approximation to AdaGrad. Later, (Morwani et al., 2024) explicitly proved that it is a 1-step power interation to optimal Kronecker product approximation. In here, we propose an alternative view as minimizing a upper bound of the approximation error. SOAP (Vyas et al., 2024) is a recently proposed adaptive optimizer that further improves Shampoo based on the connection of Adafactor and Shampoo. In this work, we make **explicit** connection of those approaches to FIM approximation, and establish the equivalence of structural assumption to optimizers. We also additionally provide connections of gradient operators to FIM approximation, and design new optimizers from this view point. We provide more discussions of our approach to many existing optimizers, including Apollo (Zhu et al., 2024), GaLore (Zhao et al., 2024a), Muon (Jordan et al., 2024), SWAN (Ma et al., 2024), Adapprox (Zhao et al., 2024b), Lars (You et al., 2017), Lamb (You et al., 2019), Fira (Chen et al., 2024a) and AdaDiag (Nguyen et al., 2025), in Sec. I.5. Preconditioning SGD (Li, 2017; Pooladzandi & Li, 2024) aims to directly approximate the inverse Hessian through different structural assumptions.

**Comparison to blockwise low-rank approximation (Yen et al., 2023)**  Recently, (Yen et al., 2023) proposed an optimizer based on the blockwise low-rank approximation to the full AdaGrad preconditioner. On the high level, it seems to be similar to our proposed low-rank extension framework. In fact, they are different in several important ways. First, their approach is not scalable due to the use of the full coefficient matrix with shape $r \times r$, where $r$ is the rank. From Section 2.2 in (Yen et al., 2023), the memory cost of their method is $(m + n)r^2$. For 130M model with $r = 256$ and weight size $m = 768, n = 768$, this corresponds to 42 times higher memory than Adam and 84 times higher than the model weight. On the other hand, Alice only requires roughly $55\%$ of Adam's memory, and 77 times smaller than their approach. Our low-rank extension framework also incorporates switching and compensation steps, which are proved to be critical for low-rank based method. More importantly, the switching step challenges the common knowledge that one should use the top eigenvectors for a low-rank preconditioner. Our analysis (Theorem G.8) suggests that less important directions should be mixed with the leading ones for better performance.

**Memory-efficient optimizer**  Practical training efficiency is a crucial factor when training large models. In particular, there are many work focusing on the memory efficiency of optimizers, since less memory consumption allows larger batch size, effectively improving the throughput. There are two lines of research: (1) use low-rank approximation to reduce memory of optimizer internal states; (2) remove the internal states. GaLore (Zhao et al., 2024a), a well-know low-rank optimizer, proposed to use singular value decomposition (SVD) for finding a low-rank projection and apply Adam within it. It can be seen as a special case of Alice. Fira, an extension to GaLore, add compensation term at each step to turn low-rank update to full-rank, substantially improves the performance. Flora (Si et al., 2024) uses randomly sampled Gaussian matrix as the subspace to save compute and memory. However, it is mainly focused on the fine-tuning tasks. ReLora (Lialin et al., 2023), an extension to LoRA (Hu et al., 2021), periodically merges the LoRA weights to enable full-rank learning. On the other hand, many optimizers requires fewer internal states compared to Adam. Lion (Chen et al., 2024b) and Signum (Bernstein et al., 2018) introduced optimizers that only requires the storage of 1 internal states, offering a balance between memory efficiency and performance. Apollo (Zhu et al., 2024), a recently proposed approach, only maintains rank 1 GaLore states for estimating the scaling matrix for the raw gradient. Although it still requires 2 internal states, rank 1 states allows it to achieve SGD-like memory. At the same time, (Ma et al., 2024) developed SWAN, which manages to

completely removes the internal states through two gradient operators: normalization and whitening, and obtain stronger performance than Adam. In this paper, we also show that normalization and whitening operators are special cases of FIM approximation.

**Comparsion to Sophia** Sophia (Liu et al., 2023) is a recently proposed second-order optimizer for training LLMs. It requires similar memory consumption as Adam but claims to achieve $2x$ speed-up. However, (Kaddour et al., 2023) argues that the speed-up is achieved under an unfair comparison. With the fair comparison (i.e. the setup adopted in this paper), Sophia consistently performs worse/or similarly compared to Adam, whereas ours (i.e. Alice and RACS) achieve near $2x$ speed-up in both training steps and wall-clock time under the fair comparison setup.

## B  RACS AND ALICE ALGORITHMS

---

**Algorithm 3** RACS

1: **Input:** learning rate $\lambda$, $\beta$, scale $\alpha$, limiter threshold $\gamma$, optimization steps $T$.
2: $\boldsymbol{s}_0 = 0; \boldsymbol{q}_0 = 0; \phi_0 = 0$
3: **for** $t = 1, \ldots, T$ **do**
4:     $\boldsymbol{G}_t = \nabla_{\boldsymbol{W}_t}\mathcal{L}$
5:     Obtain $\boldsymbol{S}_t$ and $\boldsymbol{Q}_t$ by Eq. (11)
6:     $\boldsymbol{s}_t = \beta\boldsymbol{s}_{t-1} + (1 - \beta)\operatorname{Diag}(\boldsymbol{S}_t);$
7:     $\boldsymbol{q}_t = \beta\boldsymbol{q}_{t-1} + (1 - \beta)\operatorname{Diag}(\boldsymbol{Q}_t)$
8:     $\tilde{\boldsymbol{G}}_t = \operatorname{Diag_v}(\boldsymbol{q}_t)^{-\frac{1}{2}} \boldsymbol{G} \operatorname{Diag_v}(\boldsymbol{s}_t)^{-\frac{1}{2}}$
9:     $\eta = \gamma/\max\{\frac{\|\tilde{\boldsymbol{G}}_t\|}{\phi_{t-1}}, \gamma\}$ if $t > 1$ else 1
10:     $\phi_t = \eta\|\tilde{\boldsymbol{G}}_t\|$
11:     $\boldsymbol{W}_{t+1} = \boldsymbol{W}_t - \lambda\eta\alpha\tilde{\boldsymbol{G}}_t$
12: **end for**

---

**Algorithm 4** Alice/Alice-0 optimizer

1: **Input:** learning rate $\lambda$, scale $\alpha$, compensation scale $\alpha_c$, update interval $k$, $\beta_1$, $\beta_2$, $\beta_3$ ($\beta_3 = 0$ for Alice-0), optimization step $T$, rank $r$, loss function $\mathcal{L}$, limiter threshold $\gamma$, leading basis number $l$.
2: $\tilde{\boldsymbol{Q}}_0 = 0, \boldsymbol{U}_0 = 0, \boldsymbol{p}_0 = 0, \phi = 0, \boldsymbol{m}_0 = 0, \boldsymbol{v}_0 = 0$
3: **for** $t = 1 \ldots, T$ **do**
4:     $\boldsymbol{G}_t = \nabla_{\boldsymbol{W}_t}\mathcal{L}$
5:     **if** $t == 1$ or $(t \mod K) == 0$ **then**
6:         $\boldsymbol{Q}_t = \beta_3\boldsymbol{U}_t\tilde{\boldsymbol{Q}}_{t-1}\boldsymbol{U}_t^T + (1 - \beta_3)\boldsymbol{G}_t\boldsymbol{G}_t^T$
7:         $\boldsymbol{U}_t = Switch(\boldsymbol{Q}_t, r, l, \boldsymbol{U}_{t-1})$
8:     **else**
9:         $\boldsymbol{U}_t = \boldsymbol{U}_{t-1}$
10:     **end if**
11:     $\boldsymbol{\sigma}_t = \boldsymbol{U}_t^T\boldsymbol{G}_t$
12:     $\tilde{\boldsymbol{Q}}_t = \beta_3\tilde{\boldsymbol{Q}}_{t-1} + (1 - \beta_3)\boldsymbol{\sigma}_t\boldsymbol{\sigma}_t^T$
13:     $\boldsymbol{m}_t = \beta_1\boldsymbol{m}_{t-1} + (1 - \beta_1)\boldsymbol{\sigma}_t$
14:     $\boldsymbol{v}_t = \beta_2\boldsymbol{v}_{t-1} + (1 - \beta_2)\boldsymbol{\sigma}_t^2$
15:     $\boldsymbol{\omega} = \frac{\boldsymbol{m}_t}{\sqrt{\boldsymbol{v}_t}}$
16:     $\Delta_c, \boldsymbol{p}_t, \boldsymbol{\phi}_t = Compensation(\boldsymbol{U}_t, \boldsymbol{p}_{t-1}, \boldsymbol{\phi}_{t-1}, \gamma, \beta_1)$
17:     $\boldsymbol{W}_{t+1} = \boldsymbol{W}_t - \lambda\alpha(\boldsymbol{U}_t\boldsymbol{\omega} + \alpha_c\Delta_c)$
18: **end for**

---

## C  EXAMPLES OF DIAGONAL OPERATIONS

In this section, we will give some detailed examples on each of the diagonal operations we introduced in Sec. 2.

**Example of** $\mathrm{Diag}(\cdot)$   This will extract the diagonals of a input matrix into a vector:

$$\boldsymbol{M} = \begin{bmatrix} a_{11} & a_{12} & a_{13} \\ a_{21} & a_{22} & a_{23} \\ a_{31} & a_{32} & a_{33} \end{bmatrix}, \quad \mathrm{Diag}(\boldsymbol{M}) = [a_{11}, a_{22}, a_{33}]^T$$

**Example of** $\mathrm{Diag_B}(\cdot)$   This simply stack the input matrices sequence into a larger block diagonal matrix:

$$\mathrm{Diag_B}(\boldsymbol{M}_1, \boldsymbol{M}_2, \boldsymbol{M}_3) = \begin{bmatrix} \boldsymbol{M}_1 & \boldsymbol{0} & \boldsymbol{0} \\ \boldsymbol{0} & \boldsymbol{M}_2 & \boldsymbol{0} \\ \boldsymbol{0} & \boldsymbol{0} & \boldsymbol{M}_3 \end{bmatrix}$$

**Example of** $\mathrm{Diag_v}(\cdot)$   This will stack the vector element into a pure diagonal matrix:

$$\mathrm{Diag_v}([a_{11}, a_{22}, a_{33}]^T) = \begin{bmatrix} a_{11} & 0 & 0 \\ 0 & a_{22} & 0 \\ 0 & 0 & a_{33} \end{bmatrix}$$

**Example of** $\mathrm{Diag_M}(\cdot)$   This will stack the elements in input matrix to form a larger pure diagonal matrix in a column-wise manner.

$$\mathrm{Diag_M}\left(\begin{bmatrix} a_{11} & a_{12} \\ a_{21} & a_{22} \end{bmatrix}\right) = \begin{bmatrix} a_{11} & 0 & 0 & 0 \\ 0 & a_{21} & 0 & 0 \\ 0 & 0 & a_{12} & 0 \\ 0 & 0 & 0 & a_{22} \end{bmatrix}$$

# D   BACKGROUND

In this section, we will provide a more comprehensive backgrounds.

## D.1   FISHER INFORMATION

Fisher information can also be viewed as the "sharpness" of the likelihood around the true parameters from the maximum likelihood view point. Formally, under the context of LLMs $f_\theta(\cdot)$, we consider a dataset $\{\boldsymbol{x}_i\}_{i=1}^N$, where $N$ is total batched sentences, and $\boldsymbol{x}_i$ is the token sequence of $i^{\text{th}}$ sentence. One can define sentence-level auto-regressive loss function $\mathcal{L} = \sum_{j=1} c(x_{i,j+1}, f_\theta(\boldsymbol{x}_{i,:j})$, where $j$ is the token index and $c$ is the user-defined loss metric. The corresponding likelihood can be defined as $p_\theta(\boldsymbol{x}_i) \propto \exp(-\sum_{j=1} c(x_{i,j+1}, f_\theta(\boldsymbol{x}_{i,:j})))$. The standard empirical FIM is defined as

$$\boldsymbol{F} = \sum_{i=1}^N \nabla_\theta \log p_\theta(\boldsymbol{x}_i) \nabla_\theta^T \log p_\theta(\boldsymbol{x}_i)$$

In practice, we often use the mini-batched gradient $\vec{g} = \frac{1}{N_B} \sum_{i=1}^{N_B} \log p_\theta(\boldsymbol{x}_i)$ with batch size $N_B$ for empirical FIM during training. More discussion can be found in Lin et al. (2024). Throughout this paper, we adopt the notation $\mathbb{E}[\vec{g}\vec{g}^T]$ as $\boldsymbol{F}$.

One standard application of FIM is efficient optimization. NGD leverage the inverse of FIM to smooth the local information geometry, leading to the steepest descent in the probability space with KL divergence metric. This typically leads to faster convergences compared to its parameter-space counterpart; and more stable optimization (Martens, 2020). The update of NGD with step size $\lambda$ is

$$\theta \leftarrow \theta - \lambda \boldsymbol{F}^{-1} \nabla_\theta \mathcal{L}.$$

However, square-root inverse is sometimes more favorable than inverse, and has been shown that it provides a better approximation to the geodesic flow, compared with the default natural gradient update Yang & Laaksonen (2008). Empirically, it demonstrates stronger performance and desired properties when using non-constant learning rate (Lin et al., 2024; Loshchilov & Hutter, 2016; Bergstra & Bengio, 2012; Choi, 2019). The corresponding update is to simply replace $\boldsymbol{F}^{-1}$ with $\boldsymbol{F}^{-\frac{1}{2}}$:

$$\theta \leftarrow \theta - \lambda \boldsymbol{F}^{-\frac{1}{2}} \nabla_\theta \mathcal{L}. \tag{16}$$

In this paper, we will use the square-root inverse view point, but our analysis is agnostic to it and can be easily extended to the direct inverse. However, matrix multiplication involving FIM and its inverse are computationally expensive for large models since $\boldsymbol{F} \in \mathbb{R}^{mn \times mn}$ for vectorized parameters $\theta \in \mathbb{R}^{mn}$. Next, we will briefly introduce Kronecker product and block diagonals, which are two main classes of structural assumptions used in this paper. Their nice properties can significantly reduce the assocaited computation burden.

## D.2    KRONECKER PRODUCT AND BLOCK DIAGONALS

Kronecker product, denoted by $\otimes$, is to combine two arbitrary matrices into a larger matrix with block-wise patterns. It has emerged as a powerful tool for approximating higher-order information like Hessian (Grosse & Martens, 2016) or FIM. The main advantages are their nice properties regarding matrix operations, leading to efficient practical procedures when dealing with large matrix. We will mainly use two properties:

$$(\boldsymbol{A} \otimes \boldsymbol{B})^{-\frac{1}{2}} = \boldsymbol{A}^{-\frac{1}{2}} \otimes \boldsymbol{B}^{-\frac{1}{2}} \tag{17}$$

$$(\boldsymbol{A} \otimes \boldsymbol{B}) \operatorname{Vec}(\boldsymbol{C}) = \operatorname{Vec}(\boldsymbol{B}\boldsymbol{C}\boldsymbol{A}^{T}) \tag{18}$$

where the first one holds when $\boldsymbol{A}, \boldsymbol{B}$ are square-root invertible. The second one is particular useful to reduce the computation associated with matrix-vector multiplication in Eq. (1). For $\boldsymbol{A} \otimes \boldsymbol{B} \in \mathbb{R}^{mn \times mn}$, it reduces the computation from $O(m^2 n^2)$ to $O(mn^2 + m^2 n)$ with $\boldsymbol{A} \in \mathbb{R}^{n \times n}$, $\boldsymbol{B} \in \mathbb{R}^{m \times m}$, and $\boldsymbol{C} \in \mathbb{R}^{m \times n}$.

For block diagonal matrix, one can also easily compute its square-root inverse:

$$\operatorname{Diag}_{\mathrm{B}}(\boldsymbol{M}_1, \ldots, \boldsymbol{M}_n)^{-\frac{1}{2}} = \operatorname{Diag}_{\mathrm{B}}(\boldsymbol{M}_1^{-\frac{1}{2}}, \ldots, \boldsymbol{M}_n^{-\frac{1}{2}}) \tag{19}$$

where each $\boldsymbol{M}_i$ is square-root invertible. When each $\boldsymbol{M}_i$ is also a diagonal matrix with positive values, we have the following:

$$\operatorname{Diag}_{\mathrm{B}}(\boldsymbol{M}_1, \ldots, \boldsymbol{M}_n)^{-\frac{1}{2}} \operatorname{Vec}(\boldsymbol{C}) = \operatorname{Vec}\left(\frac{\boldsymbol{C}}{\sqrt{\operatorname{Mat}(\boldsymbol{v})}}\right) \tag{20}$$

where $\boldsymbol{v}$ is the vector containing the diagonals of $\operatorname{Diag}_{\mathrm{B}}(\boldsymbol{M}_1, \ldots, \boldsymbol{M}_n)$, transforming matrix vector product into element-wise operation.

## D.3    OPERATORS FOR GRADIENT: NORMALIZATION AND WHITENING

Recently, there are some optimizers (Ma et al., 2024; Jordan et al., 2024; You et al., 2017; 2019) that apply operators to pre-process the gradient and use it in standard SGD. Empirical evidence has verified their effectiveness in training LLMs. In particular, there are two well-known operators: normalization and whitening, where the above optimizers relies on one or both of them. In particular,

$$\operatorname{Norm}(\boldsymbol{G}) = \frac{\boldsymbol{G}}{1\sqrt{\boldsymbol{s}^{T}}} = \boldsymbol{G}\boldsymbol{S}^{-\frac{1}{2}} \tag{21}$$

$$\operatorname{Whitening}(\boldsymbol{G}) = (\boldsymbol{G}\boldsymbol{G}^{T})^{-\frac{1}{2}}\boldsymbol{G}. \tag{22}$$

where $\boldsymbol{s} \in \mathbb{R}^n$, $\boldsymbol{s}_i = \sum_{j=1}^{m} \boldsymbol{G}_{ij}^2$ with $\boldsymbol{G} \in \mathbb{R}^{m \times n}$, and $\boldsymbol{S} = \operatorname{Diag}_{\mathrm{v}}(\boldsymbol{s})$. Namely, vector $\boldsymbol{s}$ contains the squared column norm of $\boldsymbol{G}$, and $\boldsymbol{S}$ is a scaling matrix to normalize the columns. Normalizing the rows can also be written in a similar format. Whitening operator essentially orthogonalizes $\boldsymbol{G}$, that compute the closest orthogonal matrix to $\boldsymbol{G}$ under Frobenius norm. In practice, the whitening operator can be iteratively solved using Newton-Schulz algorithm without explicitly computing the square-root inverse.

## D.4    SHAMPOO OPTIMIZER

Shampoo (Gupta et al., 2018) was originally proposed as the second-order optimization technique over the tensor space. Under the context of transformers, typically matrix parameters are considered. Its core design principle also aims to approximate the FIM with structural assumptions $\tilde{\boldsymbol{F}}_t = \boldsymbol{R}_{n,t}^{\frac{1}{2}} \otimes \boldsymbol{L}_{m,t}^{\frac{1}{2}}$,

with $\boldsymbol{R}_{n,t} = \boldsymbol{R}_{n,t-1} + \boldsymbol{G}_t^T \boldsymbol{G}_t$ and $\boldsymbol{L}_{m,t} = \boldsymbol{L}_{m,t-1} + \boldsymbol{G}_t \boldsymbol{G}_t^T$. The above update rule can be seen as the moving average estimate of $\mathbb{E}[\boldsymbol{G}_t^T \boldsymbol{G}_t]$ and $\mathbb{E}[\boldsymbol{G}_t \boldsymbol{G}_t^T]$. However, the Shampoo paper (Gupta et al., 2018) does not explicitly show why these design choices of $\boldsymbol{R}_n, \boldsymbol{L}_m$ approximate the FIM. In fact, they only show $\boldsymbol{R}_n^{\frac{1}{2}} \otimes \boldsymbol{L}_m^{\frac{1}{2}}$ forms an upper bound of FIM[2]. This is not helpful in understanding whether this approximates the FIM or not. Another very recent follow-up work (Morwani et al., 2024) provides an explanation of the shampoo preconditioner in terms of approximating FIM. They show the square of Shampoo's preconditioner is equivalent to a single step of power iteration of computing optimal Kronecker product approximation to FIM. It indirectly establishes the connections to FIM approximation since the approximation is expressed as an iterative algorithm. To the best of our knowledge, our result is the first to directly establish the connection of Shampoo to FIM approximation as minimizing a upper bound of the loss with the Frobenius norm (Theorem 3.2).

Nevertheless, the original Shampoo algorithm is summarized in Algorithm 5.

---

**Algorithm 5** Shampoo Optimizer

**Input:** $\boldsymbol{L}_m = \epsilon \boldsymbol{I}_m$, $\boldsymbol{R}_n = \epsilon \boldsymbol{I}_n$, learning rate $\lambda$, optimization step $T$, loss function $\mathcal{L}$.
**for** $t = 1, \ldots, T$ **do**
    $\boldsymbol{G}_t = \nabla_{\boldsymbol{W}_t} \mathcal{L}$
    $\boldsymbol{L}_{m,t} = \boldsymbol{L}_{m,t-1} + \boldsymbol{G}_t \boldsymbol{G}_t^T$
    $\boldsymbol{R}_{n,t} = \boldsymbol{R}_{n,t-1} + \boldsymbol{G}_t^T \boldsymbol{G}_t$
    $\boldsymbol{W}_t = \boldsymbol{W}_{t-1} + \lambda \boldsymbol{L}_{m,t}^{-\frac{1}{4}} \boldsymbol{G}_t \boldsymbol{R}_{n,t}^{-\frac{1}{4}}$
**end for**

---

### D.5 SOAP/ADADIAG++

SOAP/AdaDiag++ (Vyas et al., 2024; Nguyen et al., 2025) is a recently proposed adaptive optimizer aiming to improve the practical convergence and stability of Shampoo. Their main intuition behind (from the view point of Vyas et al. (2024)) is that they show Shampoo is equivalent to performing Adafactor (Shazeer & Stern, 2018) under the Shampoo's eigen-space. Namely, the eigen-matrix of $\boldsymbol{L}_{m,t}$ and $\boldsymbol{R}_{n,t}$ in Algorithm 5. Since Adafactor is an approximation to Adam, they propose to use Adam instead of Adafactor in Shampoo's eigen-space, to further improve the performance. They propose the following update rule:

$$\boldsymbol{m}_t = \beta_1 \boldsymbol{m}_{t-1} + (1 - \beta_1) \boldsymbol{G}_t \quad \text{(first moment)}$$
$$\boldsymbol{L}_{m,t} = \beta_3 \boldsymbol{L}_{m,t-1} + (1 - \beta_3) \boldsymbol{G}_t \boldsymbol{G}_t^T \quad \text{(Shampoo's } \boldsymbol{L}_{m,t})$$
$$\boldsymbol{R}_{n,t} = \beta_3 \boldsymbol{R}_{n,t-1} + (1 - \beta_3) \boldsymbol{G}_t^T \boldsymbol{G}_t \quad \text{(Shampoo's } \boldsymbol{R}_{n,t})$$
$$\boldsymbol{U}_{L,t} = \text{EVD}(\boldsymbol{L}_{m,t}) \quad \text{(Shampoo's left eigen-space)}$$
$$\boldsymbol{U}_{R,t} = \text{EVD}(\boldsymbol{R}_{n,t}) \quad \text{(Shampoo's right eigen-space)}$$
$$\boldsymbol{v}_t = \beta_2 \boldsymbol{v}_{t-1} + (1 - \beta_2)(\boldsymbol{U}_{L,t}^T \boldsymbol{G}_t \boldsymbol{U}_{R,t})^2 \quad \text{(second moment)}$$
$$\Delta = \boldsymbol{U}_{L,t} \frac{\boldsymbol{U}_{L,t}^T \boldsymbol{m}_t \boldsymbol{U}_{R,t}}{\sqrt{\boldsymbol{v}_t}} \boldsymbol{U}_{R,t}^T \tag{23}$$

These update rules exactly describes the procedure to applying Adam updates in the "rotated" space defined by $\boldsymbol{U}_{L,t}$ and $\boldsymbol{U}_{R,t}$. Due to the computational burden associated with EVD, SOAP proposed to only update $\boldsymbol{U}_{L,t}, \boldsymbol{U}_{R,t}$ at certain intervals. This leads to the following algorithm:

### D.6 ADADIAG AND ONE-SIDE SOAP

AdaDiag++ (Nguyen et al., 2025), a concurrent work to SOAP, independently develops the equivalent update rules as SOAP. The only difference is that they disable the EMA tracking for $\boldsymbol{L}_{m,t}$ and $\boldsymbol{R}_{n,t}$. The resulting optimizer is both computational and memory expensive due to the storage of $\boldsymbol{U}_R, \boldsymbol{U}_L$ and two eigenvalue decompositions. To address this issue, they both propose a one-side version

---

[2]Lemma 8 in (Gupta et al., 2018). Note that our paper assumes $\text{Vec}(\cdot)$ is stacking columns of matrix whereas Gupta et al. (2018) assumes stacking the rows, explaining the reverse order of presentation

---

**Algorithm 6** SOAP optimizer

---

**Input:** learning rate $\lambda$, update interval $K$, $\beta_1$, $\beta_2$, $\beta_3$, optimization step $T$.
$\boldsymbol{m}_0 = 0; \boldsymbol{v}_0 = 0$                 ▷*Initialize two moments*
$\boldsymbol{L}_{m,0} = 0; \boldsymbol{R}_{n,0} = 0$         ▷*Initialize two EMA states for* $\boldsymbol{GG}^T$*,* $\boldsymbol{G}^T\boldsymbol{G}$
**for** $t = 1, \ldots, T$ **do**
 $\boldsymbol{G}_t = \nabla_{\boldsymbol{W}_t}\mathcal{L}$
 $\boldsymbol{m}_t = \beta_1 \boldsymbol{m}_{t-1} + (1 - \beta_1)\boldsymbol{G}_t$
 $\boldsymbol{L}_{m,t} = \beta_3 \boldsymbol{L}_{m,t-1} + (1 - \beta_3)\boldsymbol{G}_t \boldsymbol{G}_t^T$        ▷*Accumulation for* $\boldsymbol{GG}^T$
 $\boldsymbol{R}_{n,t} = \beta_3 \boldsymbol{R}_{n,t-1} + (1 - \beta_3)\boldsymbol{G}_t^T \boldsymbol{G}_t$        ▷*Accumulation for* $\boldsymbol{G}^T\boldsymbol{G}$
 **if** $t == 1$ or $(t \mod K) == 0$ **then**
  $\boldsymbol{U}_{R,t} = \text{EVD}(\boldsymbol{R}_{n,t})$           ▷*Get right eigen-space* $\boldsymbol{U}_R$
  $\boldsymbol{U}_{L,t} = \text{EVD}(\boldsymbol{L}_{m,t})$           ▷*Get left eigen-space* $\boldsymbol{U}_L$
 **else**
  $\boldsymbol{U}_{R,t} = \boldsymbol{U}_{R,t-1}$
  $\boldsymbol{U}_{L,t} = \boldsymbol{U}_{L,t-1}$
 **end if**
 $\widetilde{\boldsymbol{m}}_t = \boldsymbol{U}_{L,t}^T \boldsymbol{m}_t \boldsymbol{U}_{R,t}$           ▷*Get rotated 1st moment*
 $\boldsymbol{v}_t = \beta_2 \boldsymbol{v}_{t-1} + (1 - \beta_2)(\boldsymbol{U}_{L,t}^T \boldsymbol{G}_t \boldsymbol{U}_{R,t})^2$     ▷*Compute second moments*
 $\boldsymbol{W}_{t+1} = \boldsymbol{W}_t - \lambda \boldsymbol{U}_{L,t}\frac{\widetilde{\boldsymbol{m}}_t}{\sqrt{\boldsymbol{v}_t}}\boldsymbol{U}_{R,t}^T$
**end for**

---

called AdaDiag and one-side SOAP by only considering either left or right eigen-space. The resulting update rule is exactly the same as our proposed OSOAP (i.e. Eq. (9)).

However, they propose this design choice purely based on intuition to reduce computation and memory consumption, and do not explicitly reveal the connections to their two-side version. Thus, it lacks the understanding on why two-side version obtains empirically better performance. Based on Sec. 3.4 and Sec. E, we show that although one-side version has similar updates as the two-side twins, they are different optimizers with distinct underlying structural assumptions. In fact, the structures of SOAP/AdaDiag++ strictly generalizes their one-side version, explaining the better empirical performance. The resulting algorithm is the following:

---

**Algorithm 7** OSOAP/AdaDiag/one-side SOAP optimizer

---

**Input:** learning rate $\lambda$, update interval $K$, $\beta_1$, $\beta_2$, $\beta_3$, optimization step $T$.
$\boldsymbol{m}_0 = 0; \boldsymbol{v}_0 = 0$                 ▷*Initialize two moments*
$\boldsymbol{Q}_0 = 0$              ▷*Initialize the EMA state for* $\boldsymbol{GG}^T$
**for** $t = 1, \ldots, T$ **do**
 $\boldsymbol{G}_t = \nabla_{\boldsymbol{W}_t}\mathcal{L}$
 $\boldsymbol{Q}_t = \beta_3 \boldsymbol{Q}_{t-1} + (1 - \beta_3)\boldsymbol{G}_t \boldsymbol{G}_t^T$         ▷*Accumulate the* $\boldsymbol{GG}^T$
 $\boldsymbol{m}_t = \beta_1 \boldsymbol{m}_{t-1} + (1 - \beta_1)\boldsymbol{G}_t$        ▷*Accumulate the first moment*
 **if** $t == 1$ or $(t \mod K) == 0$ **then**
  $\boldsymbol{U}_{f,t} = \text{EVD}(\boldsymbol{Q}_t)$            ▷*Obtain the* $\boldsymbol{U}$
 **else**
  $\boldsymbol{U}_{f,t} = \boldsymbol{U}_{f,t-1}$
 **end if**
 $\tilde{\boldsymbol{m}}_t = \boldsymbol{U}_{f,t}^T \boldsymbol{m}_t$            ▷*Rotate the first moment*
 $\boldsymbol{v}_t = \beta_2 \boldsymbol{v}_{t-1} + (1 - \beta_2)(\boldsymbol{U}_{f,t}^T \boldsymbol{G}_t)^2$     ▷*Accumulate the second moments*
 $\boldsymbol{W}_{t+1} = \boldsymbol{W}_t - \lambda \boldsymbol{U}_{f,t}\frac{\tilde{\boldsymbol{m}}_t}{\sqrt{\boldsymbol{v}_t}}$       ▷*Update in the original space*
**end for**

---

## D.7 SWAN

Recently, there is a newly proposed adaptive optimizer that completely removes the needs of storing internal states, called SWAN. It relies on two processing operators applied to raw current gradient: GradNorm and GradWhitening, as a replacement of first and second moments. For a current gradient

$G$,

$$\text{GradNorm}(\boldsymbol{G}) = \frac{\boldsymbol{G} - \bar{\bar{\boldsymbol{g}}}\boldsymbol{1}_n^\top}{\boldsymbol{s}\boldsymbol{1}_n^\top} \tag{24}$$

$$\text{GradWhitening}(\boldsymbol{G}) = (\boldsymbol{G}\boldsymbol{G}^T)^{-\frac{1}{2}}\boldsymbol{G} \tag{25}$$

where $\bar{\bar{\boldsymbol{g}}} = \frac{1}{n}\sum_{i=1}^n \boldsymbol{G}_{:,i}$ is the mean across rows; $\boldsymbol{s} = \sqrt{\frac{1}{n}\sum_{i=1}^n (\boldsymbol{G}_{:,i} - \bar{\bar{\boldsymbol{g}}})^2}$ is the standard deviation across rows; $\bar{\bar{\boldsymbol{g}}}$ and $\boldsymbol{s}$ are $m$-dimensional column vectors; and $\boldsymbol{1}_n$ is a $n$-dimensional column vector of ones. Then, SWAN performs the following to generate the update:

$$\tilde{\boldsymbol{G}} = \text{GradNorm}(\boldsymbol{G})$$

$$\Delta = \text{GradWhitening}(\tilde{\boldsymbol{G}}) \tag{26}$$

We can see that the proposed $\text{GradNorm}$ is equivalent to normalization up to a scaling $\sqrt{n}$ when the mean $\bar{\bar{\boldsymbol{g}}}$ is $0$. SWAN derives these two steps from investigating the LLM dynamics. In practice, SWAN proposes to compute the $(\boldsymbol{G}\boldsymbol{G}^T)^{-\frac{1}{2}}$ using Newton-Schulz iterations.

### D.8 NEWTON SCHULZ ITERATION

In many machine learning applications, like successive whitening and coloring transform (Li et al., 2017; Cho et al., 2019; Choi et al., 2021), one often encountered the computation of square root inverse of some SPD matrix. One standard approach is to compute the EVD and take the square root inverse of the eigenvalue matrix. However, EVD is computationally expensive. Another alternative approach is use Newton-Schulz iteration (NS), an iterative updates of two matrix. Specifically,

$$\boldsymbol{Y}_0 = \frac{\boldsymbol{A}}{\|\boldsymbol{A}\|_F} \quad \boldsymbol{Z}_0 = \boldsymbol{I}$$

$$\boldsymbol{Y}_{t+1} = \frac{1}{2}\boldsymbol{Y}_t(3\boldsymbol{I} - \boldsymbol{Z}_t\boldsymbol{Y}_t)$$

$$\boldsymbol{Z}_{t+1} = \frac{1}{2}(3\boldsymbol{I} - \boldsymbol{Z}_t\boldsymbol{Y}_t)\boldsymbol{Z}_t$$

with convergence $\boldsymbol{Y}_t \to \frac{\boldsymbol{A}^{\frac{1}{2}}}{\sqrt{\|\boldsymbol{A}\|_F}}$ and $\boldsymbol{Z}_t \to \boldsymbol{A}^{-\frac{1}{2}}\sqrt{\|\boldsymbol{A}\|_F}$. Typically, NS converges very fast with only 5 steps (Li et al., 2018; Huang et al., 2019).

### D.9 MUON

Muon (Jordan et al., 2024), is recently proposed to speed-up the training of LLMs, that relies on the whitening operator similar to SWAN. The core of the Muon is to orthogonalize the the momentum. The proposed update rule is

$$\boldsymbol{m}_t = \beta_1\boldsymbol{m}_{t-1} + (1 - \beta_1)\boldsymbol{G}_t$$
$$\Delta = \text{GradWhitening}(\boldsymbol{m}_t). \tag{27}$$

Similarly, the $\text{GradWhitening}$ step is computed using Newton-Schulz iteration. The main difference between Muon and SWAN is that Muon still requires the storage of first moments as state, whereas SWAN relies on the $\text{GradNorm}$ operator applied to the raw gradient.

### D.10 LARS

SWAN and Muon both involve the whitening operator with/without normalization, respectively. On the other hand, Lars (You et al., 2017) is an operator that only relies on layer-wise normalization. For each layer, it simply compute the first moments, followed by a normalization operation. The update rule for each layer is

$$\boldsymbol{m}_t = \beta_1\boldsymbol{m}_{t-1} + (1 - \beta_1)\boldsymbol{G}_t$$
$$\Delta = \phi(\|\theta\|)\frac{\boldsymbol{m}_t}{\|\boldsymbol{m}_t\|} \tag{28}$$

where $\phi$ is a scaling function with input of the parameter $\theta$ norm. One major difference of this layer-wise normalization to SWAN is that it is applied on the layer-wise level, whereas SWAN applies row/column-wise normalization.

## D.11 LOW RANK OPTIMIZERS

The primary goal of low-rank optimizer is to reduce the memory consumed by the states of adaptive optimizers. The popularity of low-rank based method for large models starts from the well-known LoRA (Hu et al., 2021), where each weight is inserted with an low-rank adapter $W + AB$ with $A \in \mathbb{R}^{m \times r}$ and $B \in \mathbb{R}^{r \times n}$ during the finetuning stage. This formulation directly modifies the model architecture. Si et al. (2024) explicitly show that LoRA is secretly a gradient compressor, which translates the modification to model architecture into a low-rank optimizer with randomly sampled matrix. At the same time, GaLore (Zhao et al., 2024a) popularizes the use of low-rank optimizers, which demonstrates on-par performance compared to full-rank Adam training. GaLore is proposed based on the analysis of reversible networks (Tian et al., 2020). However, in practice, transformer may not satisfy the reversibility condition. Thus, GaLore does not provide a clear understanding on why it works on LLMs.

Algorithm 8 summarizes the procedures. In practice, GaLore can be viewed as the composition of subspace search algorithm (i.e. SVD) with standard Adam optimizer. The original GaLore does not provide an explanation on the choice of Adam. On the other hand, our analysis reveals that the GaLore is an approximate low-rank extension to a different optimizer, OSOAP/AdaDiag/one-side SOAP, that is generalizes Adam (see Sec. 3.4).

---

**Algorithm 8** GaLore Optimizer

**Input: learning rate $\lambda$, decay rates $\beta_1, \beta_2$, rank $r$, update interval $k$, scale $\alpha$**
**for** $t = 1, \ldots, T$ **do**
    $G_t = \nabla_{W_t} \mathcal{L}$
    **if** $t == 1$ **or** $t \mod k == 0$ **then**
        $U_t = \text{SVD}(G_t, r)$
    **else**
        $U_t = U_{t-1}$
    **end if**
    $\sigma_t = U_t^T G_t$
    $\Delta = Adam(\sigma_t, \beta_1, \beta_2)$
    $W_t = W_{t-1} + \lambda \alpha \Delta$
**end for**

---

Since the states of Adam optimizer are based on the projected gradient $\sigma_t$, the overall memory consumption of GaLore is $mn + 2nr + mr$.

## D.12 APOLLO

Concurrent to this work, there is a recently proposed optimizer, called Apollo (Zhu et al., 2024), that only scales the raw gradient and obtains SGD-like memory with Adam-level performance. They propose to scale the columns or rows similar to normalization in SWAN, but the scaling factor is estimated following the procedure proposed by Fira (Chen et al., 2024a). The core idea of Apollo is to obtain $\Delta$ from GaLore algorithm (Algorithm 8), followed by computing the column/row-wise norm of $\Delta$. This norm will be used as the scaling factor for the raw gradient $G$. Apollo has many variants. In particular, we choose Apollo-mini and Apollo-svd, where the former uses rank-1 random projection for scaling estimation, and the latter relies on the use of top $r$ singular vectors as the projection, same as GaLore. Apollo-mini only maintains the rank-1 states, leading to significant memory savings. The memory consumption is $mn + 2n + 2$ for parameter $W \in \mathbb{R}^{m \times n}$. And Apollo-svd consumes the same memory as GaLore. Algorithm 9 summarizes the procedures.

Note that when rank $r$ is set to 1, they propose to use global scaling $\frac{\|\Delta_t\|_2}{\|\sigma_t\|_2}$ instead of row/column-wise scaling.

## D.13 SUBSPACE ITERATION

The subspace iteration method—also known as the block power method is a classical iterative technique for computing the dominant eigenvalues and corresponding eigenvectors of a matrix. It generalizes the basic power method from operating on a single vector to operating on a subspace,

---

**Algorithm 9** Apollo Optimizer

---

**Input: learning rate $\lambda$, decay rates $\beta_1$, $\beta_2$, rank $r$, update interval $k$, scale $\alpha$**
**for** $t = 1, \ldots, T$ **do**
    $G_t = \nabla_{W_t} \mathcal{L}$
    **if** $t == 1$ **or** $t \mod k == 0$ **then**
        $U_t \sim \mathcal{N}(0, \frac{1}{r})$
        **seed $\leftarrow$ an independent new random seed**
    **else**
        $U_t = U_{t-1}$
    **end if**
    $\sigma_t = U_t^T G_t$
    $\Delta_t = Adam(\sigma_t, \beta_1, \beta_2)$
    $S_t \leftarrow \mathrm{Diag}_v(s_0, \ldots, s_m) \left\{ s_i = \frac{\|\Delta_{t,:,i}\|_2}{\|\sigma_{t,:,i}\|_2} \right\}$
    $W_t = W_{t-1} + \lambda \alpha G_t S_t$
**end for**

---

typically spanned by a initial matrix. When the initial matrix is closed to the targeting eigen-matrix, the convergence is fast. Empirically, we found that only 1 step of iteration is enough to give a satisfactory performance. Algorithm 10 summarizes the subspace iteration algorithm to for finding the top $r$ eigenvectors of a matrix $A$. We can see that the computation is bottlenecked by the matrix multiplication $AU_{t-1}$ which is $O(m^2 r)$ if only performing 1 step.

---

**Algorithm 10** Subspace iteration

---

**Input: symmetric matrix $A \in \mathbb{R}^{m \times m}$, iteration step $T$, initial matrix $M \in \mathbb{R}^{m \times r}$**
$U_0 = M$
**for** $t = 1, \ldots, T$ **do**
    $H_t = AU_{t-1}$
    $U_t = \textbf{QR decomposition}(H_t)$
**end for**
$V = U_T^T A U_T$
$U = \mathrm{EVD}(V)$

---

# E   SOAP: A FURTHER GENERALIZATION TO OSOAP

All previous structures, apart from the one behind Shampoo, are under the class of block diagonal. However, this only allows one to model the off-diagonal elements near the diagonal. Structure under Kronecker product, like the one behind Shampoo, can go beyond this. Therefore, we can consider combining the structure of OSOAP with Shampoo, to obtain the most general structural assumption in this paper. We show this exactly recovers SOAP (Vyas et al., 2024).

Specifically, we consider the following structural assumption: $\mathcal{H} = \{(U_R \otimes U_L)\tilde{D}(U_R \otimes U_L)^T\}$, where $U_R \in \mathbb{R}^{n \times n}$, $U_L \in \mathbb{R}^{m \times m}$ are orthonormal matrix, and $\tilde{D} \in \mathbb{R}^{mn \otimes mn}$ is a diagonal matrix with positive values. We can easily show that structure behind OSOAP is a special case by constraining $U_R = I_n$; and Shampoo is also a special case by constraining $\tilde{D}$ to be decomposed by Kronecker product (refer to Sec. I.1).

Similar to OSOAP, it is hard to directly minimizing Eq. (2) with this assumption. We can approximate the solution by 1-step refinement procedure as OSOAP.

**Theorem E.1** (1-step refinement of SOAP). *Assuming the above structural assumptions. Consider the following 1-step refinement: (1) assume $\tilde{D}$ can be decomposed as Kronecker product of two diagonal matrix, find corresponding $U_R$, $U_L$; (2) fix $U_R$, $U_L$, find corresponding $\tilde{D}$. Step (1) admits analytic for minimizing the upper bound of Eq. (2) (i.e. Eq. (2)):*

$$U_R = \mathrm{EVD}(\mathbb{E}[G^T G]), \quad U_L = \mathrm{EVD}(\mathbb{E}[GG^T]).$$

*Step (2) admits an analytic solution for minimizing Eq.* (2):

$$\tilde{\boldsymbol{D}} = \mathrm{Diag}_{\mathrm{M}}(\mathbb{E}[(\boldsymbol{U}_L^T \boldsymbol{G} \boldsymbol{U}_R)^2])$$

.

The proof is a straightforward combination of Theorem 3.2 and Theorem 3.4, and can be found in Sec. G.7. One can show that the corresponding optimizer associated with the above result is equivalent to SOAP (refer to Sec. F.4 for details).

## F  DERIVATION OF UPDATE FORMULA

In this section, we will explicitly show how to connect the solution from minimizing reconstruction loss of FIM (Eq. (2)) to corresponding update rule.

### F.1  SHAMPOO'S UPDATE FORMULA

The key update formula of Shampoo is

$$\boldsymbol{W}_t = \boldsymbol{W}_{t-1} + \lambda \boldsymbol{L}_{m,t}^{-\frac{1}{4}} \boldsymbol{G}_t \boldsymbol{R}_{n,t}^{-\frac{1}{4}}$$

*Proof.* From Theorem 3.2, we simply apply the properties of Kronecker product to square-root version of natural gradient descent:

$$\begin{aligned}
&\mathrm{Mat}\left(\tilde{\boldsymbol{F}}^{-\frac{1}{2}}\vec{\boldsymbol{g}}\right) \\
&= \mathrm{Mat}\left((\boldsymbol{R}_n^{\frac{1}{2}} \otimes \boldsymbol{L}_m^{\frac{1}{2}})^{-\frac{1}{2}}\vec{\boldsymbol{g}}\right) \\
&= \mathrm{Mat}\left(\mathrm{Vec}\left(\boldsymbol{L}_m^{-\frac{1}{4}}\boldsymbol{G}\boldsymbol{R}_n^{-\frac{1}{4}}\right)\right) \\
&= \boldsymbol{L}_m^{-\frac{1}{4}}\boldsymbol{G}\boldsymbol{R}_n^{-\frac{1}{4}}
\end{aligned}$$

$\square$

### F.2  GENERALIZATION TO WHITENING AND NORMALIZATION

The square-root NGD update with $\tilde{\boldsymbol{F}}$ in Eq. (3) in Theorem 3.3 is

$$\mathrm{Mat}\left(\tilde{\boldsymbol{F}}^{-\frac{1}{2}}\vec{\boldsymbol{g}}\right) = \sqrt{n}\mathbb{E}[\boldsymbol{G}\boldsymbol{G}^T]^{-\frac{1}{2}}\boldsymbol{G} \tag{29}$$

*Proof.* From the solution in Eq. (3), we can simply apply the properties of Kronecker product as in the derivation of Shampoo's update:

$$\begin{aligned}
&\mathrm{Mat}\left(\tilde{\boldsymbol{F}}^{-\frac{1}{2}}\vec{\boldsymbol{g}}\right) \\
&= \mathrm{Mat}\left((\boldsymbol{I}_n \otimes \boldsymbol{M})^{-\frac{1}{2}}\vec{\boldsymbol{g}}\right) \\
&= \mathrm{Mat}\left(\mathrm{Vec}\left(\sqrt{n}\boldsymbol{M}^{-\frac{1}{2}}\boldsymbol{G}\right)\right) \\
&= \sqrt{n}\mathbb{E}[\boldsymbol{G}\boldsymbol{G}^T]^{-\frac{1}{2}}\boldsymbol{G}
\end{aligned}$$

$\square$

Similarly, the square-root NGD update with $\tilde{\boldsymbol{F}} = \boldsymbol{S} \otimes \boldsymbol{I}_m$ is

$$\mathrm{Mat}\left(\tilde{\boldsymbol{F}}^{-\frac{1}{2}}\vec{\boldsymbol{g}}\right) = \sqrt{m}\boldsymbol{G}\boldsymbol{S}^{-\frac{1}{2}} \tag{30}$$

*Proof.* This is trivial by applying the property of Kronecker product:

$$\text{Mat}((\boldsymbol{S} \otimes \boldsymbol{I}_m)^{-\frac{1}{2}}\vec{\boldsymbol{g}})$$
$$= \text{Mat}(\text{Vec}(\boldsymbol{G}\boldsymbol{S}^{-\frac{1}{2}}))$$
$$= \boldsymbol{G}\boldsymbol{S}^{-\frac{1}{2}}$$

$\square$

### F.3 UPDATE FORMULA FOR OSOAP

*Proof.* From the Theorem 3.4, we can apply the properties of block diagonal and Kronecker product with a full-rank $\boldsymbol{U}$:

$$\text{Mat}(\tilde{\boldsymbol{F}}^{-\frac{1}{2}}\vec{\boldsymbol{g}})$$
$$= \text{Mat}\left(\text{Diag}_{\text{B}}\left(\boldsymbol{M}_1^{-\frac{1}{2}}, \ldots, \boldsymbol{M}_n^{-\frac{1}{2}}\right)\vec{\boldsymbol{g}}\right)$$
$$= \text{Mat}\left(\text{Diag}_{\text{B}}\left(\boldsymbol{U}\boldsymbol{D}_1^{-\frac{1}{2}}\boldsymbol{U}^T, \ldots, \boldsymbol{U}\boldsymbol{D}_n^{-\frac{1}{2}}\boldsymbol{U}^T\right)\vec{\boldsymbol{g}}\right)$$
$$= \text{Mat}\left((\boldsymbol{I}_n \otimes \boldsymbol{U})\text{Diag}_{\text{B}}(\sqrt{\boldsymbol{D}_1}, \ldots, \sqrt{\boldsymbol{D}_n})(\boldsymbol{I} \otimes \boldsymbol{U}^T)\vec{\boldsymbol{g}}\right)$$
$$= \text{Mat}\left((\boldsymbol{I}_n \otimes \boldsymbol{U})\text{Diag}_{\text{B}}(\sqrt{\boldsymbol{D}_1}, \ldots, \sqrt{\boldsymbol{D}_n})\text{Vec}\left(\boldsymbol{U}^T\boldsymbol{G}\right)\right)$$
$$= \text{Mat}\left((\boldsymbol{I}_n \otimes \boldsymbol{U})\text{Vec}\left(\frac{\boldsymbol{U}^T\boldsymbol{G}}{\sqrt{\mathbb{E}[(\boldsymbol{U}^T\boldsymbol{G})^2]}}\right)\right)$$
$$= \text{Mat}\left(\text{Vec}\left(\boldsymbol{U}\frac{\boldsymbol{U}^T\boldsymbol{G}}{\sqrt{\mathbb{E}[(\boldsymbol{U}^T\boldsymbol{G})^2]}}\right)\right)$$
$$= \boldsymbol{U}\frac{\boldsymbol{U}^T\boldsymbol{G}}{\sqrt{\mathbb{E}[(\boldsymbol{U}^T\boldsymbol{G})^2]}}$$

$\square$

### F.4 UPDATE FORMULA FOR SOAP

Based on the Theorem E.1, we can derive the update formula of the corresponding square-root NGD following the same procedure as OSOAP:

$$\text{Mat}\left(\tilde{\boldsymbol{F}}^{-\frac{1}{2}}\vec{\boldsymbol{g}}\right) = \boldsymbol{U}_L\frac{\boldsymbol{U}_L^T\boldsymbol{G}\boldsymbol{U}_R}{\sqrt{\mathbb{E}[(\boldsymbol{U}_L^T\boldsymbol{G}\boldsymbol{U}_R)^2]}}\boldsymbol{U}_R^T.$$

*Proof.*

$$\text{Mat}\left(\tilde{\boldsymbol{F}}^{-\frac{1}{2}}\vec{\boldsymbol{g}}\right)$$
$$= \text{Mat}\left((\boldsymbol{U}_R \otimes \boldsymbol{U}_L)\text{Diag}_{\text{M}}(\mathbb{E}[(\boldsymbol{U}_L^T\boldsymbol{G}\boldsymbol{U}_R)^2])^{-\frac{1}{2}}(\boldsymbol{U}_R \otimes \boldsymbol{U}_L)^T\vec{\boldsymbol{g}}\right)$$
$$= \text{Mat}\left((\boldsymbol{U}_R \otimes \boldsymbol{U}_L)\text{Diag}_{\text{M}}(\mathbb{E}[(\boldsymbol{U}_L^T\boldsymbol{G}\boldsymbol{U}_R)^2])^{-\frac{1}{2}}\text{Vec}\left(\boldsymbol{U}_L^T\boldsymbol{G}\boldsymbol{U}_R\right)\right)$$
$$= \text{Mat}\left((\boldsymbol{U}_R \otimes \boldsymbol{U}_L)\text{Vec}\left(\frac{\boldsymbol{U}_L^T\boldsymbol{G}\boldsymbol{U}_R}{\sqrt{\mathbb{E}[(\boldsymbol{U}_L^T\boldsymbol{G}\boldsymbol{U}_R)^2]}}\right)\right)$$
$$= \text{Mat}\left(\text{Vec}\left(\boldsymbol{U}_L\frac{\boldsymbol{U}_L^T\boldsymbol{G}\boldsymbol{U}_R}{\sqrt{\mathbb{E}[(\boldsymbol{U}_L^T\boldsymbol{G}\boldsymbol{U}_R)^2]}}\boldsymbol{U}_R^T\right)\right)$$
$$= \boldsymbol{U}_L\frac{\boldsymbol{U}_L^T\boldsymbol{G}\boldsymbol{U}_R}{\sqrt{\mathbb{E}[(\boldsymbol{U}_L^T\boldsymbol{G}\boldsymbol{U}_R)^2]}}\boldsymbol{U}_R^T$$

$\square$

Therefore, one can design the optimizer based on this update formula and exactly recovers the SOAP's procedure (Eq. (23) in Sec. D.5).

# G  THEORY AND PROOF

To prove the results, we need to first introduce some useful lemmas and inequalities.

**Lemma G.1.** *Assume $\tilde{\boldsymbol{F}}$ is a block diagonal matrix with $n$ squared block matrix $\boldsymbol{M}_i \in \mathbb{R}^{m \times m}$, then*

$$\min_{\tilde{\boldsymbol{F}}} \|\tilde{\boldsymbol{F}} - \boldsymbol{F}\|_F^2 = \min_{\{\boldsymbol{M}_i\}_{i=1}^n} \sum_{i=1}^n \|\boldsymbol{M}_i\|_F^2 - 2 \operatorname{Tr}(\boldsymbol{M}_i^T \mathbb{E}[\boldsymbol{g}_i \boldsymbol{g}_i^T]) + C \tag{31}$$

*where $\boldsymbol{g}_i$ is the $i^{th}$ column of gradient $\boldsymbol{G}$, $C$ is a constant that is idenpendent of $\tilde{\boldsymbol{F}}$, and $\boldsymbol{F}$ is the FIM.*

*Proof.* This is straightforward by expanding the Frobenius norm (F-norm).

$$\begin{aligned}
&\|\tilde{\boldsymbol{F}} - \boldsymbol{F}\|_F^2 \\
&= \operatorname{Tr}\left((\tilde{\boldsymbol{F}} - \boldsymbol{F})^T (\tilde{\boldsymbol{F}} - \boldsymbol{F})\right) \\
&= \|\tilde{\boldsymbol{F}}\|_F^2 - 2 \operatorname{Tr}\left(\tilde{\boldsymbol{F}}^T \boldsymbol{F}\right) + C \\
&= \sum_{l=1}^{mn} \tilde{\boldsymbol{F}}_{l,:}^T \tilde{\boldsymbol{F}}_{:,l} - \tilde{\boldsymbol{F}}_{l,:}^T \boldsymbol{F}_{:,l} + C \\
&= \sum_{i=1}^n \|\boldsymbol{M}_i\|_F^2 - 2 \operatorname{Tr}\left(\boldsymbol{M}_i^T \mathbb{E}[\boldsymbol{g}_i \boldsymbol{g}_i^T]\right) + C
\end{aligned}$$

where $\tilde{\boldsymbol{F}}_{l,:} \in \mathbb{R}^{mn}$ indicates the $l^{\text{th}}$ row vector of $\tilde{\boldsymbol{F}}$ and $\tilde{\boldsymbol{F}}_{:,l}$ is the $l^{\text{th}}$ column vector. The last equation is obtained by the fact that $\tilde{\boldsymbol{F}}$ is a block diagonal matrix. So only the values of $\boldsymbol{F}$ at the position of non-zero values $\tilde{\boldsymbol{F}}$ contributes to the trace, which is exactly the outer product: $\mathbb{E}[\boldsymbol{g}_i \boldsymbol{g}_i^T]$.  $\square$

**Lemma G.2** (Powers-Stormer inequality). *For positive semi-definite operator $\boldsymbol{A}$, $\boldsymbol{B}$, we have the following inequality*

$$\operatorname{Tr}((\boldsymbol{A} - \boldsymbol{B})^T (\boldsymbol{A} - \boldsymbol{B})) \le \|\boldsymbol{A}^2 - \boldsymbol{B}^2\|_1 \tag{32}$$

*where $\|\cdot\|_1$ is the trace norm.*

## G.1  PROOF OF THEOREM 3.1

*Proof.* From Theorem G.1, we have

$$\begin{aligned}
&\|\tilde{\boldsymbol{F}} - \boldsymbol{F}\|_F^2 \\
&= \sum_{i=1}^n \|\boldsymbol{M}_i\|_F^2 - 2 \operatorname{Tr}\left(\boldsymbol{M}_i^T \mathbb{E}[\boldsymbol{g}_i \boldsymbol{g}_i^T]\right) \\
&= \sum_{i=1}^n \sum_{j=1}^m M_{i,jj}^2 - 2 M_{i,jj} \mathbb{E}[g_{i,j}^2]
\end{aligned}$$

By taking the derivative w.r.t $M_{i,jj}$, we have

$$M_{i,jj} = \mathbb{E}[g_{i,j}^2]$$

Thus, we have $\tilde{\boldsymbol{F}} = \operatorname{Diag}(\mathbb{E}[\vec{\boldsymbol{g}}^2])$.  $\square$

### G.2 Proof of Theorem 3.2

To prove this theorem, we need to leverage the Theorem 3.3 for generalized whitening (Eq. (3)) in Sec. 3.3. This is proved in Sec. G.3. But in the following, we will provide an alternative proof for completeness.

*Proof.* From Theorem G.1, we have

$$\|\tilde{\boldsymbol{F}} - \boldsymbol{F}\|_F^2$$

$$= \sum_{i=1}^{n} \|\boldsymbol{M}\|_F^2 - 2\operatorname{Tr}(\boldsymbol{M}^T \mathbb{E}[\boldsymbol{g}_i \boldsymbol{g}_i^T]) + C$$

$$= n\|\boldsymbol{M}\|_F^2 - 2\operatorname{Tr}(\boldsymbol{M}^T \mathbb{E}[\sum_{i=1}^{n} \boldsymbol{g}_i \boldsymbol{g}_i^T]) + C$$

$$= n\|\boldsymbol{M}\|_F^2 - 2\operatorname{Tr}(\boldsymbol{M}^T \mathbb{E}[\boldsymbol{G}\boldsymbol{G}^T]) + C$$

To minimize this, we take the derivative w.r.t. $\boldsymbol{M}$, we have

$$2n\boldsymbol{M} - 2\mathbb{E}[\boldsymbol{G}\boldsymbol{G}^T] = 0 \Rightarrow \boldsymbol{M} = \frac{1}{n}\mathbb{E}[\boldsymbol{G}\boldsymbol{G}^T]$$

$\square$

Next, we prove another proposition that is "symmetric" to the whitening results in Theorem 3.3.

**Proposition G.3.** *Assume $\mathcal{H} = \{\boldsymbol{R}_n \otimes \boldsymbol{I}_m\}$, where $\boldsymbol{R}_n \in \mathbb{R}^{n \times n}$ is SPD matrix, then Eq. (2) can be analytically solved with the optimal solution as*

$$\boldsymbol{R}_n^* = \frac{1}{m}\mathbb{E}[\boldsymbol{G}^T \boldsymbol{G}] \tag{33}$$

*Proof.* Since $\boldsymbol{R}_n \otimes \boldsymbol{I}_m$ does not have a nice block diagonal structure like the previous proposition, we need to analyze it a bit more. First, we have

$$\|\boldsymbol{R}_n \otimes \boldsymbol{I}_m - \boldsymbol{F}\|_F^2$$

$$= \|\boldsymbol{R}_n \otimes \boldsymbol{I}_m\|_F^2 - 2\operatorname{Tr}\left(\underbrace{(\boldsymbol{R}_n \otimes \boldsymbol{I}_m)^T \mathbb{E}[\vec{\boldsymbol{g}}\vec{\boldsymbol{g}}^T]}_{\boldsymbol{Z}}\right) + C$$

Since we only care about the diagonal of $\boldsymbol{Z}$, therefore, we only inspect the block diagonal of $\boldsymbol{Z}$ with each block $\boldsymbol{Z}_i$ of size $\mathbb{R}^{m \times m}$, and $i = 1, \ldots, n$. By basic algebra, we have

$$\boldsymbol{Z}_i = \sum_{k=1}^{n} R_{ik} \boldsymbol{g}_k \boldsymbol{g}_i^T$$

where $\boldsymbol{g}_k$ is the $k^{\text{th}}$ column of $\boldsymbol{G}$. Therefore, we can simplify the trace of $\boldsymbol{Z}$ as

$$\operatorname{Tr}(\boldsymbol{Z}) = \sum_{i=1}^{n} \operatorname{Tr}(\boldsymbol{Z}_i)$$

$$= \operatorname{Tr}(\sum_{i=1}^{n}\sum_{k=1}^{n} R_{ij} \boldsymbol{g}_k \boldsymbol{g}_i^T)$$

$$= \sum_{i=1}^{n}\sum_{k=1}^{n}\sum_{j=1}^{m} R_{ik}[\boldsymbol{G}]_{ji}[\boldsymbol{G}^T]_{kj}$$

where $[\boldsymbol{G}]_{ji}$ is the element of $\boldsymbol{G}$ at $j^{\text{th}}$ row and $i^{\text{th}}$ column.

Now, let's perform the same analysis of the following quantity

$$\text{Tr}\left((\boldsymbol{I}_m \otimes \boldsymbol{R}_n)\mathbb{E}[\overrightarrow{[\boldsymbol{G}^T]}\,\overrightarrow{[\boldsymbol{G}^T]}^T]\right)$$

where $\overrightarrow{[\boldsymbol{G}^T]}$ is the vectorized transposed gradient $\boldsymbol{G}^T$. Namely, it now stacks the rows of $\boldsymbol{G}$ instead of columns of $\boldsymbol{G}$ like $\vec{g}$. This object is simple to treat due to its block diagonal structure, by algebric manipulation, we have

$$\text{Tr}\left((\boldsymbol{I}_m \otimes \boldsymbol{R}_n)\mathbb{E}[\overrightarrow{[\boldsymbol{G}^T]}\,\overrightarrow{[\boldsymbol{G}^T]}^T]\right) = \underbrace{\sum_{k=1}^{m}}_{\text{over blocks}} \text{Tr}(R_{ij} \underbrace{[\boldsymbol{G}^T]_k}_{k\text{th column of } \boldsymbol{G}^T} [\boldsymbol{G}^T]_k^T)$$

$$= \sum_{k=1}^{m}\sum_{i=1}^{n}\sum_{j=1}^{n} R_{ij}[\boldsymbol{G}^T]_{jk}[\boldsymbol{G}]_{ki}$$

Now, let's change the variable $i = i$, $j = k$ and $k = j$, the above becomes

$$\text{Tr}\left((\boldsymbol{I}_m \otimes \boldsymbol{R}_n)\mathbb{E}[\overrightarrow{[\boldsymbol{G}^T]}\,\overrightarrow{[\boldsymbol{G}^T]}^T]\right)$$

$$= \sum_{j=1}^{m}\sum_{i=1}^{n}\sum_{k=1}^{n} R_{ik}[\boldsymbol{G}^T]_{kj}[\boldsymbol{G}]_{ji}$$

$$= \text{Tr}(\boldsymbol{Z}) \tag{34}$$

We should also note that

$$\|\boldsymbol{R}_n \otimes \boldsymbol{I}_m\|_F^2$$
$$= \text{Tr}\left((\boldsymbol{R}_n \otimes \boldsymbol{I}_m)^T(\boldsymbol{R}_n \otimes \boldsymbol{I}_m)\right)$$
$$= \text{Tr}\left((\boldsymbol{R}_n^T\boldsymbol{R}_n) \otimes \boldsymbol{I}_m\right)$$
$$= \text{Tr}(\boldsymbol{R}_n^T\boldsymbol{R}_n)\,\text{Tr}(\boldsymbol{I}_m)$$
$$= \text{Tr}\left((\boldsymbol{I}_m \otimes \boldsymbol{R}_n)^T(\boldsymbol{I}_m \otimes \boldsymbol{R}_n)\right)$$
$$= \|(\boldsymbol{I}_m \otimes \boldsymbol{R}_n)\|_F^2$$

Therefore, by using the above equation and Eq. (34), the original minimization problem is translated to

$$\underset{\boldsymbol{R}_n}{\arg\min}\,\|\boldsymbol{R}_n \otimes \boldsymbol{I}_m - \boldsymbol{F}\|_F^2 = \underset{\boldsymbol{R}_n}{\arg\min}\,\|\boldsymbol{I}_m \otimes \boldsymbol{R}_n - \mathbb{E}[\overrightarrow{[\boldsymbol{G}^T]}\,\overrightarrow{[\boldsymbol{G}^T]}^T]\|_F^2$$

Thus, we can leverage Theorem 3.3 to obtain the optimal solution

$$\boldsymbol{R}_n^* = \frac{1}{m}\mathbb{E}[\boldsymbol{G}^T\boldsymbol{G}]$$

$\square$

With the above two propositions, we can start to prove Theorem 3.2.

*Proof.* First, we note that

$$\|\boldsymbol{R}_n^{\frac{1}{2}} \otimes \boldsymbol{L}_m^{\frac{1}{2}} - \boldsymbol{F}\|_F^2$$
$$= \|\underbrace{(\boldsymbol{R}_n \otimes \boldsymbol{I}_m)^{\frac{1}{2}}}_{A}\underbrace{(\boldsymbol{I}_n \otimes \boldsymbol{L}_m)^{\frac{1}{2}}}_{B} - \underbrace{\mathbb{E}[\vec{g}\vec{g}^T]^{\frac{1}{2}}}_{C}\mathbb{E}[\vec{g}\vec{g}^T]^{\frac{1}{2}}\|_F^2$$
$$= \|\boldsymbol{A}\boldsymbol{B} - \boldsymbol{C}\boldsymbol{C}\|_F^2$$

Next, we will upper bound this quantity. First, we have

$$AB - CC = A(B - C) + (A - C)C$$

By triangular inequality, we have

$$\|AB - CC\|_F$$
$$\leq \|A(B - C)\|_F + \|(A - C)C\|_F$$
$$\leq \|A\|_F\|B - C\|_F + \|A - C\|_F\|C\|_F$$
$$\leq (\|A - C\|_F + \|C\|_F)\|B - C\|_F + \|A - C\|_F\|C\|_F$$
$$= \|A - C\|_F\|B - C\|_F + \|C\|_F(\|B - C\|_F + \|A - C\|_F)$$

Now, the squared norm can be upper bounded by

$$\|AB - CC\|_F^2 \leq 3\left(\|A - C\|_F^2\|B - C\|_F^2 + \|C\|_F^2\|A - C\|_F^2 + \|C\|_F^2\|B - C\|_F^2\right)$$
$$\leq 3(mn\|A^2 - C^2\|_F\|B^2 - C^2\|_F + \sqrt{mn}\|C\|_F^2\|A^2 - C^2\|_F \tag{35}$$
$$+ \sqrt{mn}\|C\|_F^2\|B^2 - C^2\|_F) \tag{36}$$

The first inequality is obtained by the fact that for any three matrix $P$, $Q$ and $H$, we have

$$\|P + Q + H\|_F^2 \leq (\|P\|_F + \|Q\|_F + \|H\|_F)^2$$
$$= \|P\|_F^2 + \|Q\|_F^2 + \|H\|_F^2 + 2\|P\|_F\|Q\|_F + 2\|P\|_F\|H\|_F + 2\|Q\|_F\|H\|_F$$
$$\leq 3\left(\|P\|_F^2 + \|Q\|_F^2 + \|H\|_F^2\right)$$

The second inequality is obtained by directly applying Powers-Stormer's inequality and Holder's inequality. For completeness, we will show how to upper-bound $\|A - C\|_F^2$, the rest can be bounded in the same way. From Theorem G.2 and both $A$, $C$ are SPD matrix, we have

$$\|A - C\|_F^2 \leq \|A^2 - C^2\|_1$$

Then, we can select $p = q = 2$ for Holder's inequaity and obtain

$$\|A^2 - C^2\|_1 \leq \sqrt{mn}\|A^2 - C^2\|_F$$

where $\sqrt{mn}$ comes from the $\|I_{mn}\|_F$ in Holder's inequality. By substitute it back, we obtain the upper bound.

We can see that minimizing the upper bound Eq. (36) is equivalent to minimize each $\|A^2 - C^2\|_F$, $\|B^2 - C^2\|_F$ individually, and

$$\|A^2 - C^2\|_F = \|R_n \otimes I_m - F\|_F$$
$$\|B^2 - C^2\|_F = \|I_n \otimes L_m - F\|_F$$

Thus, from Theorem 3.3 and Theorem G.3, we prove the theorem. $\square$

### G.3 PROOF OF THEOREM 3.3 AND THEOREM 4.1

Instead of proving the Theorem 3.3, we propose a generalization to those gradient operations, where Theorem 3.3 is a special case.

**Structure assumption**    We consider $\mathcal{H} = \{S \otimes M\}$ with identical SPD $M \in \mathbb{R}^{m \times m}$ and positive diagonal $S \in \mathbb{R}^{n \times n}$. The following theorem proves that the optimal solution can be solved by a fixed-point iteration.

**Theorem G.4.** *Assuming $\mathcal{H} = \{S \otimes M\}$ with positive diagonal $S \in \mathbb{R}^{n \times n}$ and SPD $M \in \mathbb{R}^{m \times m}$, and $\mathbb{E}_{GG'}[(G^T G')^2]$ contains positive values, solving Eq. (2) admits a fixed point procedure:*

$$\mathrm{Diag}(S) = \frac{\mathrm{Diag}(\mathbb{E}[G^T M G])}{\|M\|_F^2}, \quad M = \frac{\mathbb{E}[GSG^T]}{\|S\|_F^2}. \tag{37}$$

*The solution $\mathrm{Diag}(S^*)$ converges to the principal eigenvector of $\mathbb{E}[(G^T G')^2]$ up to a scaling with unique $S^* \otimes M^*$.*

To prove Theorem G.4, we first introduce some classic results.

**Theorem G.5** (Perron-Frobenius theorem). *For a matrix $A \in \mathbb{R}^{n \times n}$ with positive entries, the principal eigenvalue $r$ is positive, called Perron-Frobenius eigenvalue. The corresponding eigenvector $v$ of $A$ is called Perron vector and only contains positive components: $Av = rv$ with $v_i > 0$. In addition, there are no other positive eigenvectors of $A$.*

**Definition G.6** (Hilbert projective metric). For any given vectors $v$, $w$ in $C/\{0\}$ where $C$ is a closed convex pointed non-negative cone $C$, i.e. $C \cap (-C) = \{0\}$, the Hilbert projective metric is defined as

$$d_H(\boldsymbol{v}, \boldsymbol{w}) = \log\left(\max_i \frac{v_i}{w_i}\right) - \log\left(\min_i \frac{v_i}{w_i}\right)$$

This is a pseudo metric since it has a scaling invariance property: $d_H(\boldsymbol{v}, \alpha\boldsymbol{m}) = d_H(\boldsymbol{v}, \boldsymbol{m})$ for $\alpha > 0$. This means $d_H(\boldsymbol{v}, \boldsymbol{m}) = 0$ does not mean $\boldsymbol{v} = \boldsymbol{m}$ but $\boldsymbol{v} = \alpha\boldsymbol{m}$ with some positive scaling $\alpha$. However, this is a metric on the space of rays inside the cone.

**Theorem G.7** (Birkhoff-Hopf theorem). *Let $P \in \mathbb{R}^{n \times n}$ be a positive matrix and let*

$$\kappa(\boldsymbol{P}) = \inf\left\{\alpha \geq 0 : d_H(\boldsymbol{P}\boldsymbol{x}, \boldsymbol{P}\boldsymbol{y}) \leq \alpha d_H(\boldsymbol{x}, \boldsymbol{y}), \forall \boldsymbol{x}, \boldsymbol{y} \in C_+, \boldsymbol{x} \sim \boldsymbol{y}\right\}$$

*where $C_+$ is the cone that each element is non-negative and $\sim$ is the induced equivalence relation. Namely, if $\boldsymbol{x} \sim \boldsymbol{y}$, there exists $\alpha, \beta > 0$ such that $\alpha\boldsymbol{x} < \boldsymbol{y} < \beta\boldsymbol{x}$, and $\boldsymbol{x} < \boldsymbol{y}$ means $y - x \in C_+$. Then, it holds*

$$\kappa(\boldsymbol{P}) = \tanh \frac{1}{4}\Delta(\boldsymbol{P}) \quad \text{with } \Delta(\boldsymbol{P}) = \max_{i,j,k,l} \frac{P_{ij}P_{kl}}{P_{il}P_{kj}}$$

This theorem suggests that when $P$ is a positive matrix, the corresponding linear mapping is contractive since $\tanh(\cdot) \leq 1$ under Hilbert projective metric.

Now, let's prove the Theorem G.4.

*Proof.* First, we can simplify the Eq. (2) using Theorem G.1:

$$\|\boldsymbol{S} \otimes \boldsymbol{M} - \boldsymbol{F}\|_F^2$$
$$= \sum_{i=1}^n S_i^2\|\boldsymbol{F}\|_F^2 - 2\operatorname{Tr}(S_i\boldsymbol{M}\mathbb{E}[\boldsymbol{g}_i\boldsymbol{g}_i^T]) + C$$

Then, we simply take its derivative w.r.t. $s_i$, and obtain

$$2S_i\|\boldsymbol{M}\|_F^2 = 2\operatorname{Tr}(\boldsymbol{M}\mathbb{E}[\boldsymbol{g}_i\boldsymbol{g}_i^T])$$
$$\implies S_i = \frac{\operatorname{Tr}(\boldsymbol{M}\mathbb{E}[\boldsymbol{g}_i\boldsymbol{g}_i^T])}{\|\boldsymbol{M}\|_F^2}$$
$$\implies \operatorname{Diag}(\boldsymbol{S}) = \frac{\operatorname{Diag}\left(\mathbb{E}[\boldsymbol{G}^T\boldsymbol{M}\boldsymbol{G}]\right)}{\|\boldsymbol{M}\|_F^2}$$

Similarly, we have

$$\boldsymbol{M} = \frac{\sum_{i=1}^n S_i\mathbb{E}[\boldsymbol{g}_i\boldsymbol{g}_i^T]}{\|\boldsymbol{S}\|_F^2}$$
$$= \frac{\mathbb{E}[\boldsymbol{G}\boldsymbol{S}\boldsymbol{G}^T]}{\|\boldsymbol{S}\|_F^2}$$

These define an iterative procedure. Next, we will show it converges. Let's substitute $M$ into $S$, and obtain

$$\boldsymbol{S} = \operatorname{Diag}\left(\mathbb{E}_G\left[\boldsymbol{G}^T\mathbb{E}_{G'}[\boldsymbol{G}'\boldsymbol{S}\boldsymbol{G}'^T]\boldsymbol{G}\right]\right)\alpha(\boldsymbol{S})$$
$$= \operatorname{Diag}\left(\mathbb{E}_{GG'}\left[\underbrace{\boldsymbol{G}\boldsymbol{G}'^T}_{\boldsymbol{H}}\boldsymbol{S}\boldsymbol{G}'^T\boldsymbol{G}\right]\right)$$

where $\alpha(\boldsymbol{S})$ is the scaling term. In the following, we use $\mathbb{E}$ as $\mathbb{E}_{GG'}$. Since we can show

$$S_i = \mathbb{E}\left[\sum_j^n S_j\right],$$

we can write $\boldsymbol{S}$ in its vector format:

$$\boldsymbol{s} = \underbrace{\mathbb{E}\left[\boldsymbol{H}^2\right]}_{P}\boldsymbol{s}.$$

From the assumption, we know $\boldsymbol{P}$ contains only positive values, let's define a quotient space for positive vectors $\boldsymbol{s}$ and $\boldsymbol{q}$ under the equivalence relation $\boldsymbol{s} \sim \boldsymbol{s}'$ if $\boldsymbol{s} = \alpha\boldsymbol{s}'$ for some positive scaling $\alpha$. Namely, we define a space of rays inside the positive cone. Therefore, the Hilbert projective metric becomes a real metric inside the quotient space.

From the Theorem G.7, we know the linear mapping associated with $\boldsymbol{P}$ is contractive. Therefore, we can follow the proof of Banach fixed point theorem on the previously defined quotient space with Hilbert projective metric to show the convergence of this fixed point iteration on $\boldsymbol{s}$.

Now, we show the solution $\boldsymbol{s}^*$ is always positive. Since it is converging, therefor, the solution satisfies

$$\boldsymbol{s}^* = \alpha(\boldsymbol{s}^*)\boldsymbol{P}\boldsymbol{s}^*$$

This is equivalent to finding the eigenvectors of $\boldsymbol{P}$. By leveraging Perron-Frobenius theorem (Theorem G.5), we know $\boldsymbol{s}^*$ is the principal eigenvector of $\boldsymbol{P}$, and only contain positive values. It is also easy to verify that this fixed point converges upto a positive scaling factor (this is expected since the contractive mapping holds true for the quotient space with Hilbert metric, that is invariant to scaling.)

Although $\boldsymbol{s}^*$ is not unique, but $\boldsymbol{S} \otimes \boldsymbol{M}$ is, since for arbitrary positive scaling $\beta$

$$\boldsymbol{s}'^* = \beta\boldsymbol{s}^* \implies \boldsymbol{M}'^* = \frac{1}{\beta}\frac{\mathbb{E}[\boldsymbol{G}\boldsymbol{S}^*\boldsymbol{G}^T]}{\|\boldsymbol{s}\|_2^2}$$
$$\implies \boldsymbol{S}'^* \otimes \boldsymbol{M}'^* = \boldsymbol{S}^* \otimes \boldsymbol{M}^*$$

$\square$

Therefore, Theorem 3.3 is a direct consequence by substituting $\tilde{\boldsymbol{F}} = \boldsymbol{I}_n \otimes \boldsymbol{M}$ and $\tilde{\boldsymbol{F}} = \boldsymbol{S} \otimes \boldsymbol{I}_m$ into Eq. (15).

Next, we prove Theorem 4.1.

*Proof.* From the Theorem G.4, the iterative procedure for $\boldsymbol{Q}$ can be simply obtained by taking the diagonals of $\boldsymbol{M}$:

$$\boldsymbol{Q} = \frac{\mathrm{Diag}\left(\mathbb{E}\left[\boldsymbol{G}\boldsymbol{S}\boldsymbol{G}^T\right]\right)}{\|\boldsymbol{S}\|_F^2}.$$

Following the same proof strategy of Theorem G.4, we substitute $\boldsymbol{Q}$ into the update of $\boldsymbol{S}$ and re-write it into the vector format. First, let's rewrite the update of $\boldsymbol{S}$

$$S_i \propto \mathbb{E}[\sum_{j=1} G_{ji}^2 Q_j]$$
$$\implies \boldsymbol{s} = \frac{\boldsymbol{P}^T\boldsymbol{q}}{\|\boldsymbol{q}\|_2^2}$$

where $\boldsymbol{P} = \mathbb{E}[\boldsymbol{G}^2]$. Similarly, $\boldsymbol{q} = \frac{\boldsymbol{P}\boldsymbol{s}}{\|\boldsymbol{s}\|_2^2}$. Thus,

$$\boldsymbol{s} = \alpha(\boldsymbol{s})\boldsymbol{P}^T\boldsymbol{P}\boldsymbol{s}$$

From the assumption $\boldsymbol{P}$ contains only positive values, we can follow the exact same argument made in Theorem G.4 to show the convergence of this fixed point update and the positivity of the final solution $\boldsymbol{s}^*$. Precisely, $\boldsymbol{s}^*$ and $\boldsymbol{q}^*$ are the right and left principal singular vectors of $\boldsymbol{P}$, respectively, and $\boldsymbol{S}^* \otimes \boldsymbol{Q}^*$ are unique. $\square$

### G.4 Proofs of OSOAP

#### G.4.1 Proof of Theorem 3.4

*Proof.* For simplicity, we omit the subscript $f$ in $U_f$. If we assume all $D_i$ are equal and only contain positive values, then each block $UD_iU^T$ are the same for all $i$, and it is SPD matrix. Then, to minimize the the loss Eq. (2), we can directly leverage the whitening results in Theorem 3.3, and obtain $M^* = \mathbb{E}[GG^T]$. Due to the structure of $UDU^T$, the optimal $U^*$ is exactly the eigen-matrix of $M^*$.

Next, we prove for any fixed $U$, we can find the corresponding optimal $D_i$. From the block diagonal structure and Theorem G.1, we have

$$\|\tilde{F} - F\|_F^2$$
$$= \sum_{i=1}^n \|UD_iU^T\|_F^2 - 2\operatorname{Tr}\left(U^T\mathbb{E}[g_ig_i^T]UD_i\right) + C$$
$$= \sum_{i=1}^n \|D_i\|_F^2 - 2\operatorname{Tr}\left(U^T\underbrace{\mathbb{E}[g_ig_i^T]}_{H_i}UD_i\right) + C$$
$$= \sum_{i=1}^n\sum_{j=1}^m D_{i,jj}^2 - 2\sum_{i=1}^n\sum_{j=1}^m D_{i,jj}u_j^TH_iu_j + C$$

Taking the derivative w.r.t. $D_{i,jj}$, we can find the optimal $D_{i,jj}$ is

$$D_{i,jj}^* = u_j^TH_iu_j$$
$$= \mathbb{E}[(u_j^Tg_i)^2]$$

Now, by simple algebra manipulation, we have

$$D_i^* = \operatorname{Diag}_M(\mathbb{E}[(U^Tg_i)])$$

where $\operatorname{Diag}_M$ is to expand the vector to a diagonal matrix. Finally, for $\tilde{D}$, we have

$$\tilde{D} = \operatorname{Diag}_M(\mathbb{E}[(U^TG)^2]) \tag{38}$$

The optimality of $\widetilde{D}$ can also be obtained by leveraging the Lemma 1 in (George et al., 2018), and set the eigenbasis as $I_n \otimes U$. $\square$

### G.5 Proof of Theorem G.8

Here, we present the following proposition that analyzes the issue casued by using low-rank tracking $\widetilde{Q}$ compared to the true $Q$:

**Proposition G.8** (Subspace switching)**.** *Assuming the setup mentioned above and all assumptions of OSOAP are satisfied. We further assume the low-rank $U \in \mathbb{R}^{m\times r}$ is obtained at the beginning of $i+1$ time block by $\operatorname{EVD}(Q_{ik}^*, r)$ where $Q_{ik}^*$ is the true tracking state. Further, we assume the stability of the eigen-basis such that gradient $G_t$ during $i+1$ time block shares the same eigen-basis as $Q_{ik}^*$. Then, we have the following for true statistics $Q_{(i+1)k}^*$ at the end of the $i+1$ time block:*

$$Q_{(i+1)k}^* = \sum_{t=ik+1}^{(i+1)k} G_tG_t^T = \sum_{t=ik+1}^{(i+1)k} \widetilde{G}_t\widetilde{G}_t^T + U_c\Sigma_tU_c^T \tag{39}$$

*where $\widetilde{G}_t = U\sigma_t$ is the low rank reconstructed gradients, $\Sigma_t \in \mathbb{R}^{(m-r)\times(m-r)}$ is a diagonal matrix with positive values, and $U_c$ is the complement eigen-basis such that $[U, U_c]$ will form the complete eigen-basis of $Q_{ik}^*$.*

*Proof.* Within the time block $i+1$ with low-rank mapping $U$, the gradient at each step can be decomposed as

$$G_t = \underbrace{UU^TG_t}_{\widetilde{G}_t} + \underbrace{(G_t - UU^TG_t)}_{R_t}$$

Therefore, the true state $Q^*_{(i+1)k}$ can be simplified as

$$Q_{(i+1)k} = \sum_{t=ik+1}^{(i+1)k} G_t G_t^T$$

$$= \sum_{t=ik+1}^{(i+1)k} (\widetilde{G}_t + R_t)(\widetilde{G}_t + R_t)^T$$

$$= \sum_{t=ik+1}^{(i+1)k} \widetilde{G}_t \widetilde{G}_t^T + \underbrace{\widetilde{G}_t R_t^T}_{0} + \underbrace{R_t \widetilde{G}_t^T}_{0} + R_t R_t^T$$

The third equality is obtained because we assume $G_t G_t^T$ shares the same eigen-basis as $Q^*_{ik}$. Namely,

$$G_t G_t^T = [U, U_c] \begin{bmatrix} A_t & 0 \\ 0 & \Sigma_t \end{bmatrix} \begin{bmatrix} U^T \\ U_c^T \end{bmatrix}$$

$$= U A_t U^T + U_c \Sigma_t U_c^T$$

where $A_t$ and $\Sigma_t$ are diagonal matrix. Then, we have

$$\widetilde{G}_t R_t^T$$
$$= U U^T G_t (U_c U_c^T G_t)^T$$
$$= U U^T (U A_t U^T + U_c \Sigma_t U_c^T) U_c U_c^T$$
$$= U A_t \underbrace{U^T U_c}_{0} U_c^T + U \underbrace{U^T U_c}_{0} \Sigma_t U_c^T U_c U_c^T$$
$$= 0$$

In addition, we can also simplify

$$R_t R_t^T$$
$$= U_c U_c^T G_t G_t^T U_c U_c^T$$
$$= U_c U_c^T (U A_t U^T + U_c \Sigma_t U_c^T) U_c U_c^T$$
$$= U_c \Sigma_t U_c^T$$

Therefore,

$$Q^*_{(i+1)k} = \sum_{t=ik+1}^{(i+1)k} \widetilde{G}_t \widetilde{G}_t^T + U_c \Sigma_t U_c^T$$

$\square$

### G.6 PROOF OF THEOREM G.9

Here, we prove the following theorem to analytically search for the optimal scaling $S_t$ by minimizing the reconstruction error under Frobenius norm:

**Theorem G.9** (Optimal compensation). *Assume that the conditions of OSOAP are satisfied. With the proposed form of compensation (Eq. (14)), minimizing FIM reconstruction loss*

$$\|(S_t^{-2} \otimes U_c U_c^T) - \tilde{F}_c\|_F^2$$

*admits analytic solution:*

$$\mathrm{Diag}(S_t) = \frac{\sqrt{m-r}}{\sqrt{\mathbb{E}[\mathbf{1}_m^T G_t^2 - \mathbf{1}_r^T (U^T G_t)^2]}} \tag{40}$$

*where $\mathbf{1}_m \in \mathbb{R}^m$, $\mathbf{1}_r \in \mathbb{R}^r$ are the column vectors with element $1$ and $^2$ is element-wise square.*

*Proof.* For simplicity, we ignore the subscript $t$ for the following proof. First, we let $\boldsymbol{O} = \boldsymbol{S}^{-2}$ Then, the loss function can be written as

$$\|\boldsymbol{O} \otimes \boldsymbol{U}_c \boldsymbol{U}_c^T - \tilde{\boldsymbol{F}}_c\|_F^2$$

$$= \sum_{i=1}^n \|O_{ii} \boldsymbol{U}_c \boldsymbol{U}_c^T\|_F^2 - 2\operatorname{Tr}((O_{ii}\boldsymbol{U}_c \boldsymbol{U}_c^T)^T (\boldsymbol{U}_c \boldsymbol{M}_i \boldsymbol{U}_c^T))$$

$$= \sum_{i=1}^n \|O_{ii} \boldsymbol{U}_c \boldsymbol{U}_c^T\|_F^2 - 2\operatorname{Tr}(O_{ii}\boldsymbol{M}_i)$$

$$= \sum_{i=1}^n \sum_{k=1}^{m-r} \sum_{j=1}^m O_{ii}^2 U_{c,jk}^2 - 2\operatorname{Tr}(O_{ii}\boldsymbol{M}_i)$$

where $\boldsymbol{M}_i = \operatorname{Diag}(\mathbb{E}[(\boldsymbol{U}_c^T \boldsymbol{g}_i)^2])$, $\boldsymbol{g}_i$ is the $i^{\text{th}}$ column of $\boldsymbol{G}$, and $U_{c,jk}$ is the element in $j^{\text{th}}$ row, $k^{\text{th}}$ column of $\boldsymbol{U}_c$. Then, we take the derivative w.r.t. $O_{ii}$, and we have

$$2O_{ii} \sum_{k=1}^{m-r} \sum_{j=1}^m U_{c,jk}^2 = 2 \sum_{k=1}^{m-r} \mathbb{E}[(\boldsymbol{U}_c^T \boldsymbol{g}_i)_k^2]$$

$$\Longrightarrow O_{ii} = \frac{\mathbb{E}[\sum_{k=1}^{m-r} (\boldsymbol{U}_c^T \boldsymbol{g}_i)_k^2]}{m-r}$$

This form still requires the access to $\boldsymbol{U}_c$. Next, let's simplify it. First, let $\tilde{\boldsymbol{U}} = [\boldsymbol{U}, \boldsymbol{U}_c]$ to be the complete basis, we can show

$$\sum_{k=1}^m (\tilde{\boldsymbol{U}}^T \boldsymbol{g}_i)_k^2$$

$$= \operatorname{Tr}((\tilde{\boldsymbol{U}}^T \boldsymbol{g}_i)^T (\tilde{\boldsymbol{U}}^T \boldsymbol{g}_i))$$

$$= \operatorname{Tr}(\boldsymbol{g}_i^T \underbrace{\tilde{\boldsymbol{U}} \tilde{\boldsymbol{U}}^T}_{\boldsymbol{I}} \boldsymbol{g}_i)$$

$$= \boldsymbol{g}_i^T \boldsymbol{g}_i$$

Now, let's re-write the above in a different format:

$$\sum_{k=1}^m (\tilde{\boldsymbol{U}}^T \boldsymbol{g}_i)_k^2$$

$$= \operatorname{Tr}(\boldsymbol{g}_i^T \tilde{\boldsymbol{U}} \tilde{\boldsymbol{U}}^T \boldsymbol{g}_i)$$

$$= \operatorname{Tr}(\tilde{\boldsymbol{U}}^T \boldsymbol{g}_i \boldsymbol{g}_i^T \tilde{\boldsymbol{U}})$$

$$= \operatorname{Tr}\left(\begin{bmatrix} \boldsymbol{U}^T \\ \boldsymbol{U}_c^T \end{bmatrix} \boldsymbol{g}_i \boldsymbol{g}_i^T [\boldsymbol{U}, \boldsymbol{U}_c]\right)$$

$$= \operatorname{Tr}(\boldsymbol{U}\boldsymbol{U}^T (\boldsymbol{g}_i \boldsymbol{g}_i^T) + \boldsymbol{U}_c \boldsymbol{U}_c^T (\boldsymbol{g}_i \boldsymbol{g}_i^T))$$

$$= \operatorname{Tr}((\boldsymbol{U}^T \boldsymbol{g}_i)^T (\boldsymbol{U}^T \boldsymbol{g}_i)) + \operatorname{Tr}((\boldsymbol{U}_c^T \boldsymbol{g}_i)^T (\boldsymbol{U}_c^T \boldsymbol{g}_i))$$

$$= \sum_{k=1}^r (\boldsymbol{U}^T \boldsymbol{g}_i)_k^2 + \sum_{k=1}^{m-r} (\boldsymbol{U}_c^T \boldsymbol{g}_i)_k^2$$

Therefore, we have

$$\mathbb{E}[\sum_{k=1}^{m-r} (\boldsymbol{U}_c^T \boldsymbol{g}_i)_k^2] = \mathbb{E}[\boldsymbol{g}_i^T \boldsymbol{g}_i - \sum_{k=1}^r (\boldsymbol{U}^T \boldsymbol{g}_i)_k^2]$$

So, we have

$$\operatorname{Diag}(\boldsymbol{O}) = \frac{\mathbb{E}[\mathbf{1}_m^T \boldsymbol{G}^2 - \mathbf{1}_r^T (\boldsymbol{U}^T \boldsymbol{G})^2]}{m-r}$$

and

$$\text{Diag}(\boldsymbol{D}) = \frac{\sqrt{m-r}}{\sqrt{\mathbb{E}[\boldsymbol{1}_m^T \boldsymbol{G}^2 - \boldsymbol{1}_r^T (\boldsymbol{U}^T \boldsymbol{G})^2]}}$$

$\square$

### G.7 PROOF OF THEOREM E.1

*Proof.* The proof strategy is a straightforward combination of Theorem 3.2 and Theorem 3.4. First, when we assume $\tilde{\boldsymbol{D}}$ has the Kronecker product structure, one can easily write

$$(\boldsymbol{U}_R \otimes \boldsymbol{U}_L)(\boldsymbol{S}_R \otimes \boldsymbol{S}_L)(\boldsymbol{U}_R \otimes \boldsymbol{U}_L)^T$$
$$= (\boldsymbol{U}_R \otimes \boldsymbol{U}_L) \left[ (\boldsymbol{S}_R \boldsymbol{U}_R^T) \otimes (\boldsymbol{S}_L \boldsymbol{U}_L^T) \right]$$
$$= \underbrace{(\boldsymbol{U}_R \boldsymbol{S}_R \boldsymbol{U}_R^T)}_{\boldsymbol{A}} \otimes \underbrace{(\boldsymbol{U}_L \boldsymbol{S}_L \boldsymbol{U}_L^T)}_{\boldsymbol{B}}$$

Therefore the loss (Eq. (2)) becomes

$$\|\boldsymbol{A}\boldsymbol{B} - \boldsymbol{C}\boldsymbol{C}\|_F^2$$

where $\boldsymbol{C} = \mathbb{E}[\vec{\boldsymbol{g}}\vec{\boldsymbol{g}}^T]^{\frac{1}{2}}$. This is exactly the formulation used in Theorem 3.2 with $\boldsymbol{R}_n^{\frac{1}{2}} = \boldsymbol{U}_R \boldsymbol{S}_R \boldsymbol{U}_R^T$ and $\boldsymbol{L}_m^{\frac{1}{2}} = \boldsymbol{U}_L \boldsymbol{S}_L \boldsymbol{U}_L^T$.

Thus, by directly utilizing Theorem 3.2, we can see the optimal solution

$$\boldsymbol{U}_R \boldsymbol{S}_R^2 \boldsymbol{U}_R^T = \mathbb{E}[\boldsymbol{G}^T \boldsymbol{G}]$$
$$\boldsymbol{U}_L \boldsymbol{S}_L^2 \boldsymbol{U}_L^T = \mathbb{E}[\boldsymbol{G}\boldsymbol{G}^T]$$

Due to the structural assumption of $\boldsymbol{U}_R, \boldsymbol{U}_L, \boldsymbol{S}_R, \boldsymbol{S}_L$, their corresponding optimal solution can directly obtained using eigenvalue decomposition.

Now, let's prove the optimal $\tilde{\boldsymbol{D}}$ with any fixed $\boldsymbol{U}_R, \boldsymbol{U}_L$. This is also straightforward by applying the same technique as Theorem 3.4. The loss can be written as

$$\| \underbrace{(\boldsymbol{U}_R \otimes \boldsymbol{U}_L)}_{\Pi} \tilde{\boldsymbol{D}} \underbrace{(\boldsymbol{U}_R \otimes \boldsymbol{U}_L)^T}_{\Pi^T} - \boldsymbol{F} \|_F^2$$
$$= \|\Pi \tilde{\boldsymbol{D}} \Pi^T\|_F^2 - 2 \operatorname{Tr}\left( \Pi^T \mathbb{E}[\vec{\boldsymbol{g}}\vec{\boldsymbol{g}}^T] \Pi \tilde{\boldsymbol{D}} \right) + C$$

Since it is easy to verify orthonormality of $\Pi$, i.e. $\Pi^T \Pi = \boldsymbol{I}$, the above is simplified to

$$\|\tilde{\boldsymbol{D}}\|_F^2 - 2 \operatorname{Tr}\left( \Pi^T \mathbb{E}[\vec{\boldsymbol{g}}\vec{\boldsymbol{g}}^T] \Pi \tilde{\boldsymbol{D}} \right)$$
$$= \sum_{i=1}^{mn} D_{ii}^2 - 2 \sum_{i=1}^{mn} D_{ii} [\Pi]_i^T \mathbb{E}[\vec{\boldsymbol{g}}\vec{\boldsymbol{g}}^T][\Pi]_i$$

where $[\Pi]_i$ is the $i^{\text{th}}$ column of matrix $\Pi$. Then, by taking the derivative, the optimal $D_{ii}$:

$$D_{ii}^* = [\Pi]_i^T \mathbb{E}[\vec{\boldsymbol{g}} \operatorname{Vec}^T][\Pi]_i$$
$$= \mathbb{E}[([\Pi]_i^T \vec{\boldsymbol{g}})^2]$$

Therefore,

$$\text{Diag}(\tilde{\boldsymbol{D}}^*) = \mathbb{E}[(\Pi^T \vec{\boldsymbol{g}})^2]$$
$$= \mathbb{E}[((\boldsymbol{U}_R^T \otimes \boldsymbol{U}_L^T)\vec{\boldsymbol{g}})^2]$$
$$= \operatorname{Vec}((\mathbb{E}[\boldsymbol{U}_L^T \boldsymbol{G} \boldsymbol{U}_R]))$$

$$\Rightarrow \tilde{\boldsymbol{D}}^* = \text{Diag}_M((\mathbb{E}[(\boldsymbol{U}_L^T \boldsymbol{G} \boldsymbol{U}_R)^2]))$$

$\square$

# H   CONVERGENCE OF ALICE

In this section, we will prove the convergence of Alice under the continuous-time setup. We initially want to adopt the convergence proof techniques from GaLore (Zhao et al., 2024a). However, the major drawback of their approach is the assumption of the fixed projection $U$. In practice, one will not fix the projection, and instead, it is crucial to switch different $U$ to achieve satisfactory performance. Therefore, this mismatch requires us to find an alternative technique that can handle the time-varying $U$. (Maddison et al., 2018) proposed an elegant framework for analyzing the behavior of optimizers by converting the optimizer updates into continuous-time ODEs and examining a suitable Hamiltonian function. This approach has been recently adopted to analyze the optimizers' convergence behavior (Nguyen et al., 2025; Chen et al., 2023).

Let's first write down the discretized update of Alice:

$$
\begin{aligned}
\boldsymbol{W}_t &= \boldsymbol{W}_{t-1} - \eta\left(\boldsymbol{U}\frac{\boldsymbol{M}_t}{\sqrt{\boldsymbol{V}_t}+\epsilon} + \sqrt{m-r}\frac{\boldsymbol{G}_t - \boldsymbol{U}\boldsymbol{U}^T\boldsymbol{G}_t}{\sqrt{\boldsymbol{\Phi}_t}+\epsilon}\right) \\
\boldsymbol{M}_t &= \beta_1\boldsymbol{M}_{t-1} + (1-\beta_1)\boldsymbol{U}^T\boldsymbol{G}_t \\
\boldsymbol{V}_t &= \beta_2\boldsymbol{V}_{t-1} + (1-\beta_2)(\boldsymbol{U}^T\boldsymbol{G}_t)^2 \\
\boldsymbol{\Phi}_t &= \beta_1\boldsymbol{\Phi}_{t-1} + (1-\beta_1)\mathbf{1}_m\left[\mathbf{1}_m^T\boldsymbol{G}_t^2 - \mathbf{1}_r^T(\boldsymbol{U}^T\boldsymbol{G})^2\right] \\
&= \beta_1\boldsymbol{H}_{t-1} + (1-\beta_1)\mathbf{1}_m\mathbf{1}_m^T(\boldsymbol{G}_t - \boldsymbol{U}\boldsymbol{U}^T\boldsymbol{G}_t)^2
\end{aligned}
$$

where $\eta$ is the effective learning rate, $\boldsymbol{U} \in \mathbb{R}^{m\times r}$ is the low-rank projection, $\boldsymbol{A}^2$, $\frac{\boldsymbol{A}}{\boldsymbol{B}}$ and $\sqrt{\cdot}$ are elementwise operations for any matrix $\boldsymbol{A}$, $\boldsymbol{B}$.

Next, we can convert the above discretized update into continuous-time ODEs:

$$
\begin{aligned}
\frac{d\boldsymbol{W}_t}{dt} &= -\boldsymbol{U}_t\frac{\boldsymbol{M}_t}{\sqrt{\boldsymbol{V}_t}+\epsilon} - \sqrt{m-r}\frac{\boldsymbol{G}_t - \boldsymbol{U}_t\boldsymbol{U}_t^T\boldsymbol{G}_t}{\sqrt{\boldsymbol{\Phi}_t}+\epsilon} \\
\frac{d\boldsymbol{M}_t}{dt} &= \boldsymbol{U}_t^T\boldsymbol{G}_t - \boldsymbol{M}_t \\
\frac{d\boldsymbol{V}_t}{dt} &= (\boldsymbol{U}_t^T\boldsymbol{G}_t)^2 - \boldsymbol{V}_t \\
\frac{d\boldsymbol{\Phi}_t}{dt} &= \mathbf{1}_m\mathbf{1}_m^T(\boldsymbol{G}_t - \boldsymbol{U}_t\boldsymbol{U}_t^T\boldsymbol{G}_t)^2 - \boldsymbol{\Phi}_t \\
\boldsymbol{U}_t &= \tau(t, \boldsymbol{G}_t)
\end{aligned}
\tag{41}
$$

where $\tau(\cdot, \cdot)$ can represent a mapping from gradient $\boldsymbol{G}_t$ (and the history of gradients) to the low-rank projection matrix $\boldsymbol{U}_t$.

To show the convergence, we need to design a key quantity called Hamiltonian $\boldsymbol{H}(\boldsymbol{W}_t, \boldsymbol{M}_t, \boldsymbol{V}_t)$ such that it satisfies:

$$
\min_{\boldsymbol{M}_t, \boldsymbol{W}_t, \boldsymbol{V}_t} H(\boldsymbol{W}_t, \boldsymbol{M}_t, \boldsymbol{V}_t) = \min_{\boldsymbol{W}_t}\mathcal{L}(\boldsymbol{W}_t)
\tag{42}
$$

where $\mathcal{L}(\boldsymbol{W}_t)$ is the loss function we want to minimize. This property ensures that the minimization of $H$ is equivalent to the minimization of the actual loss function.

With this property, if $\frac{dH}{dt} \leq 0$, we can show the Hamiltonian is non-increasing, and it will reach to a local optimum (i.e. $\frac{dH}{dt} = 0$) at the end. We can analyze the state $\boldsymbol{W}_t$, $\boldsymbol{M}_t$ and $\boldsymbol{V}_t$ under $\frac{dH}{dt} = 0$, and if this leads to $\boldsymbol{G}_t = 0$, we can prove the weight ODE trajectory will reach to a local optimum of the loss function $\mathcal{L}(\boldsymbol{W}_t)$.

We propose the following Hamiltonian:

$$
H(\boldsymbol{W}_t, \boldsymbol{M}_t, \boldsymbol{V}_t) = \mathcal{L}(\boldsymbol{W}_t) + \frac{1}{2}\left\langle \frac{\boldsymbol{M}_t}{\sqrt{\boldsymbol{V}_t}+\epsilon}, \boldsymbol{M}_t \right\rangle
\tag{43}
$$

where $\langle \boldsymbol{A}, \boldsymbol{B}\rangle = \text{Tr}(\boldsymbol{A}^T\boldsymbol{B})$ represents the matrix inner product.

Before the proof of the convergence, let's first prove a lemma.

**Lemma H.1.** *Assume matrix $\boldsymbol{A}$ has constant columns with positive values, then for any positive semi-definite projector $\boldsymbol{P}$, we have*

$$\left\langle \frac{\boldsymbol{PG}}{\boldsymbol{A}}, \boldsymbol{G} \right\rangle \geq 0. \tag{44}$$

*Proof.*

$$\left\langle \frac{\boldsymbol{PG}}{\boldsymbol{A}}, \boldsymbol{G} \right\rangle$$

$$= \sum_{i=1}^{m} \sum_{k=1}^{n} (\frac{\boldsymbol{PG}}{\boldsymbol{A}})_{ik} \boldsymbol{G}_{ik}$$

$$= \sum_{k=1}^{n} \sum_{i=1}^{m} \frac{(\boldsymbol{PG})_{ik}}{\alpha_k} \boldsymbol{G}_{ik}$$

$$= \sum_{k=1}^{n} \frac{1}{\alpha_k} \boldsymbol{G}_k^T \boldsymbol{P} \boldsymbol{G}_k \geq 0$$

where $\alpha_k$ is the value of the $\text{k}^{\text{th}}$ column of $\boldsymbol{A}$ and $\boldsymbol{G}_k$ is the $\text{k}^{\text{th}}$ column of matrix $\boldsymbol{G}$. The last inequality is from the definition of positive semi-definite projector. $\square$

Next, we prove the convergence of Alice:

**Theorem H.2** (Convergence of Alice). *Assuming the ODEs defined in Eq. (41) satisfies the assumptions of Alice, with the Hamiltonian defined in Eq. (43), it satisfies*

$$\frac{dH}{dt} \leq 0 \tag{45}$$

*and $\frac{dH}{dt} = 0$ implies $\boldsymbol{G}_t = \frac{\partial \mathcal{L}(\boldsymbol{W}_t)}{\partial \boldsymbol{W}_t} = \boldsymbol{0}$.*

*Proof.* First, we write down the form of $\frac{dH}{dt}$:

$$\frac{dH}{dt} = \left\langle \frac{\partial H}{\partial \boldsymbol{W}_t}, \frac{d\boldsymbol{W}_t}{dt} \right\rangle + \left\langle \frac{\partial H}{\partial \boldsymbol{M}_t}, \frac{d\boldsymbol{M}_t}{dt} \right\rangle + \left\langle \frac{\partial H}{\partial \boldsymbol{V}_t}, \frac{d\boldsymbol{V}_t}{dt} \right\rangle$$

$$= -\left\langle \boldsymbol{G}_t, \boldsymbol{U}_t \frac{\boldsymbol{M}_t}{\sqrt{\boldsymbol{V}_t} + \epsilon} + \sqrt{m-r} \frac{\boldsymbol{G}_t - \boldsymbol{U}_t \boldsymbol{U}_t^T \boldsymbol{G}_t}{\sqrt{\boldsymbol{\Phi}_t} + \epsilon} \right\rangle + \left\langle \frac{\boldsymbol{M}_t}{\sqrt{\boldsymbol{V}_t} + \epsilon}, \boldsymbol{U}_t^T \boldsymbol{G}_t - \boldsymbol{M}_t \right\rangle$$

$$\quad - \frac{1}{4} \left\langle \frac{\boldsymbol{M}_t^2}{\sqrt{\boldsymbol{V}_t}(\sqrt{\boldsymbol{V}_t} + \epsilon)^2}, (\boldsymbol{U}_t^T \boldsymbol{G}_t)^2 - \boldsymbol{V}_t \right\rangle$$

$$= -\left\langle \boldsymbol{G}_t, \boldsymbol{U}_t \frac{\boldsymbol{M}_t}{\sqrt{\boldsymbol{V}_t} + \epsilon} \right\rangle - \left\langle \boldsymbol{G}_t, \sqrt{m-r} \frac{\boldsymbol{G}_t - \boldsymbol{U}_t \boldsymbol{U}_t^T \boldsymbol{G}_t}{\sqrt{\boldsymbol{\Phi}_t} + \epsilon} \right\rangle + \left\langle \frac{\boldsymbol{M}_t}{\sqrt{\boldsymbol{V}_t} + \epsilon}, \boldsymbol{U}_t^T \boldsymbol{G}_t \right\rangle$$

$$\quad - \left\langle \frac{\boldsymbol{M}_t}{\sqrt{\boldsymbol{V}_t} + \epsilon}, \boldsymbol{M}_t \right\rangle - \frac{1}{4} \left\langle \frac{\boldsymbol{M}_t^2}{\sqrt{\boldsymbol{V}_t}(\sqrt{\boldsymbol{V}_t} + \epsilon)^2}, (\boldsymbol{U}_t^T \boldsymbol{G}_t)^2 - \boldsymbol{V}_t \right\rangle$$

$$= -\sqrt{m-r} \underbrace{\left\langle \boldsymbol{G}_t, \frac{\boldsymbol{G}_t - \boldsymbol{U}_t \boldsymbol{U}_t^T \boldsymbol{G}_t}{\sqrt{\boldsymbol{\Phi}_t} + \epsilon} \right\rangle}_{\textbf{Term 1}} - \left\langle \frac{\boldsymbol{M}_t}{\sqrt{\boldsymbol{V}_t} + \epsilon}, \boldsymbol{M}_t \right\rangle - \frac{1}{4} \underbrace{\left\langle \frac{\boldsymbol{M}_t^2}{\sqrt{\boldsymbol{V}_t}(\sqrt{\boldsymbol{V}_t} + \epsilon)^2}, (\boldsymbol{U}_t^T \boldsymbol{G}_t)^2 \right\rangle}_{\textbf{Term 2}}$$

$$+ \frac{1}{4} \left\langle \frac{\boldsymbol{M}_t^2}{\sqrt{\boldsymbol{V}_t}(\sqrt{\boldsymbol{V}_t} + \epsilon)^2}, \boldsymbol{V}_t \right\rangle$$

It is clear that **Term 2** is non-negative. So, we first examine the remaining terms (i.e. other terms except Term 1 and 2). Let denote $\frac{M_t}{\sqrt{V_t}+\epsilon} = \Theta$ for simplicity:

$$- \langle \Theta, M_t \rangle + \frac{1}{4} \left\langle \frac{\Theta^2}{\sqrt{V_t}}, V_t \right\rangle$$

$$= \sum_{i=1}^{m} \sum_{k=1}^{n} \frac{1}{4} \Theta_{ik}^2 \sqrt{V_{t,ik}} - \Theta_{ik} M_{ik}$$

$$= \sum_{i=1}^{m} \sum_{k=1}^{n} \frac{1}{4} \frac{M_{ik}^2 \sqrt{V_{t,ik}}}{(\sqrt{V_{t,ik}} + \epsilon)^2} - \frac{M_{ik}^2}{\sqrt{V_{t,ik}} + \epsilon}$$

$$= - \sum_{i=1}^{m} \sum_{k=1}^{n} \frac{1}{4} \frac{\epsilon M_{ik}^2}{(\sqrt{V_t} + \epsilon)^2} \leq 0$$

Now, let's examine **Term 1**:

$$\left\langle G_t, \frac{G_t - U_t U_t^T G_t}{\sqrt{\Phi_t} + \epsilon} \right\rangle = \left\langle G_t, \frac{U_c U_c^T G_t}{\sqrt{\Phi_t} + \epsilon} \right\rangle \geq 0$$

where $U_c \in \mathbb{R}^{m \times (m-r)}$ is the complement basis of $U$ such that $[U_t, U_c]$ will form the complete orthonormal basis. The last inequality is from Theorem H.1 with $P = U_c U_c^T$, $A = \sqrt{\Phi_t} + \epsilon$. It is also trivial to see $\Phi_t$ contains constant columns with non-negative values.

Thus, combining the above, we prove $\frac{dH}{dt} \leq 0$. Next, we consider the behavior when Hamiltonian reaches the local optimum. When $\frac{dH}{dt} = 0$, we can immediately see $M_t = 0$, $U_t^T G_t = 0$ and $\left\langle G_t, \frac{G_t - U_t U_t^T G_t}{\sqrt{\Phi_t} + \epsilon} \right\rangle = 0$.

We examine the last term:

$$\left\langle G_t, \frac{G_t - U_t U_t^T G_t}{\sqrt{\Phi_t} + \epsilon} \right\rangle$$

$$\Rightarrow \sum_{i=1}^{m} \sum_{k=1}^{n} \frac{G_{ik} (U_c U_c^T G)_{ik}}{\sqrt{\Phi_{ik}} + \epsilon} = 0$$

$$\Rightarrow \sum_{i=1}^{m} \sum_{k=1}^{n} \frac{G_{ik} (U_c U_c^T G)_{ik}}{\sqrt{\Phi_{ik}} + \epsilon} + \frac{G_{ik} (U_t U_t^T G)_{ik}}{\sqrt{\Phi_{ik}} + \epsilon} = 0$$

$$\Rightarrow \sum_{i=1}^{m} \sum_{k=1}^{n} \frac{G_{ik} (U_c U_c^T G + U_t^T U_t^T G)_{ik}}{\sqrt{\Phi_{ik}} + \epsilon} = 0$$

$$\Rightarrow \sum_{i=1}^{m} \sum_{k=1}^{n} \frac{G_{ik}^2}{\sqrt{\Phi_{ik}} + \epsilon} = 0$$

$$\Rightarrow G_t = 0$$

Therefore, the weight trajectory $W_t$ will reach to a local optimum of the $\mathcal{L}(W_t)$ when $\frac{dH}{dt} = 0$. $\square$

## I FURTHER DISCUSSION

### I.1 CONNECTIONS BETWEEN DIFFERENT STRUCTURAL ASSUMPTIONS

Here, we will make explicit connections between different structures in terms of their generality.

**OSOAP generalizes Adam** Since the structure behind OSOAP is $\text{Diag}_B(U D_1 U^T, \ldots, U D_n U^T)$, when constraining $U = I_m$, the resulting structural is pure diagonal matrix $\text{Diag}_B(D_1, \ldots, D_n)$. This coincide with the pure diagonal structure behind Adam.

**OSOAP generalizes $S \otimes M$** OSOAP not only extends Adam, but also generalizes the structure considered in Theorem G.4. Consider setting $\boldsymbol{D}_i = S_i \boldsymbol{D}$, then we have

$$\boldsymbol{U}\boldsymbol{D}_i\boldsymbol{U}^T = S_i \underbrace{\boldsymbol{U}\boldsymbol{D}\boldsymbol{U}^T}_{\boldsymbol{M}}$$

Therefore, $\mathrm{Diag_B}(\boldsymbol{U}\boldsymbol{D}_1\boldsymbol{U}^T, \ldots, \boldsymbol{U}\boldsymbol{D}_n\boldsymbol{U}^T) = \mathrm{Diag_B}(S_1\boldsymbol{M}, \ldots, S_n\boldsymbol{M})$. Since the structures of normalization and whitening are special cases of Theorem G.4, OSOAP generalizes these two gradient operators as a consequence.

**SOAP generalizes OSOAP** We can re-write the structural assumption of OSOAP in the Kronecker product format:

$$\mathrm{Diag_B}(\boldsymbol{U}\boldsymbol{D}_1\boldsymbol{U}^T, \ldots, \boldsymbol{U}\boldsymbol{D}_n\boldsymbol{U}^T) = (\boldsymbol{I} \otimes \boldsymbol{U}) \underbrace{\mathrm{Diag_B}(\boldsymbol{D}_1, \ldots, \boldsymbol{D}_n)}_{\tilde{\boldsymbol{D}}}(\boldsymbol{I} \otimes \boldsymbol{U}^T).$$

It is clear that this is a special case of SOAP structure by containing $\boldsymbol{U}_R = \boldsymbol{I}$.

**SOAP generalizes Shampoo** The structure of Shampoo consists of two SPD matrices $\boldsymbol{R}_n$ and $\boldsymbol{L}_m$. If we eigen-decompose those, and let $\boldsymbol{R}_n = \boldsymbol{U}_R\boldsymbol{D}_R\boldsymbol{U}_R^T$ and $\boldsymbol{L}_m = \boldsymbol{U}_L\boldsymbol{D}_L\boldsymbol{U}_L^T$, then the structure of Shampoo can be re-write as

$$\boldsymbol{R}_n^{\frac{1}{2}} \otimes \boldsymbol{L}_m^{\frac{1}{2}}$$
$$=(\boldsymbol{U}_R\sqrt{\boldsymbol{D}_R}\boldsymbol{U}_R^T) \otimes (\boldsymbol{U}_L\sqrt{\boldsymbol{D}_L}\boldsymbol{U}_L^T)$$
$$=(\boldsymbol{U}_R \otimes \boldsymbol{U}_L)\underbrace{(\sqrt{\boldsymbol{D}_R \otimes \boldsymbol{D}_L})}_{\tilde{\boldsymbol{D}}}(\boldsymbol{U}_R^T \otimes \boldsymbol{U}_L^T).$$

This coincides with SOAP's structure when the positive diagonal $\tilde{\boldsymbol{D}}$ can be decomposed based on Kronecker product.

## I.2 MEMORY CONSUMPTION COMPARISON

Apart from the Table 1 provided in the main paper, we also include the memory consumption of more optimizers.

| | Adam | GaLore | Fira | Alice | Alice-0 |
|---|---|---|---|---|---|
| Total | $3mn$ | $mn + 2nr + mr$ | $mn + 2nr + mr$ | $mn + 2nr + mr + r^2 + n$ | $mn + 2nr + mr + n$ |
| Weight | $mn$ | $mn$ | $mn$ | $mn$ | $mn$ |
| First moment | $mn$ | $nr$ | $nr$ | $nr$ | $nr$ |
| Second moment | $mn$ | $nr$ | $nr$ | $nr$ | $nr$ |
| $\boldsymbol{U}$ | N/A | $mr$ | $mr$ | $mr$ | $mr$ |
| $\boldsymbol{C}_t$ | N/A | N/A | N/A | $n$ | $n$ |
| $\widetilde{\boldsymbol{Q}}$ | N/A | N/A | N/A | $r^2$ | N/A |

Table 5: The memory consumption of low-rank optimizers. Here, we assume the weight has a shape $m \times n$ with $m < n$ and $r \ll m$. Note that memory consumption $n$ and $r^2$ is typically very small. For example, let's take the largest possible $n = 30K$ at the output layer. For 1B LLaMA, we select $r = 512$, leading to $r^2 \approx 262K$, which is 8x larger than $n$. However, both are marginal compared to $mr = 5120 \times 512$, which is 10x larger than $r^2$, and 80x larger than $n$.

## I.3 COMPARISON TO PREVIOUS FIM APPROXIMATION

The idea of efficiently approximating the FIM is not novel, and has been extensively studied in the previous work. KFAC (Martens & Grosse, 2015) is a well-known second order optimization algorithm for neural networks based on structural approximation to FIM. In particular, they explicitly express the FIM in terms of the gradient w.r.t this layer's output and the input to this layer. Namely, they directly decompose the gradient $\boldsymbol{G}$ used in our paper, whereas in our paper, we treat $\boldsymbol{G}$ as a whole quantity. In addition, the original KFAC also makes one crucial approximation: $\mathbb{E}[\boldsymbol{A} \otimes \boldsymbol{B}] \approx \mathbb{E}[\boldsymbol{A}] \otimes \mathbb{E}[\boldsymbol{B}]$. They do not show theoretically whether this is a good approximation and under what metric, but

argue in practice this gives good accuracy. Last but not least, the original KFAC considers the FIM for entire layers, and each block (i.e. corresponding to each layer) under their setup is actually the entire FIM we aim to approximate. To be precise, the principle is to structurally approximate the diagonal block of KFAC without decomposing the gradient $G$ using KFAC.

Another more related work is AdaGrad (Duchi et al., 2011). If we only considers layer-wise gradient, the full AdaGrad matrix is the $\sum_{t=1}^{T} \vec{g}_t \vec{g}_t^T$. Although our principle is to approximate FIM, in practice, we use EMA as the approximation of $\mathbb{E}$. Therefore, our principle can also be viwed as a structural approximation to full EMA AdaGrad matrix. Under this view point, there are some previous work on structural approximations. Shampoo (Gupta et al., 2018), was originally proposed as an approximation to full AdaGrad matrix under the online learning setup. However, they do not explicit show under what metric this approximation is based on, and whether it is optimal or not. Later, there is a follow-up work (Morwani et al., 2024), showing that Shampoo is a 1-step power iteration algorithm of optimal Kronecker product approximation to FIM under Frobenius norm (Koroko et al., 2022). In this paper, we further extend the idea of approximating FIM/AdaGrad matrix to more efficient structures, under Frobenius norm.

### I.4 SOLUTIONS TO GENERAL BLOCK DIAGONAL STRUCTURES

Here, we present the solution of the most general block diagonal approximation to FIM, and discuss why this is not practical.

**Proposition I.1** (General block diagonal approximation). *Assume* $\tilde{F} = \mathrm{Diag}(M_1, \dots, M_n)$ *with SPD matrix* $M_i \in \mathbb{R}^{m \times m}$, *then minimizing Eq.* (2) *admits analytic solutions:*

$$M_i^* = \mathbb{E}[g_i g_i^T] \tag{46}$$

*where* $g_i$ *is the* $i^{th}$ *column of* $G$.

*Proof.* This is straightforward by taking the derivative w.r.t. $M_i$. By leveraging Theorem G.1, we have

$$\|\tilde{F} - F\|_F^2$$
$$= \sum_{i=1}^{n} \|M_i\|_F^2 - 2\,\mathrm{Tr}(M_i^T \mathbb{E}[g_i g_i^T])$$

Then, taking derivative w.r.t. $M_i$, we get

$$M_i^* = \mathbb{E}[g_i g_i^T]$$

$\square$

From the above, although it admits analytic solutions, its corresponding practical procedure requires the EMA to estimate $\mathbb{E}[g_i g_i^T] \in \mathbb{R}^{m \times m}$ for all $i = 1, \dots, n$, leading to expensive memory cost $nm^2$. Another issue is that it does not allow efficient computation of the inverse. From the property of block diagonal matrix, to compute $\tilde{F}^{-\frac{1}{2}} \vec{g}$, one needs to invert each $M_i$, incurring $O(m^3)$ computational cost. In total, the computational cost of inverting matrix $\tilde{F}$ is $O(nm^3)$. Due to both memory and computational constraints, this structural assumption does not lead to practical algorithms.

### I.5 CONNECTIONS TO EXISTING OPTIMIZERS

**Lars and Lamb** Lars and Lamb (You et al., 2017; 2019) relies on the normalization operation of the gradients. The main difference is that Lars proposed to normalize the raw gradient, whereas Lamb normalizes the processed gradient from Adam optimizer. They also involves a scaling parameter that is a function of the weight. Compared to the normalization discussed in Sec. 3.3, the main difference is that we use channel-wise normalization with unit column or row norm, whereas Lars/Lamb uses matrix-wise normalization where the norm of the matrix is regularized to be 1. However, if one vectorizes the matrix weight into a vector, and stacks those vectors into a larger matrix. Then, Lamb and Lars can be viewed as a 1-sample approximation to FIM under the structure considered in Sec. 3.3.

**Muon**  Muon (Jordan et al., 2024) performs the whitening operation on the momentum. Sec. D.9 gives a brief introduction. In fact, Muon can be viewed as a special case of Eq. (3) in Theorem 3.3 by considering the following approximation:

$$\mathbb{E}[\boldsymbol{G}\boldsymbol{G}^T] \approx \mathbb{E}[\boldsymbol{G}]\mathbb{E}[\boldsymbol{G}^T]. \tag{47}$$

Thus, the resulting operation becomes the whitening of the $\mathbb{E}[\boldsymbol{G}]$, which is estimated by the momentum in practice. One potential mismatch of the gradient whitening from the original Muon (Jordan et al., 2024) compared to Theorem 3.3 is the lack of the coefficient $\sqrt{n}$, preventing the parameter update to adapt to the shape of the weight matrix. Later, (Liu et al., 2025) found that this term is critical for large-scale LLM training, and add it back to further improve Muon. Their argument is from the empirical intuition to match the update RMS norm of Adam, whereas our Theorem 3.3 proved that this is indispensable from a theoretical perspective.

Another advantage of Muon is its relatively low memory consumption compared to Adam since it only requires the storage of first moments. However, our proposed Alice uses less memory than Muon. Let's consider a weight matrix with shape $m \times m$, and Alice uses a rank $0.25m$, which is consistent to the setup for experiment. Thus, Muon will use $m^2$ memory for state and Alice will only consume $\frac{13}{16}m^2$, 20% less memory than Muon.

**SWAN**  SWAN composes the gradient normalization and whitening operations as the replacement of Adam's first and second moments. Each individual operations can be viewed as a special case of Theorem 3.3. However, Sec. 3.3 does not provide an explanation for composing these two operators. Namely, the whitening operator is estimated using normalized gradients, rather than the raw gradient. We will leave the investigation of operator composition for future work.

**Adapprox**  Adapprox (Zhao et al., 2024b) uses low-rank approximation for Adam's second moments through randomized SVD to boost the memory efficiency. However, its performance will be similar to Adam. In fact, Theorem 4.1 of RACS proves that the converged $\boldsymbol{s}$ and $\boldsymbol{q}$ represents the right and left singular vectors, coinciding with the rank-1 reconstruction. However, the proposed RACS is different from rank-1 Adapprox in three main aspects: (1) Adapprox proposes to use low-rank EMA tracking on $\boldsymbol{G}^2$, whereas we use EMA directly on scaling vectors $\boldsymbol{s}$, $\boldsymbol{q}$. The resulting vectors are no longer singular vectors; (2) we do not have separate scaling apart from norm-growth limiter and user-defined scale, whereas Adapprox uses the scaling from Adafactor; (3) RACS do not use any first moment. Specifically, (1) is an important difference, since for any low-rank reconstruction with rank $r > 1$, it is not guaranteed to be positive, causing numerical issue during square-root inverse scaling. Adapprox adds a manually defined offset to ensure positivity. On the other hand, RACS is guaranteed to have positive $\boldsymbol{s}$, $\boldsymbol{q}$ at each step from Perron-Frobenius theorem. Therefore, RACS is numerically stable.

**Adafactor**  Adafactor (Shazeer & Stern, 2018) is another optimizer that uses rank-1 approximation to Adam's second moment. However, their scaling is derived by minimizing a different norm, compared to Frobenius norm in this paper. Adapprox Zhao et al. (2024b) has demonstrated the advantages of using Frobenius norm compared to Adafactor with a slightly better performance on LLM training.

**RACS v.s. Adafactor**  The update rule of RACS resembles the Adafactor. Indeed, from Eq. (11), when one initializes $\boldsymbol{S}$ and $\boldsymbol{Q}$ as identity and performs one-step fixed point procedure, the resulting scaling matrix is identical to Adafactor. Intuitively, it seems to be surprising if one regards RACS as a "multiple step" version of Adafactor and the performance difference comes solely from running the 1-step or multi-step procedure. Here, we show that extending from 1-step to multi-step can lead to drastically different parameter updates when combining with EMA of $\boldsymbol{S}$ and $\boldsymbol{Q}$. We train a 60M LLaMA model with RACS, and simultaneously calculate the Adafactor parameter update, followed by computing their relative difference as:

$$\text{Rel. Diff} = \frac{1}{N} \sum_{i=1}^{N} \frac{\|\Delta_{i,\text{RACS}} - \Delta_{i,\text{Adafactor}}\|_F}{\|\Delta_{i,\text{Adafactor}}\|_F}. \tag{48}$$

where $\Delta_i$ indicates the update for parameter $i$. Fig. 4 shows the relative difference curve throughout the training. We can observe the relative difference is small, but quickly increases as the training

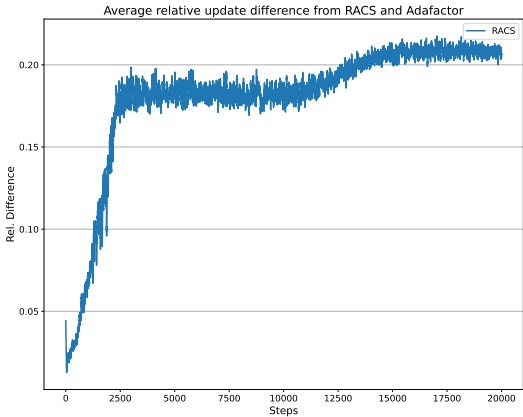

Figure 4: The relative update difference between RACS and Adafactor for training 60M LLaMA with RACS.

progresses. In the end, it settles at $0.2$, representing RACS differs from Adafactor by $20\%$ at each update step. This confirms that the RACS results in different update rule compared to Adafactor.

**AdaDiag and one-sided SOAP**    We acknowledge that there exists two concurrent work: AdaDiag (Nguyen et al., 2025) and one-sided SOAP (Vyas et al., 2024), that are mathematically equivalent to OSOAP. They are derived based on distinct view points. AdaDiag is based on the intuition to transform the gradient covariance matrix so that it is diagonalizable, where this diagonal matrix is estimated using Adam's second moments. One-sided SOAP, a memory-efficient version of SOAP, is proposed to based on intuitions that only one-sided eigenspace is enough for LLM training. On the other hand, OSOAP is derived on the basis of the structured FIM view point. providing a deeper connections to optimizers considered in this paper.

**Apollo**    There is one concurrent work, Apollo (Zhu et al., 2024), that is also based on scaling the raw stochastic gradients for memory-efficient LLM training. It proposed to scale columns **OR** rows of the $G$ through a scaling matrix estimated by similar procedure as Fira Chen et al. (2024a). The main idea is that column or row norm after scaling matches the column or row norm of the gradient from GaLore update. Thus, they require a GaLore procedure to compute this update at each step. Our proposed RACS scales both columns and rows at the same time. The main differences are: (1) the scaling estimation in RACS is different from Apollo; (2) the scaling scheme of RACS is inspired by the generalization of normalization, providing theoretical support. They enjoy a similar memory consumption (i.e. $mn + 2n + 2$ of Apollo compared to $mn + m + n + 1$ of RACS).

**Fira**    Fira Chen et al. (2024a) was proposed as an improvement towards GaLore, which modifies the low-rank update to full-rank one by adding a compensation term. Their idea is similar to the compensation used in Alice. The main differences is that our proposed compensation is inspired by approximating FIM and has the theoretical foundation on its optimality. Also, the compensation strategy is different to Fira. We have conduced the ablation study in Sec. 6 to show the advantage of our approach.

**GaLore**    Comparing the Alice procedure to GaLore (Algorithm 8), we can clearly see that GaLore is a special case of Alice by disabling tracking, switching and compensation. These three steps flows naturally from the structural approximation view point. This shows the advantage of this view point compared to ad-hoc choices when designing new efficient optimizers.

**SARA**    Sara (Zhang et al., 2025) is a concurrent work that aims to improve the quality of the projection by using importance sampling of singular vectors. This shares the similar intuition with our subspace switching mechanism (Algorithm 1), although the techniques differ: Sara relies on importance sampling of singular vectors, while Alice mixes random sampling of less important directions with leading directions. We want to emphasize that subspace switching is only one of

the three components in Alice. Based on the ablation study in Table 4, the theoretically optimal compensation step offers the largest improvement in performance.

**LDAdam**   LDAdam (Robert et al., 2024) is a recently proposed low-rank optimizer based on Adam with error-feedback mechanism. The goal of error-feedback is similar to the compensation step in Alice, aiming at reducing the information loss from the low-rank projection. However, they tackle this problem in a different manner. Error-feedback mechanism modifies the gradient by adding back the information loss from the previous steps. However, LDAdam update still relies on the low-rank projection, resulting in low-rank update in the end. Additionally, there is no theoretical guarantees on the optimality of error-feedback mechanism. On the contrary, Alice leverages the optimal compensation step, which directly add the information loss back. Therefore, the parameter update at each step of Alice is guaranteed to be full-rank.

## I.6   DISCUSSION OF LOW-RANK EXTENSION FRAMEWORK

First, we derive how to decompose the full-rank update of OSOAP into low-rank update and its residuals in the main text. We assume $\tilde{U} = [U, U_c]$, where $U, U_c$ are defined in Sec. 5.3.

$$
\begin{aligned}
\mathrm{Mat}(\tilde{F}^{-\frac{1}{2}}\vec{g}) &= \tilde{U}\frac{\tilde{U}^T G}{\sqrt{\mathbb{E}[(\tilde{U}^T G)^2]}} \\
&= [U, U_c]\frac{\begin{bmatrix} U^T \\ U_c^T \end{bmatrix} G}{\sqrt{\mathbb{E}\left[\left(\begin{bmatrix} U^T \\ U_c^T \end{bmatrix} G\right)^2\right]}} \\
&= [U, U_c]\frac{\begin{bmatrix} U^T G \\ U_c^T G \end{bmatrix}}{\sqrt{\mathbb{E}\left[\left(\begin{bmatrix} U^T G \\ U_c^T G \end{bmatrix}\right)^2\right]}} \\
&= [U, U_c]\begin{bmatrix} \frac{U^T G}{\sqrt{\mathbb{E}[(U^T G)^2]}} \\ \frac{U_c^T G}{\sqrt{\mathbb{E}[(U_c^T G)^2]}} \end{bmatrix} \\
&= U\frac{U^T G}{\sqrt{\mathbb{E}[(U^T G)^2]}} + U_c\frac{U_c^T G}{\sqrt{\mathbb{E}[(U_c^T G)^2]}}
\end{aligned}
$$

**Moore-Penrose inverse**   In Eq. (13), we define $\tilde{F}^{-\frac{1}{2}}$ using the pseudo-inverse since each block in $\tilde{F}_c$ is low-rank and not invertible. To be precise, for any block $U_c D_{ci} U_c^T$, its pseudo-inverse can be easily verified as $U_c D_{ci}^{-1} U_c^T$ by checking the definition. Similar to typical inverse, for any block diagonal $\tilde{F}_c$, the pseudo-inverse is equivalent to applying pseudo-inverse of each blocks. The square-root pseudo inverse can be defined through a similar way.

Similarly, for the proposed compensation, we can easily verify its vectorized format (ignoring the subscript $t$ for simiplicity)

$$
\begin{aligned}
\mathrm{Vec}(U_c U_c^T G S) &= (S \otimes U_c U_c^T)\vec{g} \\
&= (S^{-2} \otimes U_c U_c^T)^{-\frac{1}{2}}\vec{g}
\end{aligned}
$$

where the $-\frac{1}{2}$ is the pseudo-inverse as the above. The second inequality can be easily verified by the following facts: (1) the square root pseudo inverse of $U_c U_c^T$ is itself; (2) the square root pseudo-inverse of block-diagonal matrix is the pseudo-inverse of each individual block.

## J   EXPERIMENT DETAILS

In this section, we will include the detailed setup, hyperparameters and additional experiment results.

## K   IMPLEMENTATION DETAILS OF BASELINES

**GaLore**   We leveraged the official GaLore package released by the author (`https://github.com/jiaweizzhao/GaLore`).

**Fira**   Since the main difference compared to GaLore is the additional compensation term, we follow the implementation of the official Fira repo (`https://github.com/xichen-fy/Fira`) and add the compensation to GaLore.

### K.1   EXPERIMENT SETUP FOR PRETRAINING LLaMA

We use context length of 256 and batch size of 128 with 4 gradient accumulations, apart from 60M and 1.3B (batch size 256 with 2 gradient accumulations). RACS and Alice are applied to all linear modules of attention and MLPs, others are optimized with Adam. We use the first $10\%$ of the total training steps as the warm-up, followed by a cosine learning rate decay to $10\%$ of the original learning rate. For hyperparameters, we perform a grid search on our proposed RACS, Alice as well as GaLore, Fira and full-rank Adam. For other methods, we directly cite their results in (Zhu et al., 2024). For Adam, we use 0.9 and 0.999 for $\beta_1$ and $\beta_2$, and enable the bias correction without weight decay. Table 6 summarizes the hyperparameters used for both GaLore and Fira. Table 7 summarizes the hyperparameters for Adam optimizer. Table 8 and Table 10 summarizes the hyperparameters used for RACS and Alice, respectively. In addition, for all methods with Fira limiter, we use threshold $\gamma = 1.01$ as suggested in Chen et al. (2024a). For RACS, we use Adam to train the last layer of LLaMA, following the same setup as Apollo for fair comparison. In summary, Alice, GaLore and Fira do not use Adam to train the last layer to fully test the capabilities of low-rank methods, whereas Apollo, RACS and Adam utilize Adam for last layer. The total training steps for 60M, 130M, 350M and 1.3B are 10K, 20K, 60K and 100K, respectively. These correspond to 1.1B, 2.6B, 7.8B and 13.1B training tokens, summarized in Table 9. All experiments are conduced on NVIDIA A100 GPUs.

Table 6: The hyperparameters for GaLore and Fira

|      | learning rate | update scale | rank | update interval |
|------|---------------|--------------|------|-----------------|
| 60M  | 0.02          | 0.3          | 128  | 200             |
| 130M | 0.02          | 0.3          | 256  | 200             |
| 350M | 0.02          | 0.3          | 256  | 200             |
| 1.3B | 0.01          | 0.25         | 512  | 200             |

Table 7: The hyperparameters used for Adam optimizer.

|      | learning rate     | $\beta_1$ | $\beta_2$ | correct bias |
|------|-------------------|-----------|-----------|--------------|
| 60M  | 0.001             | 0.9       | 0.999     | True         |
| 130M | 0.001             | 0.9       | 0.999     | True         |
| 350M | 0.001             | 0.9       | 0.999     | True         |
| 1.3B | $7 \times 10^{-4}$ | 0.9       | 0.999     | True         |

Table 8: The hyperparameters for RACS.

|      | learning rate | $\beta$ | scale $\alpha$ |
|------|---------------|---------|----------------|
| 60M  | 0.02          | 0.9     | 0.05           |
| 130M | 0.02          | 0.9     | 0.05           |
| 350M | 0.02          | 0.9     | 0.05           |
| 1.3B | 0.02          | 0.9     | 0.02           |

Table 9: Model architectures and training steps

|      | Hidden | Intermediate | Heads | Layers | Steps | Data amount |
|------|--------|--------------|-------|--------|-------|-------------|
| 60M  | 512    | 1376         | 8     | 8      | 10K   | 1.3B        |
| 130M | 768    | 2048         | 12    | 12     | 20K   | 2.6B        |
| 350M | 1024   | 2736         | 16    | 24     | 60K   | 7.8B        |
| 1.3B | 4096   | 5461         | 24    | 32     | 100K  | 13.1B       |

Table 10: The hyperparmeters for Alice optimizer

|      | learning rate | scale $\alpha$ | compensation scale $\alpha_c$ | $\beta_1$ | $\beta_2$ | $\beta_3$ | update interval $K$ | rank $r$ | leading basis number $l$ |
|------|---------------|----------------|-------------------------------|-----------|-----------|-----------|---------------------|----------|--------------------------|
| 60M  | 0.02          | 0.3            | 0.4                           | 0.9       | 0.9       | 0.999     | 200                 | 128      | 40                       |
| 130M | 0.02          | 0.3            | 0.4                           | 0.9       | 0.9       | 0.999     | 200                 | 256      | 40                       |
| 350M | 0.02          | 0.3            | 0.4                           | 0.9       | 0.9       | 0.999     | 200                 | 256      | 40                       |
| 1.3B | 0.02          | 0.25           | 0.2                           | 0.9       | 0.9       | 0.999     | 200                 | 512      | 160                      |

## K.2 SETUP OF 1B V.S. 7B

We mainly follow the setup as (Zhu et al., 2024). For 1B model, we use 16 per-device batch size with 8xA100s and 4 gradient accumulations, which is 512 batch size in total, same as 7B model. For Alice, we use learning rate 0.02, scale $\alpha = 0.3$ and compensation scale $\alpha_c = 0.2$ with rank $r = 512$, leading basis number $l = 160$. We also use full-rank Adam to train the last layer for better performance. For memory estimation, we assume the use of BF16 format. For 8-bit optimizers, we assume weights are stored in BF16, but optimizer states use FP8. GaLore uses $r = 1024$ for 7B model.

## K.3 MEMORY ESTIMATION

Following the setup of Zhao et al. (2024a), we provide the estimated GPU memory for each optimizers due to the difficulty of directly measuring their practical memory consumption without considering the activations, internal storage for gradient, etc. We assume BF16 format, which consuming 2 Bytes per element. For example, 1273M parameters are optimized by Alice and 65M parameters are optimized by Adam for 1B model with rank $512$. Therefore, Alice part will consume 3.86GB and Adam will consume 0.37GB, summing to $4.42$GB for Alice optimizer.

We also report the actual memory footprint with BF16 format. We use token batch size of 25, following the same setup as Zhao et al. (2024a). "-layerwise" represents layer-wise training where we only store the gradient of the current back-propagated layer.

## K.4 THROUGHPUT ESTIMATION

We report both the actual throughput and effective throughput. The effective throughput of a target optimizer compared to a reference optimizer is defined as the ratio of training tokens consumed from the target optimizer w.r.t total time consumption of reference optimizer when reaching the same evaluation loss of the reference optimizer at the end of training. Compared to the standard throughput, this considers the speed-up effect.

## K.5 ADDITIONAL PRETRAIN RESULTS

Fig. 5 presents additional training curves with 60M, 130M and 350M models sizes. For all cases, the proposed Alice and RACS outperforms the baselines with noticable margin, and achieves clear speed-ups compared to full-rank Adam.

## K.6 RESULTS FOR ABLATION STUDIES

For all ablations, we consider 130M LLaMA model.

**Setup: ablation of tracking**  We disable the compensation step, and verify the effect of low-rank tracking under our proposed switch strategy or purely replying on EVD. The other hyperparameters follow the pre-training setup as Sec. K.1.

**Setup: switch strategy**  For this setup, we enable low-rank tracking. Apart from Gaussian, we use 40 as the leading basis number. The Gaussian distribution is a standard isotropic Gaussian with zero mean and unit variance.

**Setup: compensation strategy**  We enable the tracking and switching, but with different compensation terms. For Fira, we directly leverage the compensation proposed in Chen et al. (2024a). Fira+ modifies original Fira by the following two steps: (1) rescale the Fira compensation to have the same $l_2$ norm as the Alice low-rank update; (2) multiply this compensation by a scale parameter like $\alpha_c$ in Algorithm 4. We found this empirical trick boosts the performance of Fira.

**Effects of last layer**  One crucial setup difference during evaluation for low-rank methods is whether the last layer is trained by full-rank Adam or not. Most previous work train the last layer, arguably one of the most important layer (Zhao et al., 2024c), using full-rank Adam. This effectively reduces

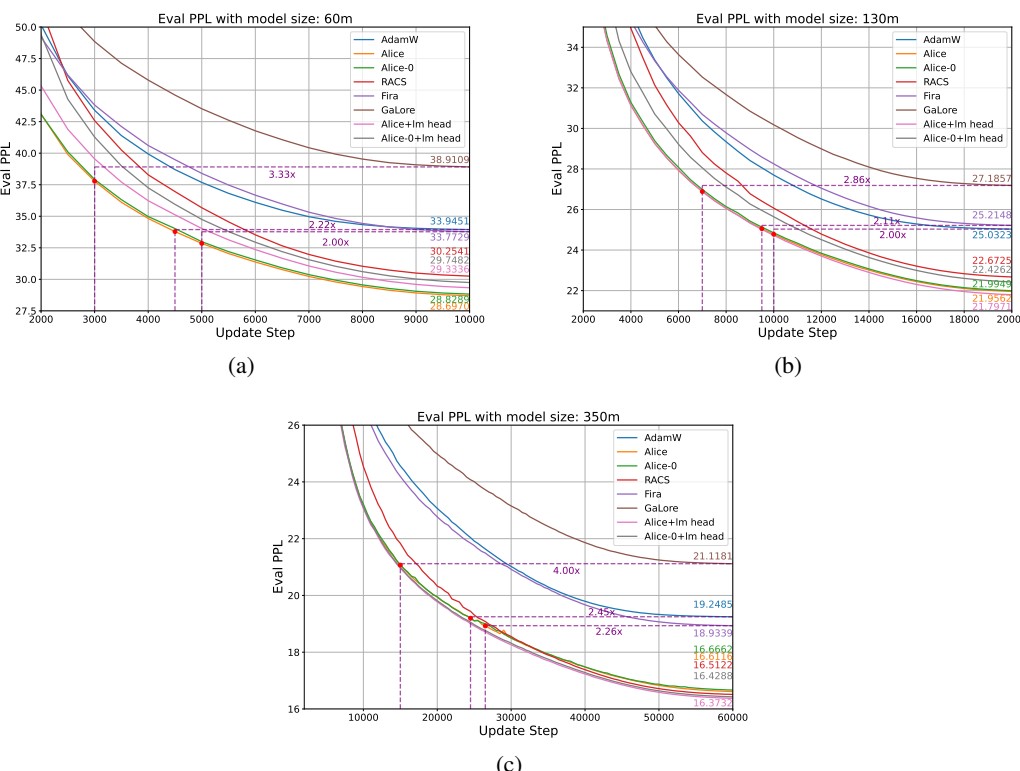

Figure 5: Additional LLaMA C4 pretrain performance curve. (a), (b) and (c) represents the 60M, 130M and 350M, respectively. "+lm head" represents that the last layer of LLaMA is trained by full-rank Adam.

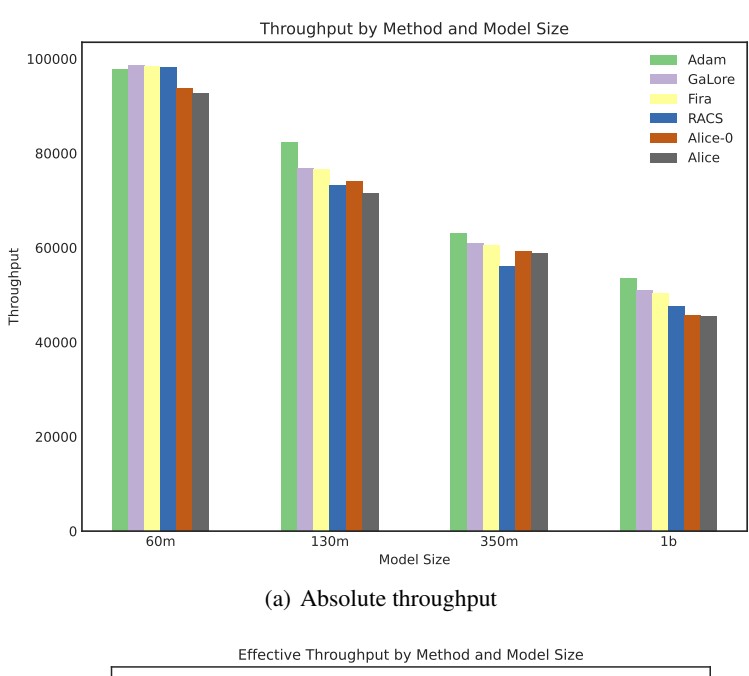

(a) Absolute throughput

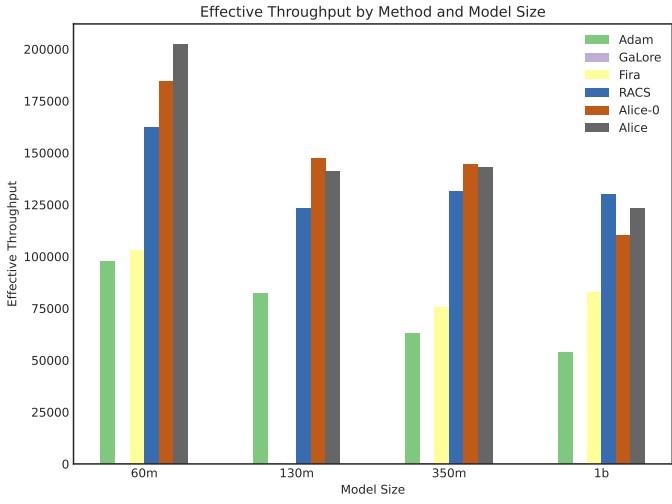

(b) Effective throughput

Figure 6: Throughput of various methods. (a) this reports the absolute throughput, representing the number of training token processed per second. (b) the effective throughput using Adam as the reference optimizer. This represents the absolute throughput adjusted by the speed-up factor. The effective throughput of GaLore is $0$ since it under-performs the Adam.

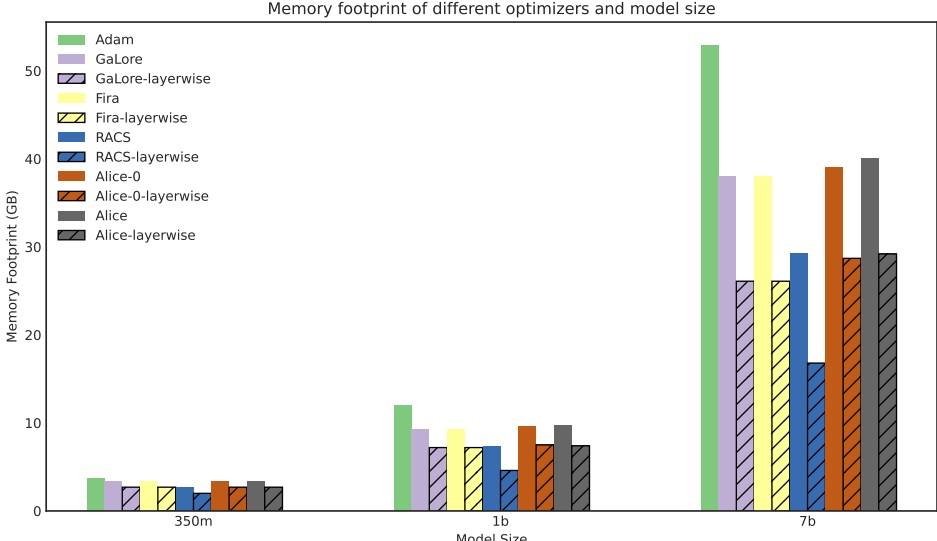

Figure 7: The memory footprint of various optimizers. We use token batch size of 1 following the same setup as Zhao et al. (2024a) under BF16 format. The suffix "layerwise" represents the memory consumption when enabling layerwise training so that only the gradient of the current layer is stored.

the performance gap compared to full-rank method, and does not reveal their true capabilities. We investigate the effect of the last layer to Alice compared to GaLore. From the Table 2, we can see that for all model sizes, GaLore and Fira are greatly affected by this effect. Training last layer with full-rank Adam will boost their performance significantly. On the other hand, Alice is less impacted with marginally worse performance. For example, Fig. 8(d) shows the training curve comparison with 130M model. When the rank is sufficiently large (i.e. 60M model), Fig. 5(a) shows that using Adam to train the last layer even decreases the performance. These serve as evidences that Alice is a better optimizer than GaLore and Fira.

**Effect of EMA in RACS** The only internal states inside RACS are the EMA tracking of two vectors $s$ and $q$. We investigate the importance of EMA scheme. From Fig. 8(e), RACS without EMA performs much worse than RACS, suggesting the EMA is necessary for satisfactory performance of RACS.

**Hyperparameter sensitivity analysis: Adam** We consider 130M Llama model, and the other setup follows the same setting as the pretraining experiments. For Adam, we consider a range of learning rate from $3 \times 10^{-4}$ to $3 \times 10^{-3}$, $\beta_1$ from $[0.9, 0.8, 0.6, 0.4]$ and $\beta_2$ from $[0.999, 0.95, 0.8, 0.6, 0.4]$. The total budget is 120 runs.

For 130M model, we found that the best performing hyperparameters for Adam are learning rate 0.001 with betas $\beta_1 = 0.8, \beta_2 = 0.95$. This achieves a ppl of 22.89. Surprisingly, apart from RACS (ppl 22.67) and Alice (ppl 21.95), fine-tuned Adam outperforms or is on par with all other candidates, despite some of them claims to achieve better convergence than Adam. One hypothesis is that Adam is sensitive to beta parameter, and most existing literature may use the untuned beta (i.e. $\beta_1 = 0.9$, $\beta_2 = 0.999$) for comparison. Even compared to fine-tuned Adam, Alice still achieves $1.4x$ speed-up. Table 11 reports a Adam performance with a selected set of hyperparameters. We observe that Adam's performance can drop significantly even by slightly changing the $\beta_2$. For example, with $\beta_1 = 0.8$, changing $\beta_2$ from 0.95 to 0.8 decreases the ppl from 22.89 to 24.51. Due to the expensive nature of each run, fine-tuned Adam is not practical.

**Hyperparameter sensitivity analysis: Alice** We follow the same setup as the sensitivity analysis of Adam. We consider the following hyperparameters of Alice: (1) compensation scale $\alpha_c$ from 0.1 to 0.9; (2) update interval $K$ from $[50, 100, 200, 300, 400]$; (3) rank $r$ from $[16, 32, 64, 128, 256]$;

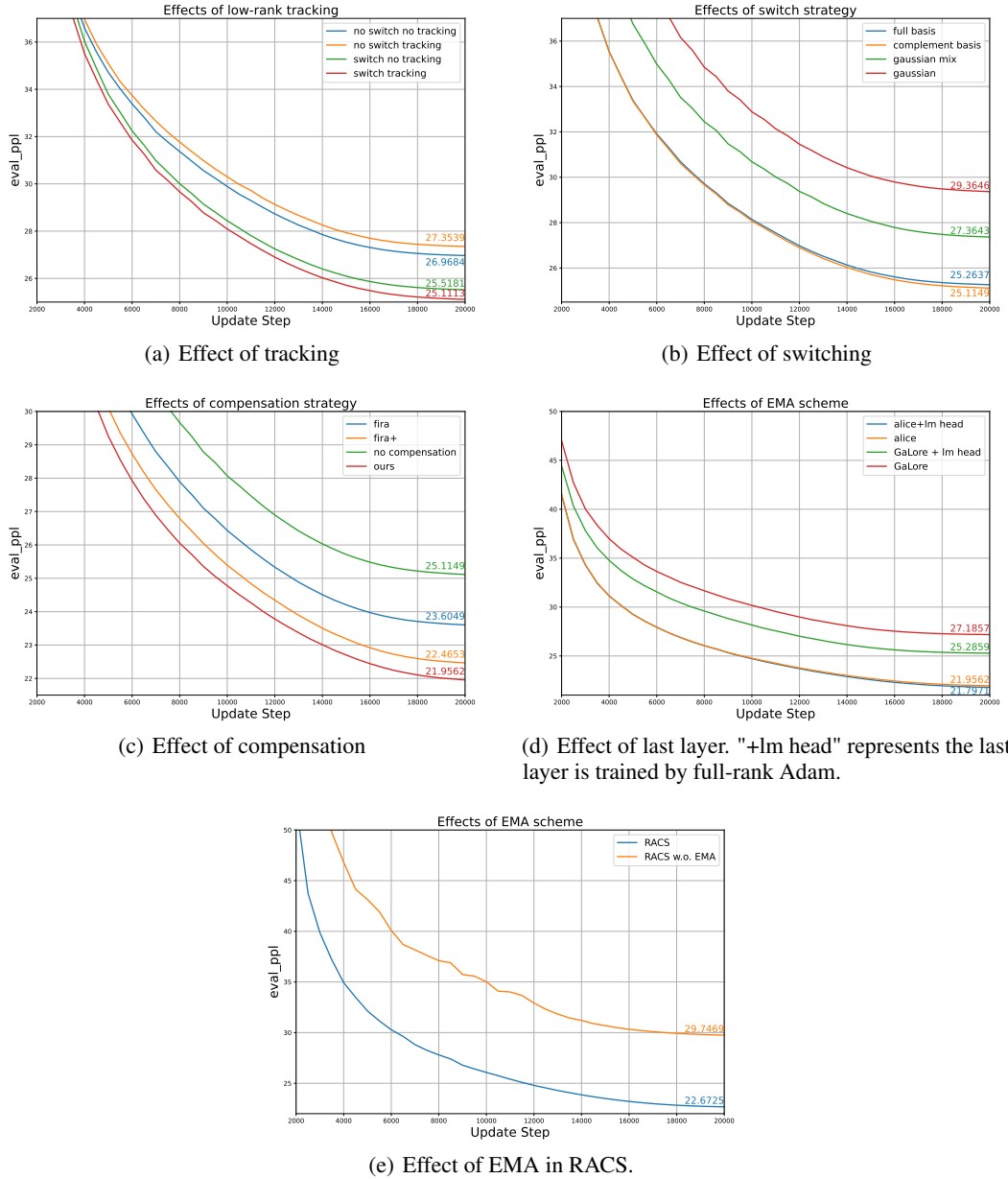

Figure 8: The pre-training curve to verify the effectiveness of the design choice. We consider 130M model size.

| Betas | (0.4, 0.6) | (0.4, 0.95) | (0.6 ,0.8) | (0.6, 0.95) | (0.8 ,0.8) | (0.8, 0.95) | (0.9,0.8) | (0.9,0.95) |
|-------|-----------|-------------|------------|-------------|------------|-------------|-----------|------------|
| Ppl | 25.18 | 23.44 | 23.79 | 23.05 | 24.51 | 22.89 | 24.90 | 23.36 |

Table 11: The validation ppl. of Adam optimizer with different choice of betas. Due to the large number of runs (120 in total), we only report Adam with lr 0.001 and some selected betas combinations. This is enough to prove that Adam is sensitive to hyperparameter selection.

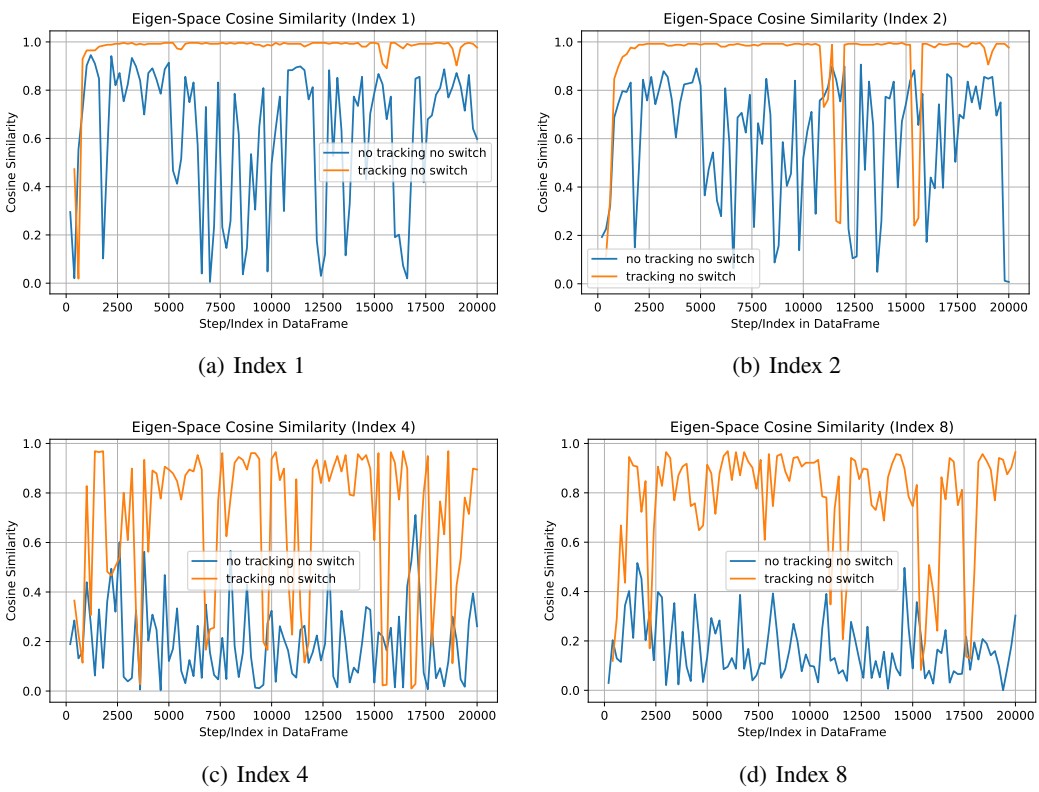

Figure 9: The cosine similarity between eigenvectors per 200 steps. Since the update interval of subspace is 200 steps, this essentially compare the similarity before and after updating the projection $U$ with a certain index. We always arrange the eigenvectors in a descending order based on the eigenvalues.

and (4) leading basis number $l$ from $[5, 10, 20, 40, 80, 160]$. Due to the combinatorial nature, we only change one particular hyperparameter of Alice at a time and fix the others as Table 10.

Tables 12 to 15 shows the performance of Alice under different choice of hyperparameters. We observed that Alice is not sensitive to the choice of hyperparameters. Among those runs (25 in total), 23 of them outperform **fine-tuned** Adam, proving that the advantage of Alice comes from its principled design, rather than carefully selected hyperparameters.

| Compensation scale $\alpha_c$ | 0.1 | 0.2 | 0.3 | 0.4 | 0.5 | 0.6 | 0.7 | 0.8 | 0.9 |
|---|---|---|---|---|---|---|---|---|---|
| Ppl. | 23.04 | 22.29 | 22.08 | 21.95 | 22.02 | 22.05 | 22.12 | 22.19 | 22.27 |

Table 12: The performance of Alice with different compensation scale $\alpha_c$.

| Update interval $K$ | 50 | 100 | 200 | 300 | 400 |
|---|---|---|---|---|---|
| Ppl. | 23.01 | 22.09 | 21.95 | 21.99 | 22.09 |

Table 13: The performance of Alice with different choice of update interval $K$.

| Rank $r$ | 16 | 32 | 64 | 128 | 256 |
|---|---|---|---|---|---|
| Ppl. | 22.20 | 22.03 | 22.00 | 22.07 | 21.95 |

Table 14: The performance of Alice with different choice of rank $r$.

| Leading basis number $l$ | 5 | 10 | 20 | 40 | 80 | 160 |
|---|---|---|---|---|---|---|
| Ppl. | 22.53 | 22.33 | 22.21 | 21.95 | 21.95 | 21.90 |

Table 15: The performance of Alice with different choice of leading basis number $l$.

### K.7 GLUE FINETUNE PERFORMANCE

Although the primary goal of this paper is to design the efficient optimizer for pre-training LLMs, we also conduct the preliminary finetune experiments with Alice using GLUE (Wang, 2018).

**Setup** We follow the same setup as (Zhao et al., 2024a) with RoBERTa (Liu, 2019) model. We select the rank $r = 8$ for both GaLore and Alice. Tables 16 and 17 reports the hyperparameters for GaLore and Alice. By default, we report prediction accuracy as the metric for most of the dataset. For CoLA, we use Matthews_Correlation; and for STS-B, we report Pearson Correlation.

Table 18 reports the GLUE finetune performance of Adam and Alice. It is clear that Alice consistently outperforms the GaLore across datasets. This is expected since each update step of Alice is full-rank compared to the low-rank nature of GaLore. Also, GaLore is a special case of Alice.

| | MNLI | SST-2 | MRPC | CoLA | QNLI | QQP | RTE | STS-B |
|---|---|---|---|---|---|---|---|---|
| Batch size | 16 | 16 | 16 | 32 | 16 | 16 | 16 | 16 |
| Epochs | 30 | 30 | 30 | 30 | 30 | 30 | 30 | 30 |
| Learning rate | 1e-5 | 2e-5 | 2e-5 | 1e-5 | 1e-5 | 2e-5 | 2e-5 | 3e-5 |
| Scale $\alpha$ | | | | 4 | | | | |
| Sequence length | | | | 512 | | | | |
| Update interval | | | | 500 | | | | |

Table 16: Galore hyperparameters for GLUE finetune experiments.

|  | MNLI | SST-2 | MRPC | CoLA | QNLI | QQP | RTE | STS-B |
|---|---|---|---|---|---|---|---|---|
| Batch size | 16 | 16 | 16 | 32 | 16 | 16 | 16 | 16 |
| Epochs | 30 | 30 | 30 | 30 | 30 | 30 | 30 | 30 |
| Learning rate | 1e-5 | 1e-5 | 2e-5 | 2e-5 | 1e-5 | 1e-5 | 1e-5 | 2e-5 |
| Scale $\alpha$ |  |  |  | 4 |  |  |  |  |
| Sequence length |  |  |  | 512 |  |  |  |  |
| Update interval |  |  |  | 500 |  |  |  |  |
| Leading basis number | 8 | 3 | 8 | 3 | 6 | 6 | 3 | 6 |
| Compensation scale | 0.1 | 0.07 | 0.1 | 0.05 | 0.1 | 0.1 | 0.3 | 0.3 |

Table 17: Alice hyperparameters for GLUE finetune experiments.

|  | MNLI | SST-2 | MRPC | CoLA | QNLI | QQP | RTE | STS-B | Mean |
|---|---|---|---|---|---|---|---|---|---|
| GaLore | 87.31 | 94.72 | 89.71 | 60.34 | 92.75 | 91.13 | 79.78 | 91.11 | 85.86 |
| Alice | **87.45** | **95.18** | **90.69** | **65.04** | **92.90** | **91.45** | **80.51** | **91.43** | **86.83** |

Table 18: The GLUE finetune performance of Adam and Alice.

