# OpenReview forum: "Towards Efficient Optimizer Design for LLM via Structured Fisher Approximation with a Low-Rank Extension"
_ICLR.cc/2026/Conference — ICLR 2026 Poster_

### Official Review · Reviewer_1U1o · 2025-10-25

**Soundness:** 4
**Presentation:** 3
**Contribution:** 4
**Rating:** 8
**Confidence:** 4

**Summary:**

The paper advocates a structured Fisher Information Matrix (FIM) approximation as a unifying lens for designing efficient optimizers for LLM training. It shows that a range of optimizers/gradient operators—including Adam, Shampoo, normalization/whitening, and SOAP/AdaDiag++—arise as solutions to a Frobenius-norm projection of the empirical FIM onto different structure families. On LLaMA pretraining, RACS/Alice outperform Adam and strong memory-efficient baselines; Alice achieves >2× step-speedup over Adam with comparable memory to GaLore, and RACS delivers strong results with ~SGD-like memory. The contribution is mainly a unifying viewpoint and design methodology, not a definitive theory of LLM optimization.

**Strengths:**

1. This paper provides unfied and clear perspective: casting popular methods as structured FIM projections.

2. On C4 pretraining, Alice yields 2.22×–2.82× step-speedup vs. Adam with strong effective throughput; RACS beats multiple memory-efficient baselines (GaLore, Fira, Apollo, Muon) under comparable or lower memory.

3. The writing is good and easy to follow.

4. Solid theorem analysis is provided.

5. Both pretraining and finetuning experiments are provided, showing the superiority of the proposed method under different cases.

**Weaknesses:**

Overall, I believe this is a good paper, some small cons:

1. The font size in some tables and figures are too small for potential readers, for example, Figure 1 and Table 1. It would be better if the font size be larger.

2. The authors may consider adding illustrative figures to accompany the complex mathematical formulations. Visual aids would help readers grasp the core ideas and intuitions behind the proposed methods more intuitively and easily

3. The current experiments are all conducted on models with fewer than 1B parameters. It would be better to see if the proposed method can scale up to 7B model pre-training, as Galore and Fira do. Alternatively, if constrained by computational resources, the authors may consider offering more in-depth discussions or analyses to compensate for the lack of large-scale experiments.

**Questions:**

see weaknesses

---

> ### Author Response · Authors · 2025-11-21
> **Author Response**
>
> We greatly appreciated the reviewer's effort to provide valuable suggestions. Here are our responses, and hope they can clarify the confusions.
>
> 1. `experiments are conducted with model size <1B`
> - Thank you for the comment. We would like to clarify that Table 2 does include a 1.3B–parameter model, and our method shows substantial improvement over the baselines at this scale.
> - Regarding larger models (e.g., 7B), we are currently constrained by the computational resources available to us. Scaling Alice to multi-billion–parameter models requires non-trivial FSDP or tensor-parallel support, because Alice/RACS's updates involves matrix operations that demand more sophisticated distributed implementations than elementwise methods like AdamW.
> - As a partial indication of scaling behavior, we include a preliminary 1B vs. 7B comparison in table 3 with slight overtraining setup, showing that both Alice and RACS achieve better performance on the 1B model than the 7B model trained with existing memory-efficient optimizers. While this is not a direct 7B pretraining experiment, it does provide encouraging evidence that the proposed optimizer has the potential to scale favorably.
>
> 2. `fontsize; illustration`
> Thanks for your suggestions. We will enlarge the font size and add more illustrations upon acceptance when there is one extra page.
>
> We hope the clarifications above address your concerns and assist your assessment, and we are happy to elaborate further if needed.

---

> > ### Comment · Reviewer_1U1o · 2025-11-27
> >
> > Besides, I believe the authors can still refine the illustrations at this stage, as the policy allows for one additional page during the rebuttal phase.

---

> > > ### Author Response · Authors · 2025-11-28
> > > **Author's response**
> > >
> > > Thanks for the reminder, we haven't noticed that we have one more page for revision. We have updated the paper and add Figure 1 and 2 to summarize the relations between different optimizers and the procedure of our proposed low-rank extension framework.
> > >
> > > Thanks again for your valuable feedback and suggestions

---

> ### Comment · Reviewer_1U1o · 2025-11-21
>
> Thanks for the rebuttal. Some of my concerns have been addressed.
>
> While additional pre-training experiments at the 7B scale (as in Galore, Fira) would further strengthen the paper (remaining concerns), I believe their absence does not affect the major contributions of the work. I thus keep my final rating at 8.

---

### Official Review · Reviewer_phAX · 2025-10-27

**Soundness:** 3
**Presentation:** 3
**Contribution:** 3
**Rating:** 6
**Confidence:** 2

**Summary:**

This paper proposes a unifying framework for designing memory-efficient LLM optimizers through structured Fisher Information Matrix (FIM) approximation. The authors show that existing optimizers (Adam, Shampoo, gradient normalization, etc.) can be viewed as special cases of FIM approximation with different structural constraints, and propose two design principles: (1) selecting structures that balance generality and efficiency, and (2) applying a low-rank extension framework. Based on these principles, they introduce two novel optimizers—RACS (achieving SGD-like memory with row-column scaling) and Alice (a low-rank variant of generalized Adam with tracking, switching, and compensation mechanisms)—which demonstrate over 2× faster convergence than Adam on LLaMA pre-training tasks while maintaining low memory overhead.

**Strengths:**

1. Originality: The unifying FIM approximation framework is highly original, elegantly connecting diverse optimizers (Adam, Shampoo, normalization, Muon) under a principled theoretical lens. The low-rank extension framework with subspace switching and optimal compensation represents creative innovation beyond existing ad-hoc approaches. The insight linking GaLore to Gadam provides valuable new perspective on existing methods.

2. Quality: The mathematical formulations are rigorous, with analytical solutions derived for various structural constraints. Experiments are comprehensive and well-designed, spanning multiple model sizes (60M-1.3B) and demonstrating consistent, substantial improvements. The convergence analysis under continuous-time setup adds theoretical grounding.

3. Clarity: The paper is well-structured with logical progression from framework exposition to practical optimizer design. The step-by-step derivation of existing optimizers as special cases effectively illustrates the framework's unifying power. The two design principles provide clear, actionable guidance.

4. Significance: High practical impact—Alice achieves over 2× convergence speedup compared to Adam with minimal memory overhead, addressing critical LLM training efficiency challenges. The framework provides valuable conceptual foundation for systematic optimizer design, moving the field beyond heuristic approaches toward principled methodology. The work opens promising research directions for incorporating model structure and task-specific information into optimizer design.

**Weaknesses:**

1. The convergence analysis is limited to continuous-time setup (Theorem H.2), which is a gap given that practical optimizers operate in discrete time. The one-step refinement approximation in Theorem 3.4 for Gadam lacks justification for why this approximation is reasonable or how the error propagates.

2. All pretraining uses only LLaMA on C4. Section K.7 adds GLUE on RoBERTa, but this is insufficient. Generalization to other architectures (GPT, T5, Mamba) and domains (code, multilingual) is unclear.

3. Table 2 shows Alice has 15% throughput drop on 1B models (45,523 vs 53,588 tokens/sec), but lacks breakdown of where this overhead originates. Specifically, no profiling data separates the cost of: (1) EVD approximation via subspace iteration (O(m²r) every K steps), (2) subspace switching with QR decomposition, and (3) per-step compensation computation. Without this profiling, practitioners cannot determine which components to optimize (e.g., increasing K to amortize EVD cost, or simplifying Eq. 14 to reduce compensation overhead) or assess when the 15% penalty is acceptable given 2× step-wise speedup.

**Questions:**

Q1. Can you provide convergence analysis for Alice under discrete-time dynamics, including convergence rates in terms of learning rate η, rank r, and switching frequency K, and how the one-step refinement approximation error affects final convergence?

Q2. Can you provide detailed profiling showing time spent on EVD approximation, subspace switching, and compensation computation, and demonstrate how throughput scales with update interval K to help practitioners optimize the 15% throughput penalty?

Q3. Why is random uniform sampling from the complement space optimal in Algorithm 1, and have you compared against informed strategies like gradient-based importance sampling or residual-guided selection using Σ_t from Theorem G.8?

---

> ### Author Response · Authors · 2025-11-20
> **Author Response**
>
> We greatly appreciated the reviewer's effort to provide valuable suggestions. Here are our responses, and hope they can clarify the confusions.
>
> 1. `convergence analysis is limited to continuous-time setup`
> - Thank you for raising this point. We want to clarify that the primary contributions of our paper are: (1) presenting a unified view of several existing optimizers (SOAP, Muon, Adam, Shampoo, etc.) through the lens of structured FIM approximation, and (2) using this perspective to design principled, memory-efficient optimizers with strong empirical performance. These contributions do not rely on a full convergence theory, and we explicitly state in Section 7 that developing such a theory is technically challenging and are outside the scope of this work.
> - Convergence analysis for structured-FIM optimizers is inherently case-specific. Even among existing methods (e.g., Adam, Muon, SOAP), their discrete-time analyses differ substantially and require distinct proof techniques. As such, proving convergence for a single optimizer (e.g., Alice) does not lead to general insights and does not extend to other structured-FIM methods. For instance, constructing a low-rank SOAP variant following the same design principle would already necessitate an entirely separate analysis.
> - A general convergence framework that handles all structured-FIM optimizers—continuous or discrete—would require new analytical tools and would itself constitute a substantial, standalone contribution. As discussed in our response to **reviewer mnjC**, even the continuous-time case poses nontrivial challenges, and extending this to the discrete-time regime introduces additional technical difficulties.
>
> - Regarding the discrete-time convergence of Alice specifically, we were not able to obtain a complete proof due to the combination of a low-rank Adam-like update and the compensation mechanism. We believe a unified discrete-time treatment of structured-FIM optimizers is a valuable direction for future work.
>
> 2. `the use of one-step refinement in theorem 3.4 is not well justified`
> - Thank you for raising this point. The reason we have to use one-step refinement is that the analytic solution is intractable.
> The choice of initialization of one-step refinement in Theorem 3.4 is not arbitrary. Initializing the Kronecker factors with identical blocks is the simplest setting that enables a closed-form solutions. This initialization along without future refinement already lead to Muon optimizer as discussed in proposition 3.3. One-step future refinement guarantees to improve the FIM reconstruction quality and should potentially lead to better performing optimizer. This is further supported empirically in LLaMA pretraining (Table 2), where Alice (i.e. low-rank version of optimizer from Theorem 3.4) indeed improves over Muon.
>
> - In principle, one could consider applying multiple refinement steps. However, beyond the first step, we were not able to derive a closed-form solution that leads to a practical and efficient optimizer update. Thus, one-step refinement is a reasonable compromise.
>
> 3. `limited generalization to other tasks`
> - Thank you for the suggestion. We agree that evaluating on additional tasks and domains would further strengthen the empirical evidence. In this work, our primary focus is on **LLM pretraining**, and we intentionally align our evaluation with the commonly adopted benchmark used in prior work (e.g., [1,2,3,4]). Using this benchmark helps ensure fair comparison, avoids unintentionally weak baselines, and makes our results directly comparable to previous studies on optimizer design for LLMs.
>
> - Our computational resources limit us to training models up to roughly 1B parameters, which is consistent with many prior optimization studies. Scaling beyond this requires non-trivial FSDP engineering for structured optimizers, and we plan to leave this for future work.
>
> - Regarding finetuning, our goal in this paper is not to propose a finetuning algorithm, but rather to study optimizer design in the pretraining regime. That is why we only provide limited finetuning experiment. But we agree that testing the performance in finetuning domain are indeed valuable, and may lead to more principled finetuning algorithm.
>
>
> [1] Zhao, Jiawei, et al. "Galore: Memory-efficient llm training by gradient low-rank projection." arXiv preprint arXiv:2403.03507 (2024).
>
> [2] Lialin, Vladislav, et al. "Relora: High-rank training through low-rank updates." arXiv preprint arXiv:2307.05695 (2023).
>
> [3] Chen, Xi, et al. "Fira: Can We Achieve Full-rank Training of LLMs Under Low-rank Constraint?." arXiv preprint arXiv:2410.01623 (2024).
>
> [4] Zhang, Haochen, et al. "Breaking the Frozen Subspace: Importance Sampling for Low-Rank Optimization in LLM Pretraining." The Thirty-ninth Annual Conference on Neural Information Processing Systems. 2025.

---

> ### Author Response · Authors · 2025-11-20
> **Author Response 2**
>
> 4. `Profiling the time consumption of each time step`
> Thanks for raising this point. The following table provide a detailed time consumption averaged over the 130M LLaMA training.
> | Component                          | Average Time (seconds) |
> |-----------------------------------|--------------------------|
> | Projection matrix update (1 per 200 steps) | 6.35 × 10⁻⁵              |
> | Project the gradient                | 3.60 × 10⁻⁵              |
> | Projection back                     | 4.30 × 10⁻⁵              |
> | Compensation step                   | 3.10 × 10⁻⁴              |
> | Adam step                           | 7.70 × 10⁻⁵              |
>
> - We can observe the compensation step consumes 10 times higher time. The main reason is because (1) the projection/projection-back and adam steps are all computed with low-rank (e.g. 0.25 of the original rank); and (2) the projection matrix update is only executed once per 200 steps, effectively amortizing the cost. However, the compensation step, aiming at adding back the information loss, are computed with remaining rank (e.g, 0.75 of the original rank), explaining why it consumes most of the time.
>
> - Further optimization of this step may not be easy, since the current compensation step only relies on simple operations like computing the norm. One possible direction is to perform this step once per K steps. But this will inevitably impact the performance. Or we can also combines the error-feedback mechanism [5] + performing compensation step onece per K steps. The former can partially mitigate the negative impact of the low-rank projection.
>
> 5. `Random subspace switching`
> Thank you for the question. We do not claim that the proposed switching strategy is optimal. Our intention is to present a practical heuristic that performs well empirically. We also evaluated an alternative strategy based on importance sampling of singular vectors proportional to their singular values (similar to SARA [6], we have included a discussion about SARA in appendix I.5). The performance of this approach was nearly identical to that of our proposed method. Given the similar empirical behavior and the simplicity of random switching, we chose the latter for clarity and efficiency.
>
> We hope the clarifications above address your concerns and assist your assessment, and we are happy to elaborate further if needed.
>
> [5] Robert, Thomas, et al. "Ldadam: Adaptive optimization from low-dimensional gradient statistics." arXiv preprint arXiv:2410.16103 (2024).
>
> [6] Zhang, Haochen, et al. "Breaking the Frozen Subspace: Importance Sampling for Low-Rank Optimization in LLM Pretraining." The Thirty-ninth Annual Conference on Neural Information Processing Systems. 2025

---

> > ### Author Response · Authors · 2025-12-02
> >
> > We appreciate the valuable feedbacks and suggestions from the reviewer. We are glad that our response has successfully addressed your concerns and the rating has been updated to 8 before.

---

### Official Review · Reviewer_mnjC · 2025-10-28

**Soundness:** 2
**Presentation:** 2
**Contribution:** 3
**Rating:** 6
**Confidence:** 3

**Summary:**

In this paper, the authors have proposed a unified framework for both memory and computational-efficient optimizers for LLMs. To achieve this goal, the authors have used the structured Fisher information matrix (FIM) approximation. Many existing optimizers, such as Adam and Shampoo, are approximations to the gradient descent update. FIM, on the other hand, is replaced by certain structured forms, such as block diagonal and Kronecker product matrices.

**Strengths:**

The strengths of this paper are summarized as follows:

1. This paper has a novel optimizer design. Row and Column Scaled SGD (RACS) introduces a minimal overhead two sided scaling method. It achieves strong results with almost no additional memory.

2. Empirically, the authors have shown experimental results on different model scales, and they are compared with many baselines, such as the well-known Adam and Galore. The experimental results have shown it has consistent performance improvements.

3. Theoretically, the authors have proved the convergence of Alice under the continuous-time setup.

**Weaknesses:**

The weaknesses of this paper are summarized as follows:

1. Theoretically, the authors have given the convergence analysis only in the continuous time setting and “leave the general treatment of the convergence to future work”. It would be better if the authors could give a brief discussion about what else needs to be done, what the difficulties are, and what the differences are with the continuous-time setup.

2. Empirically, the authors only analyze “pre-training LLaMA on the C4 dataset”. It would be better to include more evaluations in a more diverse domain. Also, the model sizes are small. Nowadays, people are more interested in larger LLMs. It would be better to include the experimental results on larger models.

**Questions:**

Galore has numerous follow-up works, such as Galore 2 [1], Golore [2], and Sara [3]. Could the authors give a comparison with these works?


[1] DiJia Su, Andrew Gu, Jane Xu, Yuandong Tian, and Jiawei Zhao. "Galore 2: Large-scale llm pre-training by gradient low-rank projection." arXiv preprint arXiv:2504.20437 (2025).

[2] Yutong He, Pengrui Li, Yipeng Hu, Chuyan Chen, and Kun Yuan. "Subspace optimization for large language models with convergence guarantees." ICML'25.

[3] Haochen Zhang, Junze Yin, Guanchu Wang, Zirui Liu, Tianyi Zhang, Anshumali Shrivastava, Lin Yang, and Vladimir Braverman. "Breaking the Frozen Subspace: Importance Sampling for Low-Rank Optimization in LLM Pretraining". NeurIPS'25.

---

> ### Author Response · Authors · 2025-11-20
> **Author Response**
>
> We greatly appreciated the reviewer's effort to provide valuable suggestions. Here are our responses, and hope they can clarify the confusions.
> 1. `Why continuous-time convergence analysis? Any difficulties with discrete time?`
> Thank you for raising this point. In principle, there are two ways to study convergence within our structured-FIM framework:
> (1) analyze each optimizer case-by-case in both continuous and discrete time, or
> (2) develop a unified convergence theory that covers the entire class of structured-FIM optimizers.
> Both directions present substantial technical challenges.
> - **case-by-case discrete study** This analysis does not scale across different FIM-structured optimizers.
> For example, optimizers that fall naturally under our FIM framework—such as Adam, SOAP, Muon, and Alice—their discrete-time convergence analyses differ completely. For instance, the discrete arguments used for Adam and Muon rely on very different techniques [1,2]. Hence, a unified case-by-case discrete analysis across all FIM-structured optimizers is a very challenging problem.
> - **general continuous-time study** A general convergence theory for structured FIM is difficult even in continuous time.
> Our continuous-time analysis for Alice already depends heavily on the specific update structure of Alice. Under different structural assumptions (SOAP vs. Muon vs. Alice), the induced dynamics differ substantially, and there is currently no known way to derive a single continuous-time formulation that all such optimizers satisfy. Extending this to discrete time further complicates the analysis.
> - ** discrete-time analysis introduce additional challenges** Continuous-time analysis tracks the instantaneous change of a Hamiltonian; if its derivative is negative, descent is guaranteed. In discrete time, however, descent depends critically on the step-size, curvature of the loss landscape, and interaction between projection operators and discretization error. Moreover, discrete-time analyses in the literature typically require restrictive assumptions (e.g., bounded gradients, diminishing learning rates, uniform smoothness). In summary, continuous-time setup relies on less assumptions compared to the discrete-time setup, but discrete-time is closer to practice compared to continuous-time setup.
> - **summary** For these reasons, we focus on continuous-time analysis of a single optimizer, Alice, as a principled and meaningful first step. Developing a general discrete-time convergence theory for structured FIM would require handling substantially different updates. If achievable, such a result would represent a significant theoretical contribution on its own, and we believe it is better suited for dedicated future work.
>
> 2. `Expand the evaluation to more diverse domains`
> Thank you for this suggestion. We fully agree that broader evaluations would strengthen the empirical evidence. Our current scope is constrained by the computational resources available to us, and for this reason we intentionally focus our study on a clearly defined LLM pretraining scenario—hence we have “Optimizer Design for LLMs.” in our title. Our goal was to avoid overclaiming and to ensure that the experiments we include are rigorous within our scope.
> - **model sizes and dataset** We follow the exact experimental setup established in prior work (e.g., [3,4,5, 6]), which has become a commonly accepted benchmark for evaluating optimizer behavior in LLM training. Using this setup ensures fairness and comparability and helps avoid the pitfalls of introducing new benchmarks where baseline may be under-tuned.
> -**resource constraints** The largest model we can reliably train with our available resources is approximately 1B parameters. Scaling beyond this requires non-trivial FSDP implementation. Unlike elementwise optimizers such as AdamW, our structured optimizers require matrix operations (e.g., high-dimensional factored updates), which introduce challenges for distributed computation—such as communication patterns. We plan to leave this for the future work and scale the training beyond 1B parameters.
>
> [1] Chang, Da, Yongxiang Liu, and Ganzhao Yuan. "On the Convergence of Muon and Beyond." arXiv preprint arXiv:2509.15816 (2025).
> [2] Reddi, Sashank J., Satyen Kale, and Sanjiv Kumar. "On the convergence of adam and beyond." arXiv preprint arXiv:1904.09237 (2019).
> [3] Zhao, Jiawei, et al. "Galore: Memory-efficient llm training by gradient low-rank projection." arXiv preprint arXiv:2403.03507 (2024).
> [4] Lialin, Vladislav, et al. "Relora: High-rank training through low-rank updates." arXiv preprint arXiv:2307.05695 (2023).
> [5] Chen, Xi, et al. "Fira: Can We Achieve Full-rank Training of LLMs Under Low-rank Constraint?." arXiv preprint arXiv:2410.01623 (2024).
> [6] Zhang, Haochen, et al. "Breaking the Frozen Subspace: Importance Sampling for Low-Rank Optimization in LLM Pretraining." The Thirty-ninth Annual Conference on Neural Information Processing Systems. 2025.

---

> ### Author Response · Authors · 2025-11-20
> **Author Response 2**
>
> 3. `Comparison to related work`
> - **GaLore 2**: GaLore 2 primarily focuses on improving the practical scalability of GaLore. Its contributions include:
> (1) replacing exact SVD with fast randomized SVD for computing the projection matrix; (2) integrating quantization and extending the method to handle high-order tensors, and (3) providing a carefully engineered FSDP implementation to reduce memory footprint.
> These modifications substantially improve usability, but the core mathematical update rule remains identical to GaLore. Under our framework, GaLore and GaLore 2 are both special cases of Alice, and all statements in our paper regarding GaLore transfer directly to GaLore 2. Importantly, the practical techniques introduced in GaLore 2 (e.g., randomized SVD, quantization, FSDP optimizations) are orthogonal to Alice and can be combined to improve scalability as well.
>
> - **GoLore** GoLore addresses two limitations of GaLore: (1) the theoretical convergence issue identified in the GaLore update, and (2) the cost of SVD projection, by replacing it with a random projection. Despite these improvements, GoLore may underperform even compared to GaLore for pretraining tasks as shown in Fig 1 of GaLore 2 paper.
> Methodology-wise, it still produces low-rank parameter updates. In contrast, Alice preserves low-rank memory while producing full-rank updates via our compensation step, which is theoretically justified (Theorem G.9). Thus, the two methods pursue different design philosophies: GoLore redueces the memory and computation by using random matrix, whereas Alice improves the expressiveness of the final update. Empirically, GoLore has so far only been evaluated on finetuning and synthetic tasks. GaLore 2 additionally reports pretraining results with random projection matrices (Fig. 1), where performance degrades noticeably even relative to GaLore. Since Alice consistently outperforms GaLore in pretraining, the gap would likely be even larger relative to GoLore.
>
> - **SARA**  Sara is a concurrent work that aims to improve the quality of the projection by using importance sampling of singular vectors. This shares the similar intuition with our subspace switching mechanism, although the techniques differ: Sara relies on importance sampling of singular vectors, while Alice mixes random sampling of less important directions with leading directions. We want to emphasize that subspace switching is only one of the three components in Alice. Based on the ablation study in Table 4 of our paper, the theoretically optimal compensation step offers the largest improvement in performance.
> In terms of performance, Alice outperforms Sara under the same settings: 130M model: Sara 22.22 vs. Alice 21.95 and 350M model: Sara 17.25 vs. Alice 16.61. We have revised our paper to include this discussion.
>
> We hope the clarifications above address your concerns and assist your assessment, and we are happy to elaborate further if needed.

---

> > ### Comment · Reviewer_mnjC · 2025-11-26
> > **Thank you for your response!**
> >
> > I acknowledge that I have read the responses from the authors. All of my concerns are addressed. I choose to maintain my original rating.

---

> > > ### Comment · Area_Chair_42k1 · 2025-11-26
> > >
> > > Dear Reviewer mnjC,
> > >
> > > Thank you very much for your time and the initial review.
> > >
> > > If authors managed to address all of your concerns and you don’t have any remaining issues with the submission, please justify your evaluation/score. Otherwise, please discuss with the authors about any remaining concerns.
> > >
> > > Thank you.
> > >
> > > Best,
> > > AC

---

### Official Review · Reviewer_CJCW · 2025-11-05

**Soundness:** 3
**Presentation:** 3
**Contribution:** 3
**Rating:** 6
**Confidence:** 5

**Summary:**

Memory, sample efficiency, and throughput are key performance metrics for optimization algorithms, and algorithmic development is essential for mitigating the cost of LLM training. This work leverages an insightful lens of Fisher Information Matrix (FIM) preconditioning (Natural Gradient Descent view) to study successful optimizers such as Adam, Shampoo, Muon, and Soap, and unifies them all under this lens by studying the specific structure of FIM approximation (different variants of Kronecker product and block diagonal structure) that each of these optimizers is made of. Interestingly, the update rule of the one-sided Soap algorithm is also derived and analyzed by leveraging block-diagonal structure, and together the one-sided Soap update rule plus the derived structure is referred to as Generalized Adam (GAdam).

Inspired by FIM unification, this work hence argues that exploration of novel structures for FIM approximations is the key design principle for systematically developing new optimizers to further balance the optimizers' performance metrics, especially memory vs sample efficiency. Building on this design principle, the paper proposes two novel optimization algorithms. The first proposed algorithm is called RACS, where the preconditioner is structured as a Kronecker product of two diagonal matrices, and by minimizing the FIM approximation error (measured in Frobenius norm), the two diagonal matrices are derived subsequently. This optimizer hence reduces the second moment memory significantly and matches the memory complexity of AdaFactor (for instance, Adam's second moment needs O(mn) while RACS needs O(m+n)).

Moreover, this work also proposes another novel low-rank optimizer called ALICE, achieving GaLore-like memory while significantly improving both sample efficiency and throughput. ALICE extends the GAdam algorithm (one-sided Soap update rule) to low-rank subspace optimizer settings by maintaining only the top r eigenvectors of the full-rank projection in GAdam, and then the projected gradients are accumulated in the subspace akin to GaLore. Subspace switching and compensation methods are also proposed to account for the residual update term that remains orthogonal to the projection matrix and is wiped out during projection.

The two proposed algorithms are validated on LLaMA pre-training and GLUE benchmarks compared to AdamW, GaLore, Fira, Apollo, and Muon. The reported results indicate that these memory-efficient optimizers (ALICE and RACS) not only match the final validation perplexity of the best baseline but both improve the wall-clock time by a factor of 2 compared to Adam.

**Strengths:**

1. The paper is well-presented and clear. The proposed algorithms are derived based on rigorous theoretical principles, and experiments are comprehensive and profile all key performance metrics (throughput, memory, and perplexity) across the LLaMA model family (from 60M to 1.3B). Comprehensive ablations verify the effectiveness of the key algorithmic components introduced for the two proposed optimizers.

2. This paper makes it clear that the most adopted and successful optimizers in deep learning indeed are different choices of FIM approximation. Principled design of optimizers based on exploring the space of FIM approximation structures is a very clever and strong idea. Finding principles for one-sided Soap based on this framework is an interesting contribution.

3. Both proposed optimizers, ALICE and RACS, achieve consistent speedup by a factor of 2 (measured in wall-clock time) and significant memory savings compared to AdamW. These gains represent very strong results, and upon reproduction of these results, the ALICE and RACS algorithms have the potential to be adopted as mainstream optimizers for LLM training based on reported results. Especially considering the benchmark of current state-of-the-art optimizers in similar pre-training settings [1], the speedup of the best SOTA optimizer compared to Adam is around 1.1-1.4, and Muon is among the best-performing optimizers. Interestingly, the paper reports both ALICE and RACS outperform Muon significantly while using less memory.

**Weaknesses:**

### Critiques on the FIM Framework (Section 3)

Applying the Shampoo preconditioner directly on gradients can recover spectral normalization (whitening) as well, and hence whitening can be understood without leveraging Proposition 3.3. Moreover, [3] already finds the optimal solution to Equation 2 (line 147), where the optimal R and L are computed as principal singular vectors of reshaped F. Then [3] leverages power iteration for approximating the optimal solution of Equation 2, and hence minimizing the proposed upper bound in Theorem 3.2 does not provide significant additional insight when compared to established results in [3]. For Theorem 3.4, if we assume all D_i are equal, then the structure would be equal to the whitening structure that is used to derive Equation 3, and hence the derivation of Equation 6 appears redundant. Moreover, since GAdam appears to be equivalent to one-sided Soap, and this algorithm is actually discussed in the original Soap paper, the justification for not referring to GAdam as one-sided Soap is unclear. While the derived principle has novelty, the update rule remains the same.

### Novelty of RACS Compared to AdaFactor

If we let the initial S and Q be identity matrices with proper scaling, then one iteration of the fixed point procedure in Equation 11 would indeed recover the AdaFactor update. Moreover, when applying the structure in Equation 10 to gradients as in Equation 1, we can write:
$\text{Vec}(\Delta W) = -\lambda (S \otimes Q)^{-1/2} \text{Vec}(G)$. Since both S and Q are diagonal, $S \otimes Q$ would be diagonal as well, and hence we can write $\Delta W = -\lambda \frac{G}{\sqrt{V}}$, where $V = qs^\top$. Since Proposition 4.1 shows that q and s converge to the left and right singular vectors of $G^2$ up to scaling factors, this shows that V would be a rank-one SVD approximation of $G^2$, and the iterative fixed point procedure of RACS is essentially a power iteration that approximates those principal vectors. Moreover, the original AdaFactor (Algorithm 4 in [4]) similarly maintains momentum on q and s. Hence, RACS appears to be very similar to AdaFactor. RACS performs 5 iterations of Equation 11, while AdaFactor performs one iteration. Therefore, the novelty of the RACS update rule could be considered limited, as RACS appears to be another rank-one approximation method of the second moment, which AdaFactor has already explored and studied in detail.

### Reproducibility

The code has not been made available, and considering the significant speedup reported in this work and the potential for improving upon the best existing optimizers in the field, making the source code and implementation public would greatly benefit the research community and enable better evaluation of this paper. Even sharing the source code for the 130M setup would be highly valuable.

**Questions:**

1. Could the authors please address my critique regarding RACS, and particularly explain the key algorithmic differences between RACS and AdaFactor beyond performing more steps of power iterations? This is indeed a very important concern to address, as AdaFactor does not achieve anywhere close to the RACS speedup compared to AdamW, and having more power iterations as the only key algorithmic component does not seem sufficient to explain these gains.

2. Could the authors please provide a detailed comparison of the compensation step in ALICE to LDAdam? Line 467 mentions Fira as the closest work for compensation, but LDAdam has also made significant contributions regarding this compensation mechanism, which merits discussion and comparison.

3. Proposition 4.1 in this work appears to be a diagonal version of Proposition 1 (Section 3.1 in [3]). It would be interesting to see what additional challenges the proof of Proposition 4.1 introduces compared to Proposition 1 in [3], and how these relate to power iteration convergence.



[1] Wen, Kaiyue, et al. "Fantastic pretraining optimizers and where to find them." arXiv preprint arXiv:2509.02046 (2025).

[2] Robert, Thomas, et al. "Ldadam: Adaptive optimization from low-dimensional gradient statistics." arXiv preprint arXiv:2410.16103 (2024).

[3] Morwani, Depen, et al. "A New Perspective on Shampoo's Preconditioner." arXiv preprint arXiv:2406.17748 (2024).

[4] Shazeer, Noam, and Mitchell Stern. "Adafactor: Adaptive learning rates with sublinear memory cost." International Conference on Machine Learning. PMLR, 2018.

---

> ### Author Response · Authors · 2025-11-20
> **Author Response**
>
> We greatly appreciated the reviewer's effort to provide valuable suggestions. Here are our responses, and hope they can clarify the confusions.
> 1. `Whitening can be understood without leveraging proposition 3.3...`
> We fully agree that whitening admits multiple interpretations (e.g., via Shampoo-style preconditioners), and our contribution is not to claim the novelty in deriving whitening itself. Our intention with Proposition 3.3 is to show that whitening naturally emerges as a structured approximation of the Fisher Information Matrix (FIM) and what the underlying structure is.
> The key message is that **many existing optimizers—including whitening-based ones—can be viewed through the common lens of FIM approximation**. Proposition 3.3 provides this conceptual bridge; it is not meant to assert that this is the only or original interpretation of whitening, but rather to demonstrate how it fits nicely into our FIM framework.
>
> 2. `Derivation of eq.6 is redundant`
> Yes, we agree and have revised the paper by directly citing proposition 3.3.
>
> 3. `Gadam and one-sided SOAP are identical`
> You are correct that the update rules are mathematically identical. Our original intention for using a different name was to highlight that the one-sided SOAP relies on a different structural assumptions than SOAP. As formalized in Theorem 3.4 and Theorem E.1. there structural distinction leads to a more nuanced interpretation than the “apply one-sided eigen-projection” description in the original SOAP paper.
> However, we agree that using a separate name may cause unnecessary confusion. To avoid ambiguity, we have renamed the method to one-sided SOAP (OSOAP) in the revised version. The equivalence of the update rules is now explicitly acknowledged in the text.
>
> 4. `RACS v.s. Adafactor`
> Thank you for raising this important point. RACS is indeed related to Adafactor, and also to Adapprox, as mentioned in the paper, but it is not equivalent to either of them, conceptually or empirically. We clarify the distinctions here.
> - **Adafactor**: Adafactor obtains its update by minimizing the I-divergence between the empirical second moment and a Kronecker-factored approximation with (S,Q), yielding a closed-form analytic solution.
> In contrast, RACS minimizes the Frobenius reconstruction error to the empirical FIM, which leads to a different objective and requires an iterative fixed-point procedure. Importantly, the two methods coincide only in the degenerate case of a single fixed-point iteration with special initialization. With multiple steps (we use 5 in our experiments), the resulting $S$ and $Q$ deviates substantially from Adafactor’s closed-form solution, especially combined with EMA. We provide a figure in the appendix I.5 of the revised paper, quantifying how large these deviations become during the training. We compute a term called relative difference as $\frac{\|\Delta_{RACS}-\Delta_{Adafactor}\|}{\|\Delta_{Adafactor}\|}$ averaged over all parameters for a 60M LLaMA model, where $\Delta$ is the parameter update. We observe that this error is small at the beginning of training, and starts to increase up to 0.2 at the end of the training. This suggests the difference between RACS and Adafactor deviates as the training progresses, explaining the performance difference.
> - **Adafactor and SinkGD** We also want to point-out a recently published optimizer SinkGD [1], which relying on iterative normalization of rows and columns of the gradient matrix. This is even more closely related to Adafactor, since one-step normalization is equivalent to Adafactor. However, they also report better performance than Adam. Therefore, multiple-steps of a procedure can indeed lead to large difference compared to single step of this procedure.
> - **Adapprox**: Adapprox also searches for a low-rank approximation to the second moment, but differs from RACS in two key ways:
> Adapprox uses rank-k approximation with no guarantee that the reconstruction remains positive; thresholding is required and can harm stability. RACS uses rank-1 with guaranteed positivity (Prop. 4.1 / Thm. G.4). More importantly, RACS also involves the EMA of $S$ and $Q$. Therefore, the accumulated $S$ and $Q$ cannot be regarded as the top singular vectors. We leave the analysis of the effect of EMA to the future work.
> - **summary** RACS can be viewed as combining (i) the rank-1 approximation idea from Adapprox with (ii) the EMA strategy from Adafactor, but the deviates from both optimizer in practice.
>
> [1] Scetbon, Meyer, et al. "Gradient Multi-Normalization for Efficient LLM Training." The Thirty-ninth Annual Conference on Neural Information Processing Systems.

---

> ### Author Response · Authors · 2025-11-20
> **Author Response 2**
>
> 1. `Comparison to LDAdam`
> Thank you for pointing out this related work. LDAdam and our compensation step in Alice share a similar motivation—addressing the information loss induced by low-rank projection—but they differ substantially in how they tackle the problem.
> - **LDAdam** LDAdam modifies the gradient via an error-feedback buffer that accounts for the mismatch between the current and previous projection bases $U_t$ and $U_{t-1}$. The momentum and second moment are then computed on the projected modified gradient. It has the form of $update = Proj_back(Adam(Proj(modified_G)))$. Ultimately, the parameter update remains **low-rank**, because it is derived entirely from projected statistics. Thus information loss is unavoidable, and the error-feedback has no optimality guarantee.
> - **Alice** In contrast, Alice applies compensation directly to the projected update, not to the gradient. The update takes the form
> $update = Projback(Adam(Proj(G))) + compensation$. This produces a **full-rank update**, because the compensation term explicitly reconstructs the lost information. Theorem G.9 shows that this reconstruction step is optimal in the sense of minimizing the projection-induced error under the chosen structural assumption. This is fundamentally different from LDAdam, where the final update always remains low-rank despite with the error-feedback. We add relevant discussion in appendix I.5 in the revised paper.
>
> 2. `Proof used in prop 1 in [3] v.s. prop 4.1 in our paper`
> Thank you for noting the connection. At a high level, both results concern optimal Kronecker-structured approximations, and Proposition 1 in [3] may appear more general. However, the proof technique used in [3] cannot be directly applied to our setting. The key issue is that their argument fundamentally relies on unconstrained rank-1 factorization, whereas Proposition 4.1 concerns a constrained optimization problem.
> - **Proof outline in [3] prop 1** The prop 1 in [3] relies on the following steps:
> (1) Use Lemma 4 to convert the optimal Kronecker product problem into an unconstrained rank-1 approximation.
> (2) Apply classical linear algebra to conclude that the top singular vectors give the optimal solution.
> (3) Use power iteration to compute these singular vectors; this yields an update rule similar in spirit to ours (but without diagonal constraints).
> - **why it cannot be used in our case** In Proposition 4.1 of our paper, the matrices S and Q are restricted to be diagonal. Under this structural constraint, the vectors $vec(S)$ and $vec(Q)$ (i.e. $vec(L)$ and $vec(R)$ in Lemma 4 [3]) are no longer freely to be optimized. Their pattern is fixed to be diagonal only. Consequently, the reduction to an unconstrained rank-1 approximation (step (2)) is invalid: the top singular vectors are not feasible solutions under diagonal constraints. Because step (2) is essential for the argument in [3], their proof technique cannot be carried over.
> - **our approach** Due to this mismatch, Proposition 4.1 is proved via a direct analysis of the constrained optimization problem, rather than through the singular-vector argument used in [3]. The resulting update rule therefore arises from a completely different reasoning path.
> 3. `Opensourcing the code`
> Thanks for your interests in our code, we plan to fully opensource it upon acceptance.
>
> We hope the clarifications above address your concerns and assist your assessment, and we are happy to elaborate further if needed.

---

### Author Response · Authors · 2025-12-03
**Summary for AC**

---
`Reviewer CJCW`
1. **Novelty of RACS vs Adafactor**:
We have provided explanations that a seemingly small change can result in big difference in training. We also plot the matrix update difference of Adafactor and RACS, which can differ up to 20% at each step.

2. **Alice compared to LDAdam**:
We also provided detailed explanation on their differences. The quick summary would be: each update step in LDAdam is still low-rank in nature, but the update in Alice is guaranteed to be full-rank.
3. **Proof technique of proposition 4.1 v.s. proposition 1 in [3]**:
We explicitly explain why the technique in proposition 1 cannot be applied to 4.1, since proposition 1 only considers unconstrained optimization problem but proposition 4.1 assumes a structural constraint. So a new technique is required.

The reviewer does not have a chance to reply back. But we believe that all the concerns are addressed to our best effort.

---

`Reviewer mnjC`:
1. **Continuous convergence analysis is limited and more experiment should be added**: The main scope of this paper is to propose a potential framework candidate to unify existing optimizers, and provide possible principles for future optimizer design. We have clearly stated the limitation in the paper. For large scale experiment, we did not provide 7B experiment due to the limited resources. But we do include a 1B v.s. 7B preliminary result to support the effectiveness.
2. **Comparison to related work**: We have included a detailed comparison to all the methods mentioned by the reviewer, which some of them share similar intuition, but the resulting methodology is completely different.

The reviewer does not have a chance to reply back. But we believe that all the concerns are addressed to our best effort.

---

`Reviewer phAX`:
This reviewer shares similar concerns as Reviewer mnjC, with one additional concern regarding the profiling of the optimizer. We have provided detailed profiling statistics of each step in Alice to identify the key bottleneck, and propose some possible solutions.

**Despite the reviewer did not have a chance to reply, but they raise the score from 6 to 8**

---

`Reviewer 1U1o`:
The main concern is that the experiment is smaller than 1B, which we have addressed it for Reviewer phAX and mnjC.

The other concern is that we do not have graphical illustration. We did not realize that the revised version can have 1 additional page. After the reminder from the reviewer, we added two illustrations in the revised paper for easier understanding.

**The reviewer remains their original rating of 8**

---

### Meta-Review · Area_Chair_jdPU · 2026-01-11

**Summary:**

This paper proposes a unifying framework that views many modern LLM optimizers as structured Fisher Information Matrix (FIM) approximations, and uses this lens to derive two new memory-efficient optimizers that achieve significantly faster convergence than Adam in LLaMA pre-training.

The reviewers reached consensus on accepting this paper primarily based on its novel theoretical framework unifying existing LLM optimizers through structured Fisher Information Matrix (FIM) approximation and the strong empirical results demonstrating 2× faster convergence than Adam. While initial concerns were raised about the novelty of RACS relative to AdaFactor, the distinction of Alice from LDAdam, and the theoretical convergence analysis being limited to continuous time rather than discrete time, the authors' rebuttal successfully addressed most of these issues by clarifying the algorithmic differences (including quantitative analysis showing up to 20% deviation between RACS and AdaFactor updates), explaining that Alice produces full-rank updates while LDAdam remains low-rank, and acknowledging the limitations of their theoretical scope.

**Reviewer Concerns:**

Reviewers broadly agree that the structured FIM perspective is novel and insightful, providing a principled unification of existing optimizers.
Overall, the rebuttal successfully addressed most reviewer concerns.  The authors convincingly clarified the novelty of RACS relative to AdaFactor raised by reviewer CJCW, distinguished Alice from LDAdam, explained why prior proof techniques do not apply to Proposition 4.1, and resolved naming confusion by aligning GAdam with one-sided SOAP; the only remaining issue is that code release is deferred until acceptance. For Reviewer mnjC, all concerns—limited convergence analysis, experimental scope, and missing related-work comparisons—were adequately addressed, and the reviewer explicitly acknowledged this. For Reviewer phAX, detailed profiling and clearer justification of the algorithmic design choices addressed the main practical concerns, but the lack of discrete-time convergence guarantees and broader empirical validation remain outstanding. For Reviewer 1U1o, concerns about clarity, illustrations, and experimental scale were largely resolved, with only the absence of very large-scale (e.g., 7B) pretraining experiments remaining as a minor open point.

**Reviewer Scores:**

The most concerning issue among reviewers was the limited theoretical and empirical scope relative to the strength of the claims. For example the convergence analysis is restricted to continuous time (and only for Alice), while experiments focus primarily on LLaMA/C4 and models up to ~1B parameters. Although reviewers generally accept the authors’ explanations and resource constraints, they encourage (i) clearer positioning of theory as partial and exploratory, (ii) stronger discussion of discrete-time challenges and implications for practice, and (iii) broader empirical validation (larger models, architectures, or domains) in future work. A secondary but recurring concern is clarifying novelty relative to prior work (e.g., AdaFactor, LDAdam, SOAP), which the rebuttal largely addresses but could be communicated more directly in the paper.

---

### Decision · Program_Chairs · 2026-01-26

Accept (Poster)